# Identifying clinically relevant cell state interactions in the tumor microenvironment of IDH-mutant gliomas using CSI-TME

Arashdeep Singh [ID][1], Bharati Mehani [ID][2,4], Vishaka Gopalan[1], Sushant Puri [ID][3], Kenneth Aldape[2] & Sridhar Hannenhalli [ID][1✉]

## Abstract

Tumor microenvironment (TME) is characterized by a milieu of distinct cell types that exist in heterogenous transcriptional states across tumors. Functional interactions among these cell states drive tumor progression and therapy response. Systematic characterization of functional cell-state interactions (CSIs) remains challenging due to the paucity of scRNA-seq cohorts with clinical information on one hand, and the lack of cellular context in bulk RNA-seq cohorts on the other. We present CSI-TME, a computational pipeline that extends the concept of gene interactions, such as synthetic lethality, to cell states, to infer prognostic CSIs by directly leveraging large cohorts of bulk transcriptomic datasets. Applied CSI-TME to IDH-mutant gliomas, we identified a highly reproducible cell-state interaction network (CSIN) that is predominantly pro-tumor and differentially activated in IDH-mut astrocytoma versus oligodendroglioma. Malignant cell states within the CSIN resemble multiple neuronal lineages, including astrocyte-like and oligodendrocyte-progenitor-like programs, and reveal key interactions between glioma stem cells and T cells. CSIN stratifies patient response to immune-checkpoint blockade therapy. Roughly 20% of CSIs involve direct ligand–receptor communication, and co-localize in spatial-transcriptomic datasets, most notably for a pro-tumorigenic interaction between tip-like endothelial cells and hypoxic malignant cells supported by multiple ligand–receptor interactions. Interestingly, anti-tumor CSIs correlated with oncogenic mutations are preferentially active in early stages of cancer, hinting at tissue homeostatic response. Overall, CSI-TME is a novel approach that, leveraging clinical bulk transcriptomic data, identifies prognostic CSIs and therapeutic ligand–receptor targets, while providing novel insight into how interactions among the cell states shape the TME in IDH-mutant glioma.

**Keywords** IDH-mut Glioma; Tumor Microenvironment; Cell State Interactions; Independent Component Analysis; Homeostasis
**Subject Category** Cancer

## Introduction

According to WHO CNS 2021 classification, diffuse gliomas in adults are broadly classified into two classes based on the mutation status of the gene Isocitrate Dehydrogenase (IDH) into IDH wild-type (IDH-wt) and IDH-mut—those with mutations in *IDH1* or *IDH2* enzymes (Reuss, 2023; Whitfield and Huse, 2022). IDH-mut gliomas are characterized by distinct metabolic, epigenetic, molecular, and histological features (Malta et al, 2024). IDH-mut gliomas are further classified into "Astrocytoma" (IDH-A) and "Oligodendroglioma, (IDH-O) (Reuss 2023; Whitfield and Huse 2022). IDH-mut tumors respond favorably to the standard of care therapies, including surgical resection followed by radiation and chemotherapy (Alshiekh Nasany and de la Fuente, 2023). Treatment with FDA-approved IDH mutation inhibitor Vorasidenib has been shown to delay progression in select lower-grade gliomas (Cloughesy et al, 2025; Kamson et al, 2023; Mellinghoff et al, 2023). However, recurrence is common, with 54.1% of the cases recurring within 6.4 years, with increasing odds over longer follow-up periods (Bell et al, 2020; Miller et al, 2019), and treatment options thereafter are limited. Thus, there is a critical need to uncover additional mechanisms contributing to the heterogeneity and therapy resistance in IDH-mut gliomas.

Tumor microenvironment (TME) is an ecosystem of distinct immune, endothelial, stromal, and tumor cells (de Visser and Joyce, 2023), and the precise TME composition significantly impacts the tumor physiology (Bremnes et al, 2011; Qu et al, 2019), patient survival, and therapy response. For instance, despite similar expression profiles and developmental hierarchy of glial cells, IDH-A and IDH-O tumors have significantly different TME and prognosis (Venteicher et al, 2017). Recent studies have highlighted that the interactions between distinct cell types in the TME, mediated either via cell surface molecules or paracrine factors,

[1]Cancer Data Science Laboratory, National Cancer Institute, National Institutes of Health, Bethesda, MD, USA. [2]Laboratory of Pathology, National Cancer Institute, National Institutes of Health, Bethesda, MD, USA. [3]School of Medicine, Oregon Health & Science University, Portland, OR, USA. [4]Present address: Lombardi Comprehensive Cancer Center, Department of Oncology, Georgetown University, Washington, DC, USA. ✉E-mail: sridhar.hannenhalli@nih.gov

influence the clinical course as well as therapy response in gliomas (Sharma et al, 2023). T-cell dysfunction via PD1-PDL1 interaction with tumor cells (Han et al, 2020) and immune exclusion via TGF-beta signaling from cancer-associated fibroblasts are two well-known examples of such interactions (Batlle and Massagué, 2019).

Further, tumor cells, as well as other microenvironmental cell types, can exist in multiple transcriptional states in the TME (Abdelfattah et al, 2022; Ochocka et al, 2021; Tirosh et al, 2016), such as hypoxic tumor cells or exhausted T cells. Beyond a compositional characterization of TME in terms of cell types and their distinct transcriptional states, the co-occurrence of specific transcriptional states may be functionally critical. For instance, in glioblastoma, the secretion of interleukin-10 by *HMOX1+* myeloid cells state can derive T-cell exhaustion, jointly resulting in poor prognosis (Ravi et al, 2022).

To directly assess the clinical importance of joint activity of a pair of transcriptional states, we took inspiration from the analogous gene-centric concept, viz., synthetic lethality (Jerby-Arnon et al, 2014; Lee et al, 2018), whereby the simultaneous deletion of two specific genes, but not either one alone, is detrimental to cellular fitness, leading to several therapeutic targets. For instance, PARP1 and BRCA1/2 genes are deemed to be a synthetic lethal gene pair, and PARP inhibitors are specifically employed as a treatment for BRCA-mut breast cancer patients (Helleday, 2011). Analogous to synthetic lethality, our goal here is to identify pairs of transcriptional states of distinct cell types whose joint activity state tracks clinical outcome; we refer to such pairs of transcriptional states as functional cell state interactions, or CSIs. And like synthetic lethal gene pairs, that do not necessarily interact physically (e.g., PARP and BRCA), CSIs may be mediated by mechanisms other than direct ligand receptor pairing.

An ideal dataset to identify CSIs would be large cohorts of tumor scRNA-seq data with clinical outcome information. Unfortunately, such datasets are currently lacking for most cancer types, including IDH-mut gliomas. While bulk RNA-seq data for large IDH-mut glioma cohorts are available, methods to leverage such datasets to identify CSIs are currently lacking. In this work, extending the concept of synthetic lethality (Lee et al, 2018) and other more general types of functional gene interactions (Magen et al, 2019), we develop a data-driven computational pipeline (CSI-TME) to infer clinically relevant functional interactions between specific transcriptional states of distinct cell types in the TME (Fig. 1). CSI-TME leverages large cohorts of bulk tumor RNA-seq data by first deconvolving the data, identifying transcriptional states of each cell type, and then identifying cell state pairs that significantly track clinical outcome.

Applying CSI-TME to 425 IDH-mut glioma RNA-seq samples from TCGA, we identify a robust and reproducible network of transcriptional state interactions across various cell types. A majority (70%) of these interactions were pro-tumor, with a minority (30%) having anti-tumor associations. CSI-TME identified interactions that involved known states of glioma cells, such as astrocyte-like, oligodendrocyte progenitor-like, and cycling stem cells. The pro- and anti-tumor CSIs were significantly associated with the response to PD-L1 inhibition therapy in gliomas, highlighting the crucial role of cellular crosstalk in mediating successful immune response against cancer. We show that the subset of identified CSIs that are supported by ligand–receptor pairs exhibits significant spatial proximity in multiple IDH-mut glioma spatial transcriptomics datasets. CSI-TME thus additionally

provides a framework to prioritize ligand–receptor interactions with their predicted clinical outcomes. Integrating the identified CSIs with paired somatic mutation data, we observed that despite their overall under-representation in the CSIN, anti-tumor interactions were significantly over-represented in the samples bearing somatic mutations in IDH-mut glioma patients, specifically in the earlier stages of tumor progression, raising the possibility that such CSIs represent the TME's homeostatic response to somatic mutations. Overall, implementing a novel computational strategy, CSI-TME provides new insights into the cell state interactions in the IDH-mut glioma TME with a potential application in patients stratification for therapeutic interventions. The CSI-TME pipeline is available at https://github.com/hannenhalli-lab/CSI-TME.

## Results

### An overview of CSI-TME

A schematic of the key steps involved in CSI-TME is illustrated in Fig. 1, and the details are provided in Methods. Briefly, given a large cohort of bulk tumor transcriptome (RNA-seq) data and cell-type-specific marker genes for key cell types, CSI-TME first deconvolves the bulk expression data into their constitutive cell-type-specific expression profiles using CODEFACS (Wang et al, 2022). Next, for each cell type, CSI-TME infers their distinct cell states or gene expression programs based on independent component analysis (ICA), to obtain an IC-by-sample matrix for that cell type where each IC represent a distinct transcriptional state of a particular cell type (Figs. 1A and EV1). Finally, CSI-TME screens for the pairwise combinations of transcriptional states (ICs) between two distinct cell types whose simultaneous activity or inactivity (for instance, senescent T cells and stem-like malignant cells) significantly associates with patient survival using Cox proportional hazard test, controlling for the independent activities of the two states. We note that, CSI-TME does not calculate the correlation between two ICs from distinct cell types and consequently, interacting pair of cell states do not necessarily significantly co-occur. Instead, it identifies the IC pairs (from two distinct cell types) whose joint activity is significantly associated with the patient's survival. The output of CSI-TME is a list of tuples (C1, S1, C2, S2, B, D), where the co-occurrence of transcriptional state S1 of cell type C1 and transcriptional state S2 of cell type C2 in a specific co-activity bin B is associated with survival, where $D = +1$ indicates a positive log-hazard ratio, and $D = -1$ indicates a negative log-hazard ratio. For a given cell type, samples are partitioned into three equal-sized bins —Low, Med, and High—based on the score of the IC. We consider three co-activity bins (Fig. 1B) corresponding to co-inactivity of S1 and S2 (bin 1), co-activity of S1 and S2 (bin 9), and high activity of S1 and low activity of S2 (bin 3), with the converse symmetric case being included (bin 7).

### CSI-TME identifies distinct transcriptional states and their pro- and anti-tumor interactions in the TME of IDH mutant glioma

We integrated a publicly available scRNA-seq data (Abdelfattah et al, 2022) with a cohort of our in-house scRNA-seq data from

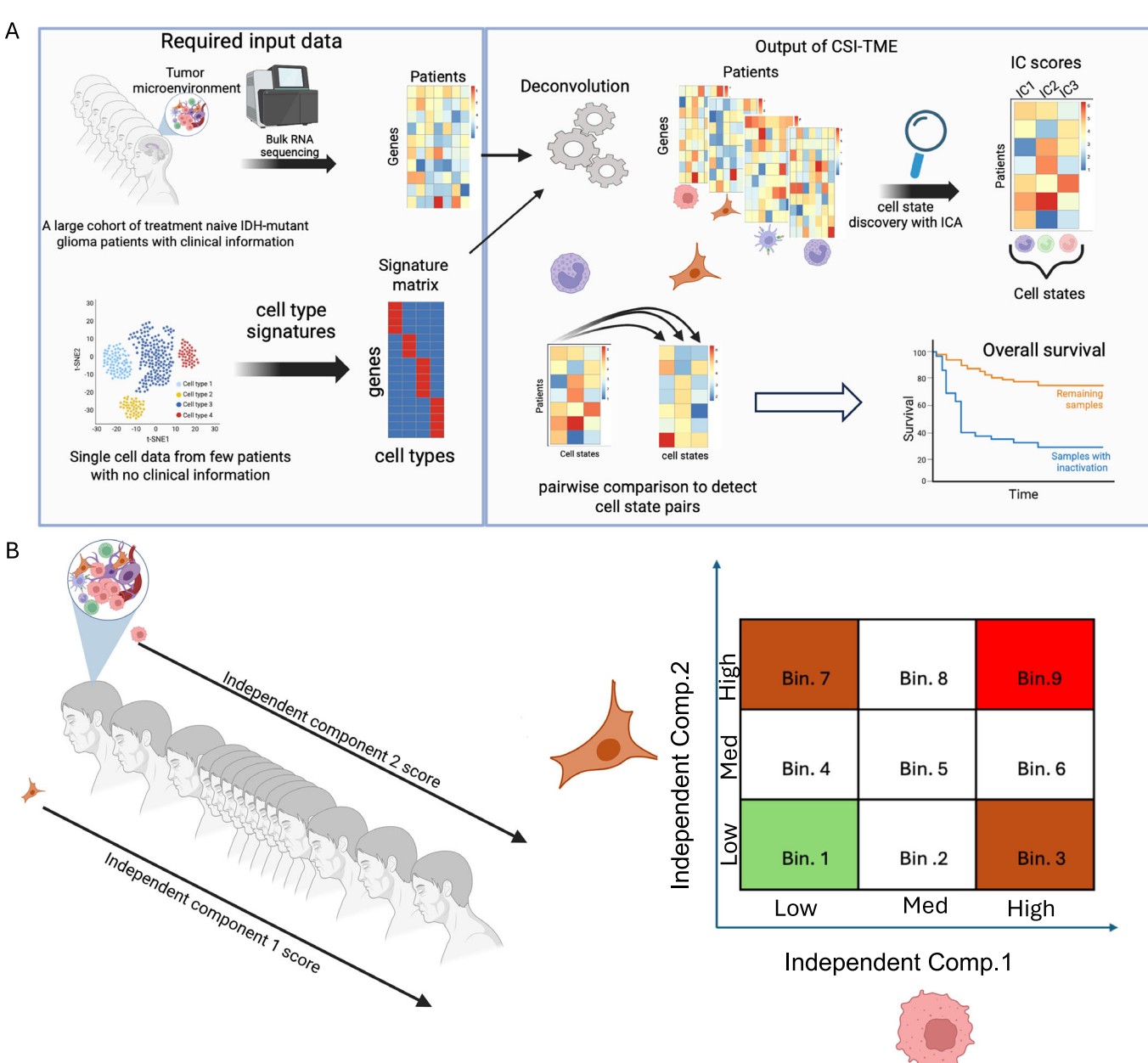

**Figure 1. A schematic of the key steps involved in CSI-TME.**

(**A**) Given (i) bulk transcriptomic data from a large cohort of patients along with survival data and (ii) cell type-specific gene markers inferred from cancer-specific single cell data, CSI-TME first deconvolves the data into cell type-specific gene expression matrices, and then infers cell states (independent components or ICs) for each cell types and represents each sample as cell type-specific IC vectors, where each IC score is deemed to quantify the abundance of distinct cell state. Finally, CSI-TME performs a Cox regression to model the overall survival of patients based on the joint activity of IC-pairs taken from two different cell types. To infer the joint activities of two ICs, we stratified the patients into two categories using an AND condition such that patients where both IC1 and IC2 levels belonged to their respective activity bin were assigned a value of 1 and remaining patients were assigned a value of 0 (for instance, in Bin.9, both IC.1 and IC.2 were upregulated and corresponding patients were assigned a value of 1). (**B**) For any two pairs of ICs, CSI-TME considers their joint activity in the following three categories: both ICs are low (Bin 1), both ICs are high (Bin 9) and either of the IC is low and other IC is High (Bin 3 and Bin 7).

glioma patients at NCI ("Methods"), with a total of 60,751 cells, and identified seven cell types based on the expression of established cell type-specific marker genes ("Methods"; Fig. 2A).

Using the seven-cell type-specific gene signatures, we used CODEFACS to deconvolve two large cohorts of bulk gene expression data from TCGA (discovery cohort; 425 samples) and

CGGA (validation cohort; 325 samples) IDH-mut gliomas, yielding seven cell type-specific expression profiles for each patient ("Methods"), covering, on average 9954 genes in TCGA and 11631 genes in CGGA. We next ascertained cross-cohort robustness of deconvolution as follows. Firstly, we found that in the deconvolved data, the top 5% highly expressed genes in each cell

 

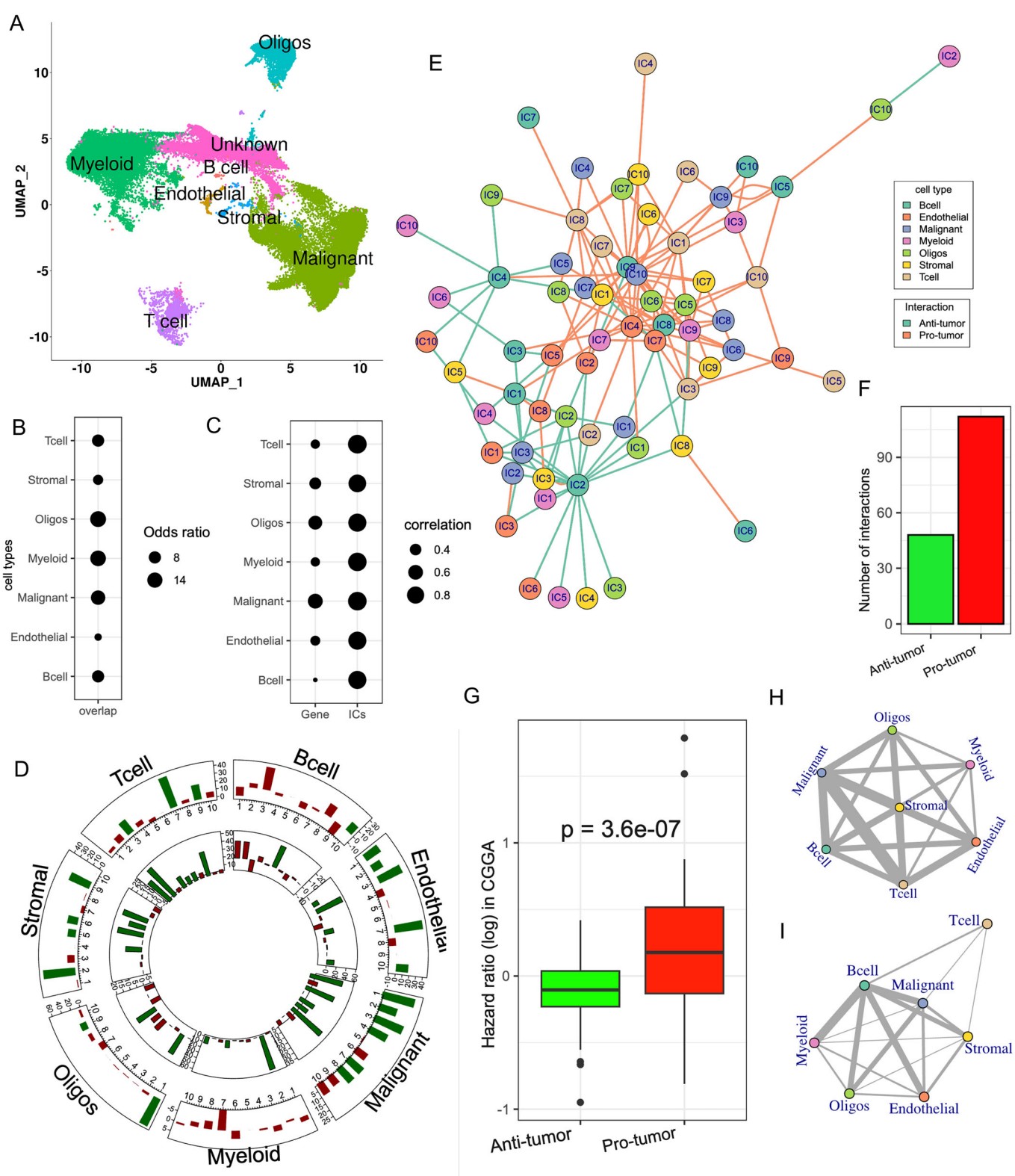

◀

**Figure 2. Transcriptional states and their interactions in IDH mutant gliomas.**

(A) UMAP plot of the scRNA-seq data from IDH-mutant patients. Cells are colored, and the cell type labels are shown for each cluster. (B) A dot plot showing the odds ratio of overlap among the top genes expressed by each cell type between TCGA and CGGA in the deconvolved data; all overlaps were significant (FDR < 0.20, P values are as follows: B cells = 7.95E-11, Endothelial = 3.05E-05, Myeloid = 2.36E-12, Malignant = 5.98E-63, Oligos = 4.11E-13, T cells = 1.86E-05, Stromal = 1.50E-24) based on Fisher's exact test. (C) Dot plot showing the cross-gene correlations (left column) and cross-IC correlations (right column) of log-hazard ratio between TCGA and CGGA for each cell type. (D) Bar plots in circular layout. Each sector corresponds to a cell type, and each bar represents a distinct IC (x-axis) for that cell type. All inter-cell distances (Euclidean distance in PC space) were computed and partitioned into intra-IC (both cells have high IC score) and inter-IC (exactly one cell has high IC score). The height of bars denotes the percent difference between the inter-IC and the intra-IC distances, such that a positive (respectively, negative) value indicates higher (respectively, lower) similarity between intra-IC cell pairs. A bar is shown in green if the inter-IC distances are significantly smaller than intra-IC distances (difference >10% and FDR < 0.20), implying the tendency of that ICs to cluster in PC space. The outer track is for positive signature genes, and the inner track is for negative signature genes of the ICs (high score for negative signature implies inactivation of the IC). (E) CSIN involves distinct cell states and gene expression programs derived from seven distinct cell types. Node colors represent the cell type and edge color represent the prognostic effect of each interaction. (F) Bar plot showing the number of pro-and anti-tumor interactions in the CSIN. (G) Boxplots showing the hazard ratio of TCGA-inferred pro- (N = 48) and anti-tumor (N = 112) interactions in CGGA. P value from Wilcoxon's rank-sum test is shown. The horizontal line in the middle is the median value with lower and upper edges of the box corresponding to the 25th and 75th percentiles and vertical lines corresponding to 1.5 times the interquartile range. (H, I) CSIN summarized at the level of cell types for pro-tumor (H) and anti-tumor (I) interactions. The width of the edges is proportional to the number of interactions between each cell type pair. Source data are available online for this figure.

type in TCGA significantly overlapped with those in the same cell type of CGGA samples (Fig. 2B), far greater than the overlap between top genes in different cell types (Fig. EV2B). We next performed a Cox regression for each gene using the cell-type-specific deconvolved gene expression in TCGA and CGGA (Methods). We found that for the same cell type, the cross-gene correlation of HR between two cohorts ranged from 0.22 to 0.6 (Fig. 2C, left column), far greater than that when comparing different cell types in the two cohorts (Fig. EV2C). Further, the functional enrichment analysis of the top 5% genes expressed by each cell type revealed several cell-type-specific functions (Fig. EV2A, "Methods"). For instance, immune system terms related to myeloid and leukocytes were enriched in the myeloid cells. Functions related to the central nervous system, such as axon ensheathment, neurogenesis, neuronal differentiation, and glial cell fate commitment, were enriched among the malignant cells and OPCs. Together, these results support the robustness and cross-cohort consistency of the deconvolved data.

Given the deconvolved cell type-specific expression profiles, in each cell type, we identified distinct transcriptional states by using independent component analysis (ICA) on the TCGA cohort (Methods). For each cell type, we derived 10 independent components (IC), where each IC represents a distinct transcriptional cell state or gene expression program ("Methods").

To assess cross-cohort robustness, we projected the deconvolved data from the CGGA cohort onto the latent factor space (ICs) derived from the TCGA cohort ("Methods") and performed Cox regression for each IC across all cell types independently in TCGA and CGGA. Interestingly, many of the ICs were significantly prognostic in TCGA and cross-IC correlation of HR for the same cell type between TCGA and CGGA was significatnly higher than those for cross-gene correlations (Figs. 2C and EV2D). These results underscore the value added by the latent factor projection of the deconvolved data in capturing the cell-type-specific signals. To confirm the biological meaningfulness of the ICs, we performed additional analysis of the ICs using the randomized input data. Under the assertion that ICs are mere statistical noise, the same level of concordance should also be observed if we fitted the IC models in randomized gene expression data. Therefore, we randomly permuted the deconvolved TCGA gene expression data and repeated the entire procedure. We computed the hazard ratios of the randomized ICs in TCGA and compared them against the hazard ratios of the projected ICs in the CGGA data. Interestingly, we observed that across 10 iterations, for

each cell type, the median correlation in the randomized data was ~0 in most cases (Fig. EV2E). These results further establish the biological relevance of the ICs.

As mentioned above, each IC may either represent a distinct cell state or a gene expression program shared across cell states. To assess these two possibilities, in a scRNA-seq dataset of IDH-mut glioma ("Methods"), we inferred each ICs activity in each cell and labeled cells that do or do not express IC signature genes ("Methods"). We observed that in several cases, cells expressing an IC's signature genes tend to be significantly clustered in the principal component space (Fig. 2D). UMAP projections of some of the ICs that show significant clustering in principal component space are provided in Fig. EV2F–H. A total of 54 out of 70 ICs showed a significant tendency for clustering across all cell types (Fig. 2D). These 54 ICs might thus represent distinct cell states of the respective cell type, while the remaining ICs may represent transcriptional programs shared across cell states. The notion that ICs might represent cell states is further strengthened by the observation that signature genes of several ICs had a significant linear clustering on the genome (Appendix Fig. S1C,D), consistent with a coordinated epigenetic regulation underlying cell state switches (Madrigal et al, 2023). Through functional enrichment analyses, we further observed that several housekeeping as well as cell-type-specific functions were enriched among the signature genes of ICs (Dataset EV1).

We further assessed the robustness of detected ICs based on bootstrapping 70% gene expression data for each cell type. For each of the 10 bootstraps, we observed an unambiguous one-to-one mapping between the ICs detected from the full data and bootstrapped data, i.e., each of the IC from full data was strongly correlated (PCC > 0.9) with at most one of the IC in the bootstrapped data, whereas the second-best correlation was significantly lower (Appendix Fig. S1E). We then performed a much stronger test, i.e., we detected ICs independently in TCGA and CGGA and performed the correlation analysis of ICs as before. Again, we observed that for the majority of the cases, each IC detected in TCGA unambiguously mapped to an IC independently ascertained in CGGA (Appendix Fig. S1F).

Together, the fact that (i) 70% of the ICs were clustered in PCA space, (ii) several ICs had their signature genes linearly clustered along the genome, (iii) and ICs exhibited one-to-one mapping in bootstrapped TCGA data and independently in CGGA data suggest that ICA of the deconvolved cell types robustly captures distinct

 

cell states, or their coordinated gene expression programs in the TME.

Finally, we identified functional interactions between transcriptional states (represented by the cell-type-specific ICs) across cell types, associated with patient survival in TCGA. Using Cox regression, we performed a computational screen for all pairs of ICs from two distinct cell types whose joint activity levels (Bins 1, 3, 7, 9 in Fig. 1B) were significantly associated with the survival of IDH-mutant glioma patients while controlling for their individual activities and other confounders ("Methods"). For instance, we tested for assertions such as "Joint occurrence of low activity of IC1 in malignant cells and high activity of IC3 of T cells, corresponding to Bin 3 or 7 in Fig. 1B, is associated with worse prognosis". At FDR threshold of 20% and 70% internal cross-validation accuracy ("Methods"), we detected 160 significant interactions, a majority (70%) of which were associated with worse prognosis (Fig. 2E,F; Dataset EV2); we term the interactions associated with worse or better prognosis as pro-tumor or anti-tumor, respectively. Together, these 160 interactions form a CSI network (CSIN). To ascertain if the detected CSIs are reproducible, we deconvolved an additional cohort of 325 IDH-mutant glioma patients from CGGA and scored each sample using the rotation matrix (gene weights) of the ICs derived from TCGA (Methods). We then performed an analogous detection of CSIs in the CGGA cohort. Encouragingly, pro- and anti-tumor interactions ascertained in TCGA tended to have a positive/negative log-hazard ratio in CGGA as well (Fig. 2G). While the concordance in CSI survival association between TCGA and CGGA is imperfect, such a discrepancy is to be expected across independent bulk transcriptome cohorts owing to differences in the technical, genetic, and environmental differences; for instance, even the gene-gene correlations derived from the two cohorts exhibit limited concordance (PCC ~ 0.62). Additionally, as a negative control, we generated sets of interactions by randomizing the clinical data of patients in TCGA ten times ("Methods") and observed that random interaction sets did not significantly overlap across the cohorts (Appendix Fig. S1A). We additionally assessed the extent to which CSI-TME can detect the CSIs, that were detected in TCGA, entirely independently in CGGA cohort, without relying on ICA space learned in TCGA. We observed that 51% of CSIs detected in TCGA could be independently detected in CGGA (Appendix file and Appendix Table S1). We repeated the same procedure for 10 randomized sets of TCGA CSINs and observed a median recovery rate of 7%. Together, these results support the robustness as well as potential biological significance of the detected CSIs.

In the CSIN, each cell type had a variable number of interactions (Appendix Fig. S1B), with malignant cell and B cell states involved in the greatest number of interactions, having both pro- and anti-tumor effects. A majority of the detected interactions involving T-cell states (ICs) associated with worse survival (i.e., pro-tumor) and largely involved interactions with malignant cells (Fig. 2H). This is consistent with the idea of tumor cell-driven T-cell exhaustion as well as emerging pro-tumor roles of certain T-cell states (Reis et al, 2022).

## CSI-TME identifies glioma stem cells and their associated interactions in TME

Tumor cells exist in multiple transcriptional states shared across cancer types (Barkley et al, 2022). We assessed the extent to which the ICs derived from IDH-mut malignant cells capture these conserved cell states. Toward this, we assessed the overlap between the signature genes of the recurring cancer cell states obtained from Barkley et al and the signature genes of the ICs (positive as well as negative) from glioma malignant cells ("Methods"). We observed that negative genes of three of the ICs significantly resembled oligodendrocyte progenitor (IC5), astrocytic (IC6), and cycling cells (IC7) (Fig. 3A). GO functional enrichment analysis further supported the proliferative nature of IC7 (Fig. EV3A).

Gain-of-function mutations in IDH gene induce stemness (Haddock et al, 2022; Zhang et al, 2022b), and IDH-mutant glioma scRNA-seq analysis has revealed the existence of highly proliferative glioma stem cells (GSCs) (Tirosh et al, 2016). Therefore, we assessed whether the proliferative IC7 cell state (Fig. 3A) might represent GSCs. Toward this, we defined two putative IDH-mut glioma stemness signature genes: PSS1, comprising 1143 genes that were significantly positively correlated with *IDH1* expression across TCGA IDH-mut glioma samples ("Methods"), and PSS2, derived from single-cell investigations of IDH-mut gliomas from a previous study (Venteicher et al, 2017). We observed that both PSS1 and PSS2 had significantly negative gene weights in IC7 (Fig. 3B), and significant overlap with negative signature genes of IC7 (Fig. 3C), suggesting that the negative signature genes of IC7 capture ID-mut stem cells. Probing IC7 signature further, we found that *SOX11*, a bona fide neurodevelopmental transcription factor, is included in the IC7 negative signature as well as in PSS2. Therefore, we further probed the IC7 signature in a publicly available dataset of the developing human brain (Cardoso-Moreira et al, 2019) ("Methods"). Interestingly, the IC7 negative signature genes had higher expression during the embryonic phases of brain development and significantly decreased in fetal brain and mature adults (Fig. 3D). In addition, *CD44* which is highly expressed by GSCs and has a hypoxia-specific effect on invasion and proliferation (Inoue et al, 2023) had a negative loading value in IC7 (Fig. 3B). Together, these results establish that the negative direction of IC7 represents a population of highly proliferative GSCs resembling developing neuronal cells. The majority of the remaining ICs were unrelated to brain development, and IC3 showed a much weaker trend (Fig. EV3B) as compared to IC7, hinting at the existence of an additional stemness axis. Notably, negative signature genes of IC3 also included multiple stemness-related genes, including NKX3-1, Esrrb/Nr5a2, and Tfcp2l1 (Festuccia et al, 2021; Mai et al, 2018), as well as factors critical for cancer stemness, including IL-8 and miR-221 (Jin et al, 2018; Roscigno et al, 2015).

Studies have shown that crosstalk between GSCs and the immune system contributes to therapy resistance (Eckerdt and Platanias, 2023). Focusing on the CSIs involving GSCs (IC7 of malignant cells) and various ICs of the immune system (Fig. 3E), we observed that IC7 engaged in pro-tumor CSIs with four immune compartment ICs -- IC9 of B cells and IC1, 7, 8 of T cells spanning all three activity bins (Fig. EV3C). The interaction of T-cell IC1 occurred in Bin 1 and was pro-tumor. We found that negative signature genes in the IC1 of T cells showed a significant enrichment of functions related to cell cycle progression, such as G1/S transition and chromosomal segregation (Fig. EV3D). These genes were particularly enriched for T-cell proliferation (GO: 0042098) (Fig. 3F). Given that bin 1 CSI implies upregulation of negative signature genes of both ICs (Fig. EV3C), it is intriguing that malignant cell stemness, together with T cell proliferative state, is associated with worse survival. Proliferation of T cells in TME

 

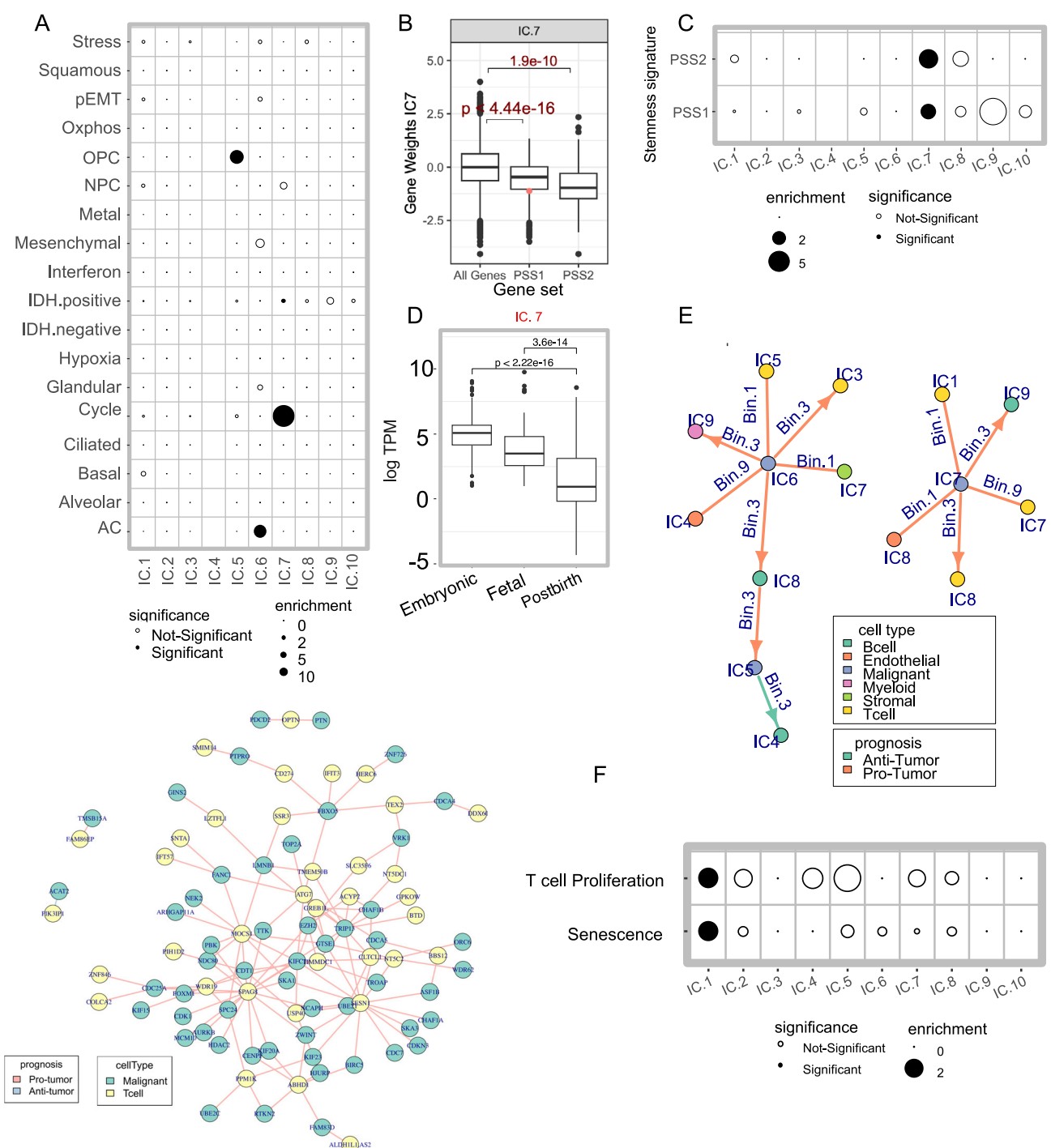

can induce their senescence due to telomere shortening (Woroniecka et al, 2018). However, proliferative GSCs have the ability to maintain telomeres (Woroniecka et al, 2018). We assessed whether the negative signature genes of T cell IC1 also resembled the markers of senescence. We observed that certain lymphocyte-specific markers of senescence, notably, PLAUR (Amor et al, 2020), curated from literature, had negative weights in IC1 (Fig. EV3E). In addition, we obtained a gene signature that is upregulated in the senescent fibroblasts (Wechter et al, 2023). Interestingly, we

observed that negative signature genes of T cells IC1 significantly overlapped with the markers of senescent fibroblasts (Fig. 3F). These results suggest that the synergistic effect of GSCs coupled with T cell senescence might lead to poor prognosis.

Another pro-tumor CSI involved T cell IC7 in Bin 9. Negative signatures of IC7 in T cells significantly overlap the markers of interferon response (Fig. EV3F, "Methods"). Since interaction in bin 9 involves the downregulation of negative signature genes of both interacting ICs, this interaction suggests that a reduction in

**Figure 3.   Identification of glioma stem cells and their interactions with other cell types in TME.**

(A) A dot plot showing the odds ratio for the overlap between the negative signature genes of various Malignant cell IC and previously derived recurrent cancer cell states. The size of dots is proportional to the odds ratio derived from Fisher's exact test, and solid points indicate significance (FDR < 0.20). The missing values are shown for IC4 as it did not have any signature genes (B). A box plot showing the contribution of two stem cell signature genes to IC7 of malignant cells. Sample sizes (N) are—10115 for all genes, 1078 for PSS1, and 65 for PSS2. Red dot is the contribution of the *CD44* gene, which is not included in the PSS. *P* value from Wilcoxon's rank-sum test is shown. (C) A dot plot showing the enrichment of two stem cell signatures among the malignant cell ICs. The shape and size of the points follow a similar convention to that in (A). (D) A box plot showing the distribution of log-normalized TPM values for the negative signature genes of IC7 during different developmental stages of the human brain (N = 111 for each boxplot). *P* value from Wilcoxon's rank-sum test is shown. (B, D) The horizontal line in the middle is the median value with lower and upper edges of the boxes corresponding to the 25th and 75th percentiles, and vertical lines corresponding to 1.5 times the interquartile range. (E) A network plot of CSIs involving IC5, IC6, and IC7 of the malignant cells. (F) A dot plot showing the odds ratio for the overlap between negative signature genes of T cells ICs against the markers for T-cell proliferation (GO:0042098) and senescence (from Wechter et al). (G) A network plot showing the interactions between negative signature genes of IC7 of Malignant cells and IC7 of T cells derived from bulk gene expression data. The nodes represent signature genes corresponding to Malignant cells (green) and T cells (red). Note that this plot is for two specific IC pairs and not the full network of interactions between malignant and T cells. Source data are available online for this figure.

GSCs along with downregulated interferon signaling in T cells is associated with worse survival. To further confirm this CSI, for all pairs of genes among the negative genes of the two ICs, we directly assessed the effect of simultaneous downregulation of the two genes on patient survival (Methods). This analysis resulted in 129 significant pairwise gene interactions with almost all (128/129) having pro-tumor effect upon their simultaneous downregulation (Fig. 3G). Among these, bona fide interferon response genes IFIT1 and IFIT3 in T cells exhibited a significant pro-tumor interaction with PTN and FBXO5 genes in malignant cells, respectively. In particular, the interaction between IFIT1 gene in T cells and the PTN gene in malignant cells, shown to have potential immune-regulatory function (Sorrelle et al, 2017), implies that downregulation of PTN in malignant cells may suppress interferon signaling in T cells and the associated cytotoxic response (von Locquenghien et al, 2021). Together, these results indicate a functional crosstalk and synergistic effects of activity of GSCs with distinct T cell states and gene expression programs associated with survival. We also observed that IC5, which represents the OPC-like lineage, interacted with IC4 and IC8 of B cells. Likewise, IC6, which represents Astrocyte-like lineage, also interacts with immune cells, including IC5 and IC3 of T cells and IC8 of B cells, hinting at rich prognostic crosstalk that exists in the TME beyond glioma stem cells (Fig. 3E).

## IDH-mut astrocytoma and oligodendrogliomas exhibit differential CSI activities

The two major subtypes of IDH-mut glioma--Oligodendrogliomas (IDH-O) that carry co-deletion of chromosomal arms 1p and 19q and Astrocytomas (IDH-A) that do not carry this co-deletion, differ substantially in their prognosis—IDH-O patients having better prognosis, and in their TME composition (Venteicher et al, 2017). Therefore, we investigated whether the detected CSIs had differential distributions and effects in these two tumor types. Toward this, for each CSI, we quantified a metric called interaction 'penetrance' ("Methods") defined as the proportion of patients in which the CSI was active (for instance, the proportion of tumors with simultaneously downregulated states, i.e., in Bin 1). In the TCGA cohort, we observed a significantly higher penetrance ("Methods") in IDH-A tumors compared with the IDH-O tumors (Fig. 4A). We assessed differential activity of CSIs between the two tumor types using Fisher's exact test ("Methods") and observed that CSIs dominant in IDH-A tumors were predominantly anti-tumor (Fig. 4B,C; Fisher's exact test Odds = 2.26, one-tailed *P* value =

0.078). The enrichment of anti-tumor CSIs in IDH-A is intriguing, given that IDH-A tumors generally have a poorer prognosis (van der Vaart et al, 2024). As patients diagnosed with IDH-A type are significantly younger than IDH-O (Carstam et al, 2022; Lee et al, 2023), we assessed whether the observed enrichment is simply due to their relatively younger age as compared to IDH-O. Fitting a linear regression between IDH subtype and anti-tumor load while controlling for age, we still observed a significant enrichment of anti-tumor CSIs in IDH-A (β-coefficient = 0.155, *P* value < 2.2E-16). To further probe this observation, we quantified an additional metric called interaction 'load', defined as the total number of interactions active in each patient (Methods). While *penetrance* is a cohort-level metric, *load* is a patient-level metric. Estimating the interaction load separately of pro- and anti-tumor interactions in each IDH-mut sample, we partitioned the samples into those that had predominantly pro-tumor CSIs active and those that had predominantly anti-tumor CSIs ("Methods"; Fig. 4D,E), and analyzed the survival patterns of these four groups of patients (IDH-A/O and dominant pro/anti-tumor load). We observed that in both IDH-A and IDH-O cohorts, patients dominated by anti-tumor interactions exhibit significantly better survival (Fig. 4F,G). Thus, accounting for the activity of specific types of CSIs in a tumor refines the current understanding of the prognosis of the two subtypes of IDH-mut gliomas. We validated these findings in an additional independent cohort of IDH-mut glioma patients from the CGGA cohort ("Methods"). For this analysis, we considered the interactions that had a similar direction of hazard in TCGA and CGGA (i.e., 103/160 interactions) and scored the samples in CGGA for their joint activity ("Methods"). We observed that, consistent with observations in the TCGA cohort, stratifying the patients with IDH-A tumors based on dominance pro/anti-tumor CSIs revealed a significantly better survival for patients with dominant anti-tumor CSIs (Fig. 4I,J). We note that in Fig. 4G,J, despite the small number of patients in the IDH-O subtype dominated by anti-tumor interactions, we observed a consistent and statistically significant survival difference among these patients in both TCGA and CGGA.

Taken together, these observations point towards the impact of CSIs in the TME on the overall survival of IDH-A and IDH-O patients, which are not necessarily explained by co-deletion of 1p/10q chromosomal arms.

## Ligand–receptor interactions partly mediate CSIs

As mentioned in the introduction, our inferred CSIs may be mediated either via direct intercellular physical interactions or via

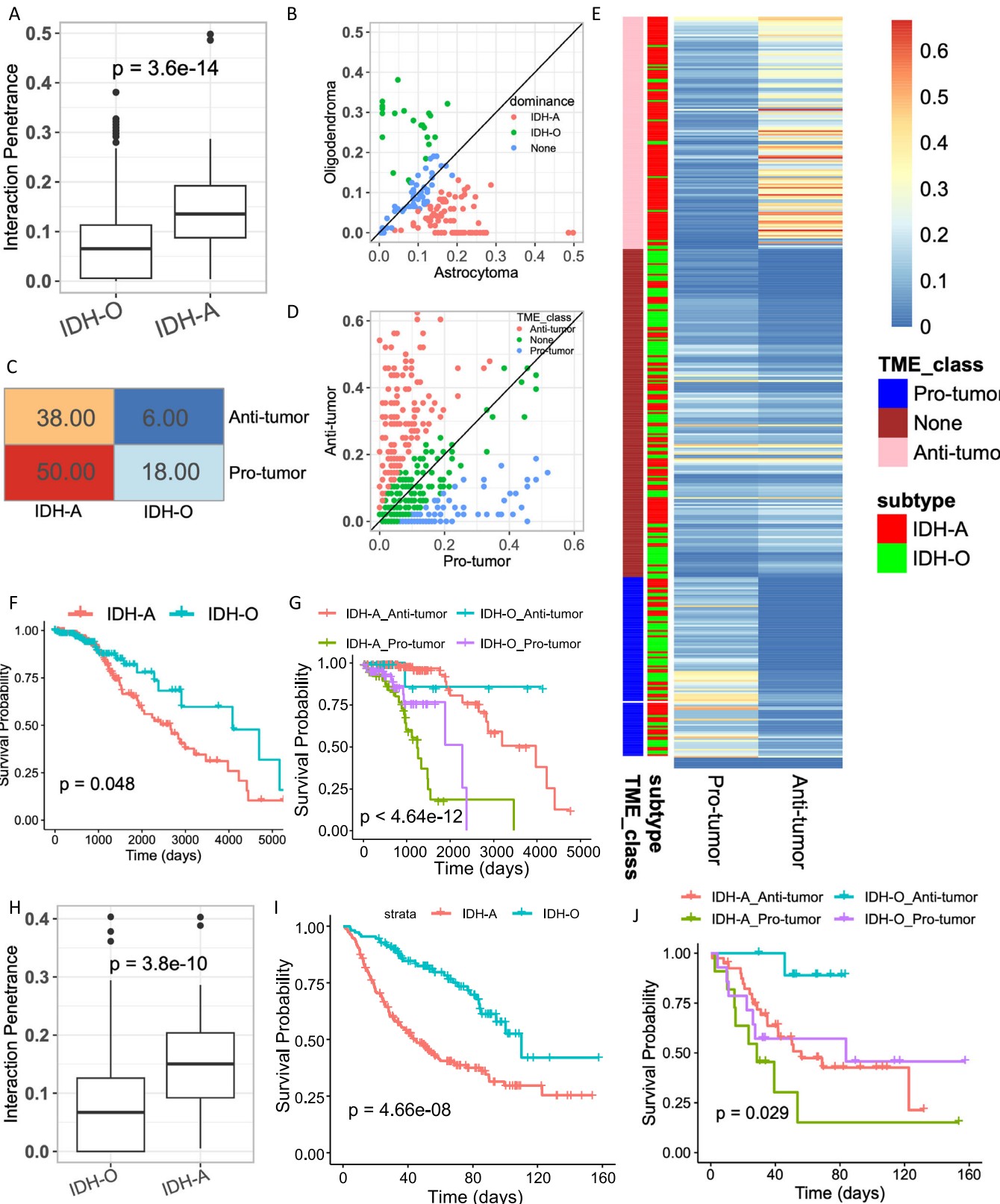

◄ **Figure 4. Analysis of CSIN in IDH-A and IDH-O tumors.**

(**A**) Boxplot showing the CSI *penetrance* among the TCGA IDH-A ($N = 160$) and IDH-O tumors ($N = 160$). *P* value from Wilcoxon's rank-sum test is shown. (**B**) A scatter plot compares CSI penetrance between TCGA IDH-A and IDH-O cohorts. Each dot represents a CSI; those enriched among the IDH-A and IDH-O are colored red and blue, respectively. (**C**) Fisher's exact test contingency table showing the numbers of pro- and anti-tumor CSIs enriched among the IDH-A and IDH-O subtypes (ascertained in **B**) in the TCGA cohort. (**D**) Scatter plot showing the interaction *load* for the pro- and anti-tumor CSIs in the TCGA cohort. The samples where dominant pro- and anti-tumor interactions are colored as blue and red, respectively, representing two TME subtypes. (**E**) Heatmap showing the interaction load of pro- and anti-tumor interactions for the samples with defined TME subtypes in (**D**) in the TCGA cohort. Each row represents a sample and is annotated by the dominant TME subtype and glioma subtype. The colors represent the interaction load, calculated as the fraction of pro- and anti-tumor interactions that were active in that sample. (**F**) Kaplan–Meier survival curves of IDH-A and IDH-O in the TCGA cohort. *P* value from the log-rank test is shown. (**G**) Kaplan–Meier survival curves of IDH-A and IDH-O stratified based on the dominant TME class defined in D in the TCGA cohort. (**H**) Boxplot showing the CSI *penetrance* among the IDH-A ($N = 103$) and IDH-O ($N = 103$) glioma subtypes in the CGGA cohort. *P* values from Wilcoxon's rank-sum test are shown. (**A, H**) The horizontal line in the middle is the median value, with lower and upper edges of the boxes corresponding to the 25th and 75th percentiles, and vertical lines corresponding to 1.5 times the interquartile range. Kaplan–Meier survival curves of IDH-A and IDH-O in the CGGA cohort. (**J**) Kaplan–Meier survival curves showing IDH-A and IDH-O stratified based on the dominant TME class in the CGGA cohort. (**I, J**) *P* values from log-rank test are shown. Source data are available online for this figure.

indirect functional interactions. Here, we assessed the extent to which the detected CSIs are mediated *via* ligand–receptor (LR) interactions (Armingol et al, 2021). Toward this, we first quantified for each CSI, the cognate LR pairs that were expressed by the cell states comprising the CSI ("Methods"). We did not observe a statistically significant difference in the LR support of the inferred CSIs compared to randomized control pairs of cell states (Fig. EV4A). However, encouragingly, the cell types (as opposed to cell states) that interacted more frequently in the CSIN (Fig. 2H,I) were enriched for complementary LR pairs among their ICs (Figs. 5A and EV4B), supporting a broader physical link between interacting cell types in the CSIN.

Next, we further explored the cognate LR pairs among the signature genes of the ICs in the detected CSIN. A total of 20% of the interactions in the CSIN (32/160) involved complementary LR pairs, comprising a total of 69 unique LR pairs among the 7 cell types present among either positive and/or negative signature genes of interacting ICs (Fig. 5B). In 20 of the 69 LR pairs both the ligand and the receptor were activated in the corresponding CSI (Figs. 5C and EV3C) and interestingly, 15 (75%) of the corresponding CSIs anti-tumor. On the other hand, 49/69 LR pairs were inactivated, and 39 (80%) of the corresponding CSIs were pro-tumor (Fig. 5C).

We reasoned that the cell state pairs comprising the 32 LR-supported CSIs should exhibit significant spatial proximity. To investigate this, we obtained six publicly available spatial transcriptomic data (Greenwald et al, 2024) and scored each spot in the Visium slide for the activity of each cell state. Then we identified the cell state pairs that were significantly spatially proximal ("Methods") and tested for the enrichment of CSIs among these. We observed that LR-supported CSIs were significantly enriched among spatially proximal cell state pairs in all six datasets (combined Fisher's exact *P* value < 0.001) while there was no overall spatial proximity among the CSIs (Fig. 5D). This strongly supports evidence for the spatial proximity of the LR mediated interactions in the CSIN.

Closely inspecting the LR-supported and spatially proximal CSIs, we observed that negative signature genes of endothelial IC-3 and positive signature genes of the malignant IC-2 were significantly proximal in all six Visium slides (Fig. 5E). Interestingly, this CSI in Bin 9 involved 25 distinct LR pairs. Comparing the corresponding ICs' signature genes with known endothelial and malignant cell states, we observed that the positive signature genes

of malignant IC-2 mapped to markers of hypoxia, stress, partial EMT, and mesenchymal state (Fig. 5F). Hypoxia induces the stabilization of hypoxia inducible factor (HIF) and activates EMT (Hapke and Haake, 2020). Interestingly, the negative signature genes of the endothelial IC-3 significantly resembled the markers of tip-like endothelial cells (Fig. 5G, odds ratio ~16 and FDR < 3.0e-06) involved in angiogenesis (del Toro et al, 2010). Angiogenesis is typically associated with worse prognosis, and indeed, we confirmed in our data that the endothelial IC-3 had a negative hazard ratio, implying a pro-tumor role of its negative signature genes (Fig. 5H). Thus, while downregulation of tip-like cells is associated with improved prognosis, our results suggest an opposite (pro-tumor) effect, in conjunction with hypoxic malignant cells (Fig. 5I,J), rooted in the loss of the LR-mediated communication between these two cell states.

Another interaction between malignant IC-3 and T cells IC-2 occurred in Bin 1 and was associated with better prognosis. These ICs contained NOTCH2 and JAG2, respectively, among the negative signature genes, implying their simultaneous activation (Fig. 5C). Consistent with the role of NOTCH2-JAG2 in the adhesion of cells (Murata et al, 2014), we observed that negative signature genes of T cells IC-2 significantly resembled the markers of T-cell adhesion (Fig. EV4D, "Methods"). Therefore, this interaction indicates that attachment of T cells to malignant cells via JAG2-NOTCH interaction might be an important factor in T-cell-mediated anti-cancer immunity and consequently associated with the improved patient prognosis (Kelliher and Roderick, 2018).

## CSIs are associated with therapy response and evolve during tumor relapse

Here, we assessed whether the CSIs in the TME contribute to therapeutic response and how they change in relapsed tumors. Toward this, we first tested if the pro- and anti-tumor CSIs are associated with response to immunotherapy. We performed this analysis on a robust subset of interactions in CSIN consisting of 24 CSIs ("Methods", Dataset EV2). We obtained a pre-treatment transcriptomic dataset from 29 patients undergoing neoadjuvant anti-PD1 immune checkpoint blockade therapy ("Methods"). We deconvolved this dataset using CODEFACS and in each sample, projected each cell type onto the ICA space established in the TCGA dataset ("Methods"), enabling us to determine the activity status of each cell type-specific IC in each sample. We calculated

 

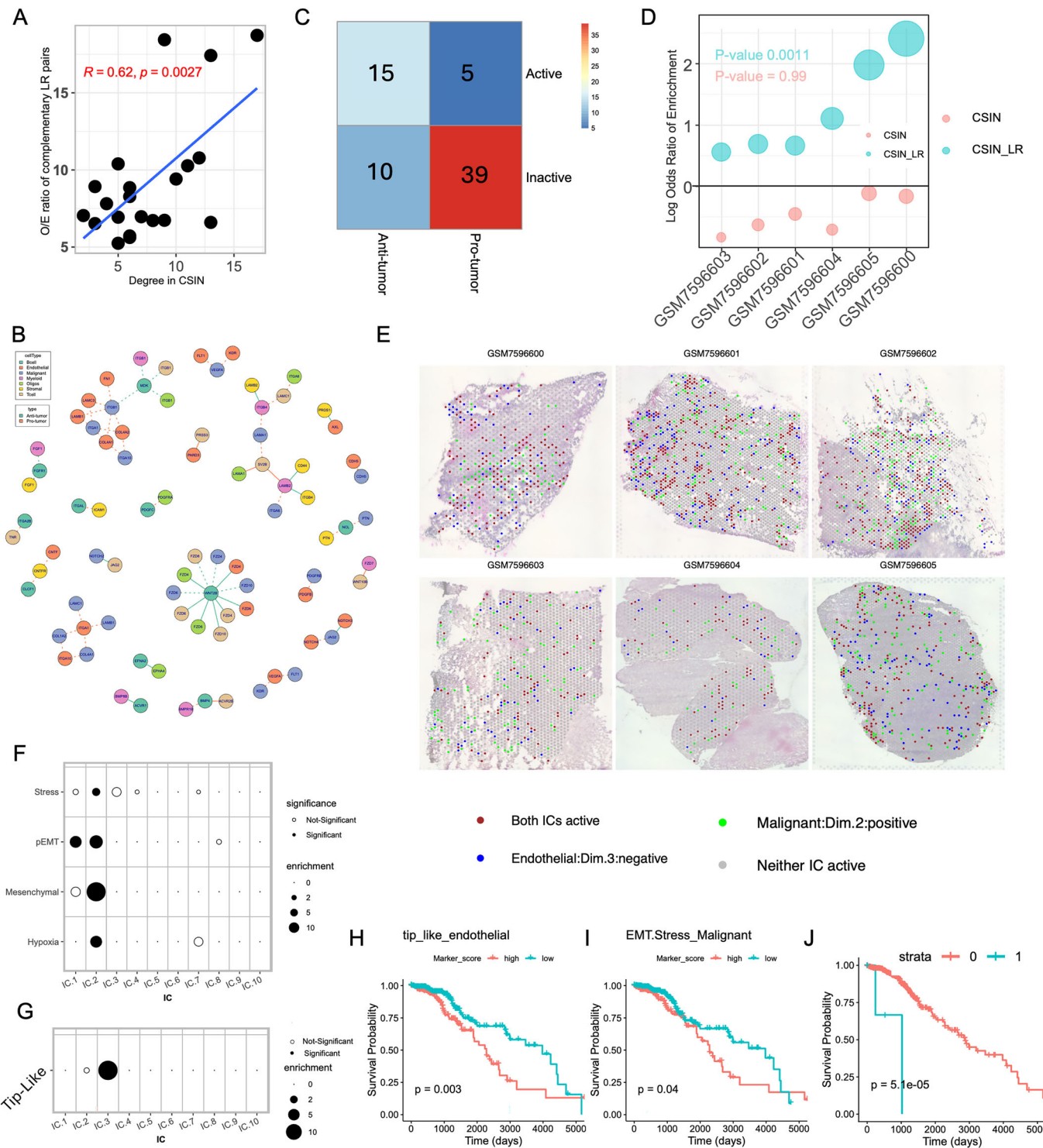

the penetrance of pro- and anti-tumor CSIs in these patients ("Methods") and, encouragingly, we observed that the penetrance of pro-tumor interactions was significantly higher in non-responders than in responders, and the converse was true for anti-tumor CSI, although the latter achieved only marginal significance, possibly due to only four anti-tumor CSIs being used

(Fig. 6A). These results support a role for TME CSIs in successful immune checkpoint blockade therapies.

We next investigated how the CSIs evolve as the tumor relapses after initial shrinkage following the standard of care therapies. For this, we obtained the bulk transcriptomic data from paired primary and recurrent tumor biopsies from a cohort of 25 IDH-mut glioma

 

Figure 5.   Analysis of ligand–receptor mediated cell state interactions.

(A) Scatter plot showing the degree of interactions between cell type pairs calculated from CSIN and observed by the expected ratio of the complementary LR pairs present across all IC pairs between each cell type. Spearman's correlation and corresponding P value is shown. (B) A network showing the known LR pairs that mapped to the CSIs. LR gene names are shown, and color represent the corresponding cell type in the CSIN. Color of edges indicate the pro-tumor (red), or anti-tumor (green) effect of the CSI and the shape of edge indicates the state of ligand–receptor interactions, i.e., activated (solid line) or inactivated (dotted line). (C) A heatmap showing the number of activated and inactivated LR pairs having either pro- or anti-tumor interactions in the CSIN, the color of cells is scaled as per the value displayed in that cell. (D) A dot plot showing the enrichment of LR-mediated CSIs among the spatially proximal IC pairs. For each dataset along the x-axis, enrichment is calculated using Fisher's exact test where the size of dots represents the odds ratio for the enrichment. The CSIs mediated by LR pairs are shown in green, while the overall CSIs is shown in red. For both cases, P values shown are calculated using Fisher combination for the one-sided P values across datasets. (E) Visium slides of IDH-mut spatial transcriptomes showing the co-localization of spots expressing the positive signature genes of Malignant cells IC.2 (green spots) and negative signature genes of Endothelial cells IC. 3 (blue spots). Some spots expressed both the ICs and are shown in red. (F) A dot plot showing the odds ratio for the overlap between the positive signature genes of various Malignant cell ICs and various known cancer cell states. (G) A dot plot showing the odds ratio for the overlap between the negative signature genes of various Endothelial cell ICs and the markers of tip-like endothelial cells. In both (F, G), the size of dots is proportional to the odds ratio derived from Fisher's exact test and solid points indicate the comparisons where the FDR-corrected P value < 0.20. (G–I) Kaplan–Meier's plot showing the survival probability of the IDH-mutant glioma patients stratified based on the activity of tip-like cells (i.e., IC3 of endothelial cells) in (H), Hypoxic malignant cells undergoing EMT (i.e., IC 2 of the malignant cells) in (I), and the interaction between these two in (J). In (G–J), P values from the log-rank test are shown. Source data are available online for this figure.

patients collected by GLASS consortium ("Methods"). As above, after deconvolution using CODEFACS, we projected the cell type-specific expression profiles onto the ICA space established in the TCGA dataset ("Methods"). We hypothesized that following tumor relapse, there should be an increase in pro-tumor CSIs, and a decrease in anti-tumor CSIs relative to the primary tumors. Directly comparing the interaction *loads* of the primary and recurrent biopsies from the same patients, we observed a significantly greater pro-tumor CSI *load* in the recurrent tumors as compared to the corresponding primary tumors; the CSI *load* of anti-tumor interactions did not exhibit statistically significant difference (Fig. EV5A,B). To identify the specific CSIs underlying this trend, we tested for the CSIs whose penetrance among the recurrent tumors was significantly enriched relative to the primary cases ("Methods"). Consistent with interaction load, we observed that penetrance of pro-tumor interactions was also significantly higher in recurrent tumors compared with primary cases (Fig. 6B,C). At a nominal P value cutoff of 0.05, we identified three interactions (Fig. 6C), and all three had pro-tumor survival roles. Two of these interactions involved stromal cells, which were recently proposed to stimulate glioma progression by regulating the tumor microenvironment (Cai et al, 2021). These results suggest that the pro-tumor interactions in the TME might mediate and be selected for in patients with relapsed tumors.

## CSIs exhibit tumor stage-specific associations with somatic mutations

Somatic mutations in cancer cells can affect the emergence of unique cell states within the TME (Ramirez et al, 2024) and shape the composition and functionality of the TME (Mansouri et al, 2022). Therefore, we explored the extent to which our inferred cellular states (ICs) and CSIs are associated with somatic mutations. Towards this, through regression, we identified ICs (among the 70 ICs across cell types) that were significantly associated with the mutational status of any of the 139 frequently mutated (>1% of samples) genes ("Methods"). This yielded 135 significant associations between 36 ICs (predominantly with Endothelial cells and Stromal cells) and 9 genes, including TP53, ATRX, and CIC (Fig. EV6A,B). We next assessed whether the inferred CSIs were associated with the somatic mutations. Towards this, we first plotted the total mutational burden ("Methods")

against the total interaction *load* (as defined above) across the IDH-mut glioma samples. We observed a significant, albeit small, correlation between mutational burden and CSI *load* (Fig. 7A). These results support the potential influence of somatic mutations on the cell state compositions as well as CSI *load* in the TME.

Next, we identified a specific gene-CSI pair where the CSI activity was significantly associated with the gene's mutation status across the patients ("Methods"), yielding 266 significant gene-CSI associations (Fisher's exact test, FDR < 0.2) involving 22 mutated genes and 91 interactions from the CSIs (Fig. 7B). The gene-CSI associations in TCGA were highly reproducible in CGGA ("Methods"; Fig. 7C; Correlation of 0.6). Interestingly, we found that overall, the anti-tumor CSIs are highly enriched among the 91 mutation-associated CSIs (Fisher's exact test P value = 4.2e-09, odds ratio = 12.15), raising the possibility that the TME may elicit a homeostatic response against somatic mutations to restore normal physiology. Presuming that a homeostatic response may be greater in earlier stages (grade I/II) of cancer versus late (grade III/IV), it would suggest that the gene-CSI association may be greater in the earlier stages of cancer. To test this, for each associated gene-CSI pair, we quantified whether the CSIs early:late split was greater among the mutated samples than in wild-type samples ("Methods"). We observed that for anti-tumor interactions, the early:late ratio was significantly higher for mutated samples as compared to the wild-type samples (Fig. 7D). The pro-tumor interactions, on the other hand, showed a reverse trend. These results are consistent with our hypothesis that anti-tumor CSIs may, in part, be an early homeostatic response to the oncogenic somatic mutations, and as the tumor progresses to more advanced stages, the TME is reprogrammed to support tumor progression via pro-tumor CSIs. We further strengthened this finding through an alternative approach. We defined a binary variable called mutation interaction co-occurrence (MIC), which is "1" for a sample where the two co-occur and "0" otherwise ("Methods") and modeled the relationship between tumor grade and MIC while controlling for the effect of mutation and interaction alone ("Methods"). Consistently, we observed MIC was negatively associated with tumor grade for anti-tumor interactions and positively associated with pro-tumor interactions (Fig. 7E), suggesting that early on, anti-tumor interactions are enriched in mutated samples relative to the wild-type samples, potentially representing a homeostatic response.

    

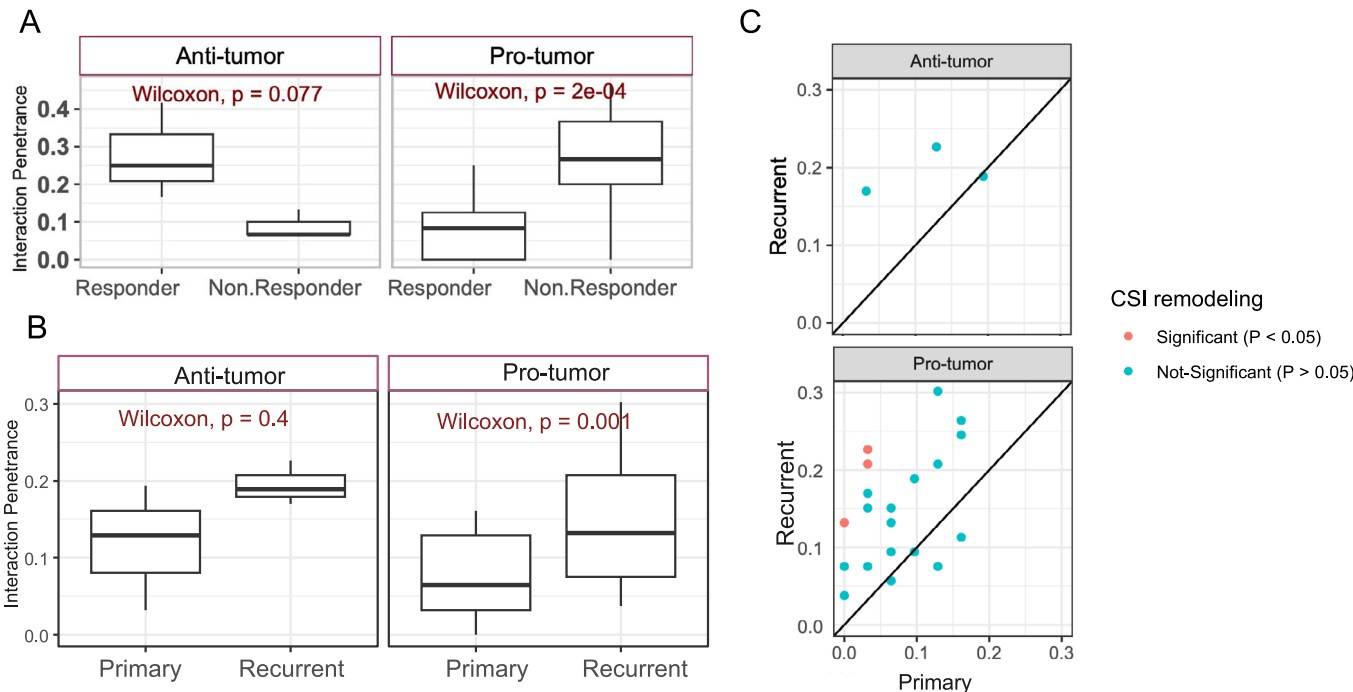

**Figure 6.  Association of CSIN with therapy response.**

(A) Boxplots showing the distributions of penetrance of pro- (N = 21) and anti-tumor (N = 3) CSIs among the responding and non-responding glioma patients under immune checkpoint blockade therapy. (B) Boxplots showing the distributions of penetrance of pro- (N = 21) and anti-tumor (N = 3) CSIs among the primary and recurrent glioma patients. (A, B) The horizontal line in the middle is the median value, with lower and upper edges of the boxes corresponding to the 25th and 75th percentiles, and vertical lines corresponding to 1.5 times the interquartile range. P values from Wilcoxon's rank-sum test are shown (C). A scatter plot with the same data as in (B) where each dot represents the penetrance of each pro- or anti-tumor CSI among the primary (x-axis) and recurrent tumors (y-axis). The CSIs whose penetrance changed significantly (i.e., nominal P value < 0.05, Fisher's exact test) between primary and recurrent patients are shown in red. Source data are available online for this figure.

## CSI-TME is broadly applicable to multiple cancer types in identifying clinically relevant cell state interactions

Having established the utility of CSI-TME in detecting clinically relevant interactions in IDH-mutant glioma, here we aimed to assess the general applicability of CSI-TME in diverse cancer types. Toward this, we applied CSI-TME to TCGA breast cancer (BRCA), head and neck cancer (HNSC), and melanoma (SKCM) as there is an abundance of independent clinical datasets for validation in these cancer types (Methods). Briefly, as in the case of IDH-mutant glioma, for each of the three cohorts, we performed ICA factorization of the CODEFACS deconvolved cell-type-specific gene expression data and screened for the prognostic cell state combinations using Cox regression and retained the significant cell state pairs ("Methods"). Consistent with our observation in IDH-mut glioma, pro-tumor interactions were far more abundant than anti-tumors in all three cases (Fig. 8A). We investigated the cross-cancer consistency of CSIs by projecting the transcriptomic data from one cancer type onto the ICA space of the other cancer type. We observed that a majority of the CSIs exhibited cancer-type-specific prognostic effects (Fig. EV7). We next tested if the detected interactions were associated with the tumor progression or response to therapies in additional independent clinical cohorts.

Toward this, we first utilized the RNA-seq data from two large cohorts of breast pre-malignant lesions from HTAN, monitoring their progression up to a median time of 7.3 years (Strand et al,

2022). Using the loadings from ICA models fitted to TCGA BRCA data, we projected the two cohorts of HTAN data onto cell state space and calculated the penetrance of CSIs in each lesion. Interestingly, we observed that pro-tumor interactions were significantly more penetrant in the pre-cancer lesions that progressed as compared to the ones that did not (Fig. 8B). Applying an identical approach to an additional clinical cohort of BRCA treated with Trastuzumab (Dieci et al, 2016), a Her2 targeting drug ("Methods"), we found that the pro-tumor interactions were more penetrant in patients that showed resistance to Trastuzumab as compared to the patients that showed a favorable response (Fig. 8C).

We performed a similar analysis for melanoma and observed that anti-tumor interactions were significantly more penetrant in patients that responded favorable to anti-PD1 (Riaz et al, 2017) as well as BRAF inhibition therapies (Guarneri et al, 2015). On the other hand, pro-tumor interactions were more penetrant in patients that were resistant (Fig. 8D,E). We further tested CSI-TME on HNSC and observed that the penetrance of pro-tumor interactions was again higher in patients that did not respond to Cetuximab (Bossi et al, 2016) (Fig. 8F); however, the anti-tumor interactions did not show a significant difference between the responders and the non-responders. Collectively, these results suggest that the CSI-TME can be successfully used to mine clinically relevant interactions in diverse cancer types that are not only associated with the progression of pre-malignant lesions, but

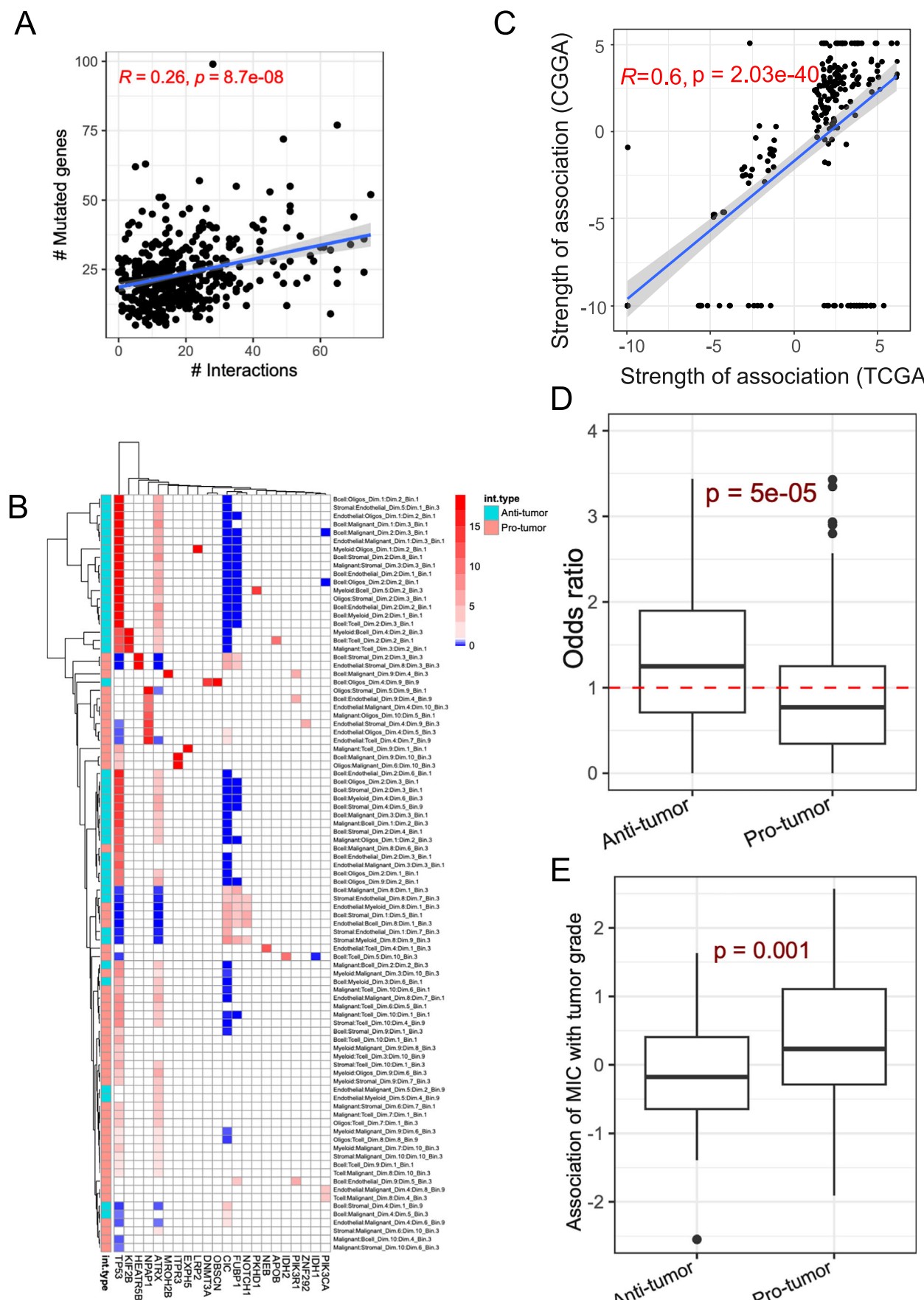

**Figure 7.   Association between somatic mutations and CSIN.**

(**A**) Scatter plot between the interaction *load* (*x*-axis) and mutation burden, calculated as the number of mutated genes (*y*-axis) in the TCGA samples. The Spearman's correlation and *P* value are shown. (**B**) A heatmap showing the strength of significant associations between specific CSI and the mutations in the frequently mutated genes in IDH-mutant gliomas. Cell colors represent the association strength in terms of odds ratio obtained from Fisher's exact test. The non-significant association are shown in white. (**C**) A scatter plot showing the mutation-CSI association (i.e., odds ratio) in TCGA and CGGA cohorts of IDH-mutant glioma patients. The Spearman's correlation and *P* value are shown. (**D**) Boxplot showing the enrichments (odds-ratio) of co-occurrence between somatic mutations and interactions in lower-grade samples relative to higher-grade for pro- ($N = 110$) and anti-tumor ($N = 100$) interactions. The dotted line shows the baseline expectation value under the null hypothesis that the mutation-CSI association is independent of the tumor grade. (**E**) Boxplot showing the estimate of the association between (i) strength of mutation ~ interaction co-occurrence (MIC) and (ii) tumor grade for pro- ($N = 109$) and anti-tumor ($N = 98$) interactions. In both (**D, E**), the Wilcoxon's rank-sum *P* values are shown. In both (**D, E**), the horizontal line in the middle is the median value with lower and upper edges of the boxes corresponding to the 25th and 75th percentiles and vertical lines corresponding to 1.5 times the interquartile range. Source data are available online for this figure.

also with the response to various targeted as well as immunotherapies.

## Discussion

Genes evolve and function in the context of the genetic and regulatory networks (Phillips, 2008), a phenomenon termed epistasis in molecular genetics. The knowledge of such interactions has implications in precision oncology and combination therapy (Szczurek et al, 2013). One special case of gene interactions is synthetic lethality, where the joint, but not individual, inactivity of two genes results in reduced cell viability, exemplified by the discovery and FDA approval of PARP-inhibitors which are particularly effective in BRCA deficient cancers because of the synthetic lethal interactions between these genes (Helleday, 2011). These ideas have fueled the development of several data-driven methods to identify clinically relevant interactions between genes (Jerby-Arnon et al, 2014; Lee et al, 2018; Magen et al, 2019; Matlak and Szczurek, 2017).

In this work, we extend the concept of functional interactions between genes to those between cellular states across the cell types in the TME. A myriad of cell types are present in TME in addition to the malignant cells. While single-cell and spatial transcriptomics datasets from cancer patients provide a robust means to understand the composition, heterogeneity, and intercellular communication (Cheng et al, 2023) in the TME, due to technical difficulty and cost, such data substantially lag behind the rich clinical information available in large cohorts of bulk transcriptomic data such as TCGA. In this work, we developed a novel computational pipeline that leverages the large cohorts of clinical bulk transcriptomics data to understand clinically relevant crosstalk in TME.

Applying our method to bulk transcriptomic and clinical data from TCGA IDH-mutant glioma cohort, we detected interactions involving multiple cellular states in the TME. The resulting cell state interaction network had a greater than two-fold enrichment for pro-tumor interactions relative to the anti-tumor interactions, suggesting that the TME is globally wired to promote tumor progression. The detected interactions were significantly reproducible in another independent cohort (CGGA) of IDH-mutant glioma patients ensuring the technical as well as biological robustness of the CSIN; however, the reproducibility between the two cohorts is limited, owing to substantial genetic and transcriptomic inter-cohort differences. The biologically meaningful nature of our approach is also underscored by the fact that we could recover multiple known states of various cell types present in the

TME through our approach. For instance, through the ICA of deconvolved bulk transcriptomic data, we could identify multiple malignant cell states which have been previously proposed to exist in IDH-mutant gliomas, including GSCs, Astrocytic-like, and OPC-like. Further, the ICs derived from deconvolved cell types had a significantly consistent effect on patients' survival between TCGA and CGGA, ensuring the technical robustness of our approach and enabling us to assess their effect on patient survival.

Single-cell RNA sequencing, combined with known markers of various cell types, has helped characterize cell type compositions in a tissue. However, reliance on a handful of known markers cannot reveal subtle shifts in transcriptional states of various cell types. Latent space embedding of the scRNA-seq data using non-negative matrix factorization (Kotliar et al, 2019), or supervised learning approaches (Kunes et al, 2024) can provide a more resolved view of transcriptional states of various cell types in the tumor micro-environment. Given a resolved cell state composition, intercellular communication can be inferred based on the expression of complementary ligand–receptor pairs by distinct cell types (Armingol et al, 2021); however, our knowledge of functional ligand–receptor pairs is limited. The broader goal, however, is to understand how the cell state composition and intercellular interactions vary across tumors and whether those variations are linked to clinical features, to ultimately be modulated for therapeutic purposes. A previously developed method called Dialog can bypass the need for a database of known ligand–receptor interactions to infer cell-communication by defining a concept of multiple cellular communities based on the canonical correlation analysis of cell-type-specific gene expression programs in scRNA-seq data (Jerby-Arnon and Regev, 2022). However, it only captures the cell states or gene expression programs that co-vary between multiple cell types, unlike CSI-TME, which infers prognostic cell state combinations that do not necessarily co-vary across patients. The main challenge toward the broader goal stated above is insufficient clinical scRNA data. CSI-TME overcomes these limitations by (i) deconvolving a large cohort of clinical bulk transcriptomic data, (ii) identifying cell states and their transcriptional programs, and (iii) explicitly identifying clinically relevant cell state pairs based on their joint activity across cell types in the TME. Previously, a non-negative matrix factorization (NMF) based approach, called EcoTyper, was developed to identify multicellular tumor ecosystems across cancer types in bulk, single cell, or spatially resolved gene expression data (Luca et al, 2021). Tumor ecotypes were defined based on the significant co-occurrence of NMF-based cell states across multiple tumor types. However, in CSI-TME, we infer interactions between cell state pairs by directly

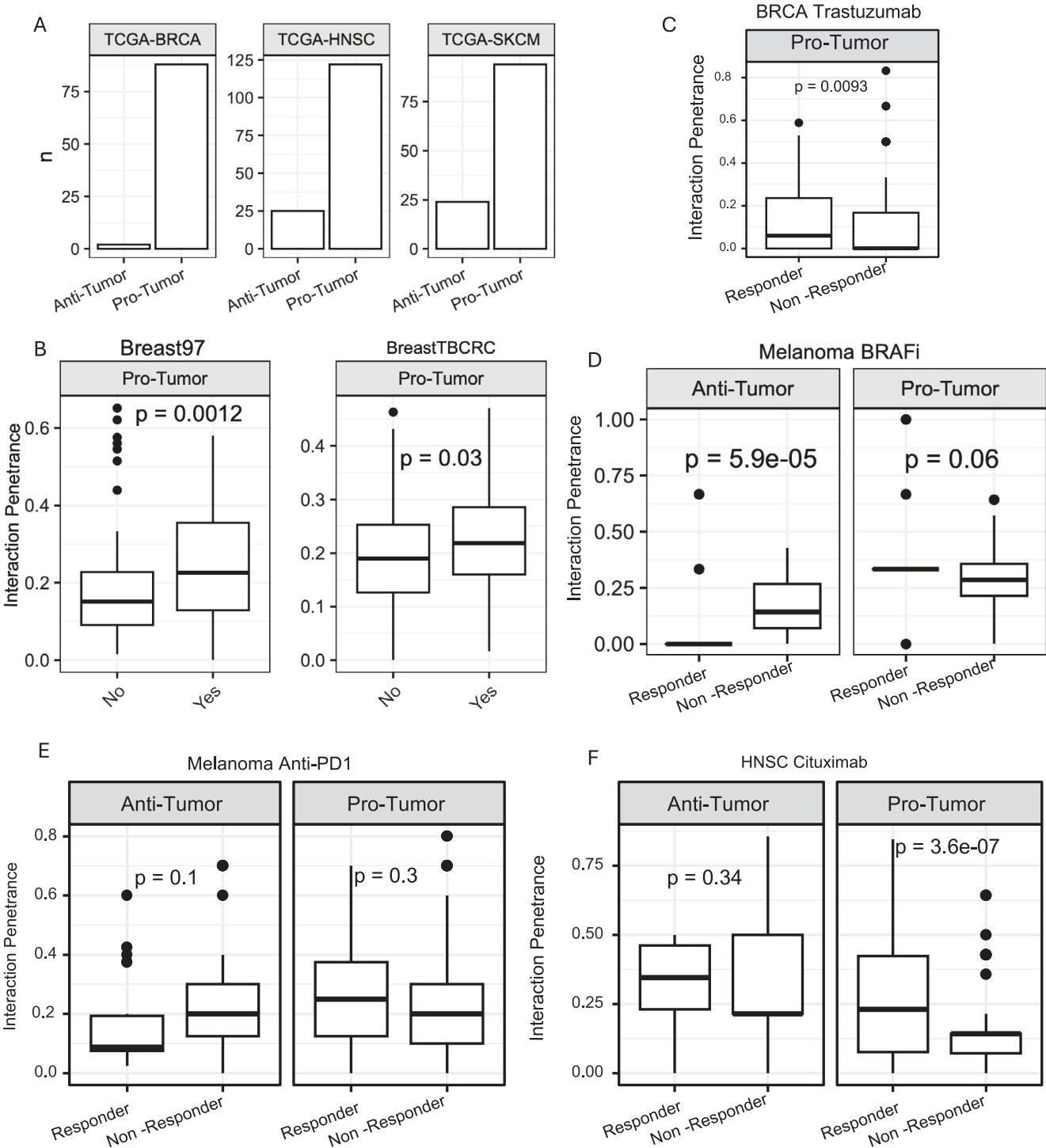

**Figure 8. Application of CSI-TME to additional cancer types.**

(A) Bar plots showing the counts of pro and anti-tumor interactions detected by CSI-TME in TCGA-BRCA, TCGA-SKCM, and TCGA-HNSC. (B, C) Boxplot distribution of the penetrance of pro-tumor interactions, identified in TCGA breast cancers, in the breast pre-malignant lesions from two different HTAN cohorts (B) and patients treated with trastuzumab (C). (B, C) The sample size (N) for all the boxplots is 89. (D, E) Boxplot distribution of the penetrance of pro-tumor (N = 97) and anti-tumor (N = 26) interactions, identified in TCGA-SKCM, in patients treated with BRAF-inhibitors and anti-PD1 therapy. (F) Boxplot distribution of the penetrance of pro-tumor (N = 123) and anti-tumor (N = 25) interactions, identified in TCGA-HNSC, in patients treated with Cetuximab. In all cases, P values from Wilcoxon's rank-sum test are shown. Anti-tumor interactions were plotted if at least 3 were detected. (B–F) The horizontal line in the middle is the median value, with lower and upper edges of the boxes corresponding to the 25th and 75th percentiles, and vertical lines corresponding to 1.5 times the interquartile range. Source data are available online for this figure.

modeling the survival data based on joint activity/inactivity of the pairs of cell state ICs, a key difference that enable us to identify prognostic cell-cell interactions

It is intriguing that despite the lack of immunotherapy response information in TCGA, the prognostic cell state interactions that we identified in TCGA were associated with clinical response to PD-L1 blockade immunotherapy in an independent cohort. Since the cytolytic activity of immune cells in TME contributes to the survival of patients, the cell state interactions associated with worse patient prognosis in TCGA might represent a microenvironment conducive for tumor growth with low cytotoxic activity. Patients with such a microenvironment might not benefit from the PD-L1 blockade which is intended to activate the suppressed T-cells. One could potentially use CSI-TME to prioritize ligand receptor interactions underlying pro-tumor microenvironment and/or resistance to immunotherapy. Further, CSI-TME models the overall survival of patients based on the joint activity of distinct cell state combinations. However, given sufficient cohort size, CSI-TME can be directly used to model other clinical variables of interest such as response to therapies, tumor relapse, or metastasis based on the activity of TME. Intriguingly, we observed that B cell states were involved in the greatest number of interactions with distinct malignant cell states. These included a terminally differentiated and antibody-producing B cell state represented by IC.4 and marked by SDC1 (McCarron et al, 2017). Although the role of B cells in IDH-mut glioma is less appreciated, studies in other cancer types, including IDH-wt glioblastoma has shown the critical role of B cells in mediating both pro- and anti-tumor interactions (Laumont and Nelson, 2023; Lee-Chang et al, 2021). While the rarity of B cells in tumor microenvironment makes it challenging, a better characterization of B cells through targeted profiling would be imperative to better understand the anti-tumor role of B cells in IDH-mut gliomas.

Lastly, we observed that the anti-tumor interactions in the TME were differentially enriched among the samples bearing oncogenic mutations in early stages relative to the non-mutant samples. On the other hand, in later stages of tumorigenesis, this balance shifts towards the pro-tumor interactions. These results indicate the potential protective role of the TME during the initial stages of tumor, which gradually acquires the tumor-promoting characteristics.

We note that transcriptional deconvolution of bulk gene expression data is a rapidly evolving field (Im and Kim, 2023), and in addition to the core model assumptions, the accuracy depends on several factors such as sequencing depth, cell type abundance, and inter-tumor heterogeneity (de Vries et al, 2020). Further, the deconvolution accuracy might be improved by incorporating additional data modalities such as DNA methylation (Im and Kim, 2023). In this work, we used a recently published transcriptional deconvolution method called CODEFACS, which uses several heuristics to maximize the number of genes that are deconvolved confidently with an option to exclude low-confidence genes (Wang et al, 2022). The observed consistency between two different cohorts (i.e., TCGA and CGGA) based on the ICA of the deconvolved cell types (Fig. 2B,C) and recovery of known cell states (Fig. 3A) supports the accuracy and robustness of the approach.

Taken together, leveraging the large cohorts of bulk transcriptomic and clinical datasets, our work provides a data-driven computational pipeline to discover a significantly reproducible set of prognostic cell state combinations, prioritize clinically targetable ligand–receptor interactions, and stratify patients for immunotherapy.

# Methods

**Reagents and tools table**

| Reagent/resource | Reference or source | Identifier or catalog number |
|---|---|---|
| **Experimental models** | N/A | |
| **Recombinant DNA** | N/A | |
| **Antibodies** | N/A | |
| **Oligonucleotides and other sequence-based reagents** | N/A | |
| **Chemicals, enzymes, and other reagents** | N/A | |
| **Software** | | |
| CODEFACS | https://pubmed.ncbi.nlm.nih.gov/34983745/ | |
| R | https://www.r-project.org | |
| CSI-TME | This study | |
| **Other** | N/A | |

## Bulk RNA seq datasets

We used publicly available bulk RNA-seq datasets from IDH-mutant glioma patients from The Cancer Genome Atlas (TCGA) and Chinese Glioma Genome Atlas (CGGA) (Zhao et al, 2021). Read count matrix for 425 IDH-mutant glioma patients from TCGA was downloaded from UCSC Xena browser (Goldman et al, 2020) and was normalized to TPM (Transcripts Per Million) using a shiny R-based COEX-seq package (https://github.com/kimsc77/COEX-seq). To ensure the robustness and reproducibility of our findings, an additional bulk RNA-seq dataset (ID CGGA693) comprising 325 IDH-mut samples was downloaded from the Chinese Glioma Genome Atlas (CGGA) in TPM units (http://www.cgga.org.cn/) (Kim et al, 2018). Raw RNA-seq data for developing the human brain were obtained from (Cardoso-Moreira et al, 2019) and processed to generate gene-level TPM as part of a previous publication (Singh et al, 2022). RNA-seq data for longitudinal biopsies from primary and recurrent IDH-mutant gliomas was sourced from GLASS (Glioma Longitudinal Analysis) consortium (http://www.synapse.org/glass) and consisted of 85 RNA-seq profiles comprising primary and recurrent tumors from 31 patients-seq data. RNA-seq data from pre-treatment biopsies from 29 pembrolizumab treated glioma patients was sourced from a previous publication (Cloughesy et al, 2019).

## Clinical for TCGA and CGGA

Clinical data, including the overall survival, age, sex, tumor grade, and IDH-A/IDH-O classification of IDH-mut glioma patients in TCGA, were downloaded from PanCanAtlas (Nawy, 2018). Similar clinical data for IDH-mut patients in the CGGA cohort were downloaded from their online portal.

 

## Single-cell RNA-seq dataset

scRNA-seq dataset consisting of 60751 cells from 4 IDH-mutant tumors was obtained from a previous publication (Abdelfattah et al, 2022) (GSE1821091) and 9 IDH-mutant tumors from in-house samples at NCI. Read demultiplexing and alignment to the GRCh38 human reference genome was performed using the Cell Ranger Single Cell Software v2.0 (10X Genomics) with its mkfastq and count functions, respectively. Downstream analysis was performed on the filtered counts with cells having at least 200 genes and the percentage of mitochondrial unique molecular identifiers (UMI) counts less than 20%. Filtered cells were then merged, batch-corrected with Harmony (Korsunsky et al, 2019) and clustered using Seurat v3 package using 5000 most variable genes (based on the "vst" selection method) at 0.1 resolution. Malignant cells were identified using CONICsmat and CopyKat V0.1.0 with default parameters (Gao et al, 2021; Müller et al, 2018) while the remaining nonmalignant cells were annotated by using cell type-specific markers using provided in Dataset EV3. Finally, all cells were classified into seven major cell types, viz., Malignant, Myeloid, B cells, T cells, Endothelial cells, Oligodendrocytes, and Stromal cells. Several cells remained uncharacterized. Besides technical reasons multiple potential reasons may contribute, including the lack of comprehensive reference atlas, incomplete knowledge of marker genes, and the presence of transitional or rare cell states that are difficult to classify (Kiselev et al, 2019; Luecken and Theis, 2019). To ensure robustness, we decided to use only the cell types that could be annotated unambiguously and discarded the ambiguous or uncharacterized cells.

## Pipeline for CSI-TME

To identify cell state interactions, CSI-TME employs the following three main steps.

### Deconvolution

In the first step, CSI-TME employs a recently published transcriptional deconvolution tool CODEFACS (COnfident DEconvolution For All Cell Subsets) to deconvolve bulk RNA-seq datasets. For a given set of cell types along with their transcriptional signatures derived from single-cell RNA-seq data, CODEFACS estimates cell-type-specific expression profiles in bulk transcriptomic data. We first supplied the annotated scRNA-seq data to CIBERSORT (Chen et al, 2018) with default parameters to generate the signatures for seven cell types—B cells, T cells, Malignant cells, Myeloid cells, Oligodendrocytes, Stromal cells, Endothelial cells (Dataset EV3). Using CIBERSORT's Impute Cell Fraction module with its S-mode batch-correction, these signatures were then used to estimate cell type fractions in each bulk transcriptomic sample in large glioma datasets from TCGA and CGGA. Finally, CODEFACS was applied to derive their cell type-specific gene expression profiles in each TCGA and CGGA sample. For downstream analysis, we only considered genes that had a CODEFACS estimated confidence score >0.95.

### Cell state identification using ICA

Independent Component Analysis (ICA) is a computational technique to separate a multivariate signal into additive, statistically independent components. Recently, ICA has been increasingly used to model the transcriptomic datasets to uncover hidden patterns in complex cancer omics datasets (Sompairac et al, 2019). CSI-TME uses ICA to identify distinct cell states or gene expression programs, independently for each of the seven cell types, based on their deconvolved gene expression profile across tumors. We performed ICA using the JADE algorithm implemented in the MineICA package (Biton, 2024) in R. A key parameter in ICA is the number of components (k). Two methods—Maximally Stable Transcriptome Dimension (MSTD) (Kairov et al, 2017) and optICA (McConn et al, 2021)—have been proposed previously to determine the optimal k. However, our application of these methods produced discordant results. For instance, for stromal cells, the MSTD-based approach suggested 48 components, whereas optICA method resulted in just 8. To determine optimal $k$ for the current study, we implemented a simple strategy. Iterating from $k = 1$ to $k = 20$, for each IC decomposition, we reconstructed the original matrix from the factorized latent space and computed the Frobenius Norm between the original and reconstructed gene expression matrix. We performed a segmental curve fitting procedure to find an inflection point beyond which the IC factorization did not result in substantial improvement in reconstruction error. As shown in Fig. EV1, we observed five or six components were sufficient for most of the cell types with no substantial decrease in reconstruction error beyond this. We choose to extract 10 independent components for each cell type, reasoning that it should be sufficient to capture the transcriptomic and functional heterogeneity in a cell type. Prior to ICA, the input gene expression profiles of cell types were log-transformed and z-scored across the samples. From the output of MineICA, source matrix (sample by components) was used as the estimate of the sample-specific score of latent cell states or their gene expression programs.

### Screening for prognostic cell state combinations

In the last step, CSI-TME screens for pairs of ICs across different cell types whose joint activity or inactivity is significantly associated with the patient's survival. For this purpose, we first partitioned the sample-specific scores of each IC into three equal parts defining three activity bins, viz., low, medium, and high. An IC is considered "active" in the "high" portion, and "inactive" in "low" part; "medium" part was excluded. Next, for each pair of ICs (say, IC1 and IC2) taken from two different cell types, we assigned each sample to one of the three joint activity combinations (Low-Low, Low-High or High-Low, and High-High); Low-High and High-Low are treated equivalently without loss of generality. Lastly, iterating through each pair of ICs in each of the three joint activity bins, we modeled the overall survival of patients using a Cox-proportional hazard model based on joint activity of ICs while controlling for the effect of each individual ICs as described below:

$$Hazard \sim IC1 + IC2 + I + age + sex$$

Where *IC1* and *IC2* are the sample-specific scores of the two *ICs*, and I code for the interactions, i.e., the joint activity bin of the *IC1* and *IC2*. Age and sex were used as covariates to control for additional demographic confounders. From this model, the hazard ratio and *P* values for the interaction term *(I)* were considered, and multiple testing correction based on Benjamini–Hochberg's procedure was

performed for each IC pair. The output of CSI-TME consisted of list of tuples (C1, IC1, C2, IC2, B, D) at an FDR threshold of 0.20, where IC1 and IC2 are the two ICs from cell type C1 and C2 respectively, and B represents their joint activity state and D represent the direction of prognostic association between hazard and interaction term, i.e., whether the joint activity state associated with better or worse survival.

Finally, CSI-TME performs internal cross-validation of the detected interactions by creating 10 bootstrapped samples of the clinical data and repeating step 3 to detect significant interactions. CSI-TME only retains the interactions that had a consistent effect on patient survival with $P$ value < 0.05 in at least 70% of the trials.

## Projection of external datasets onto the ICA space of TCGA dataset

The CSIs detected in the discovery cohort of TCGA patients were validated and assessed in different contexts: (i) CSIs were validated in an independent CGGA bulk RNA-seq dataset; (ii) CSIs were evaluated in response to immunotherapy in Cloughsey et al dataset (Cloughesy et al, 2019); and (iii) CSIs were evaluated in relapsed tumors from the GLASS consortium (Barthel et al, 2019). In each of these three cases, to quantify the ICs discovered in the TCGA cohort, we first deconvolved the bulk RNA-seq dataset using the CODEFACS and signature genes of 7 cell types as before. Next, we projected the RNA-seq profiles of each purified cell type onto the latent space of independent components established in the TCGA dataset by performing the matrix multiplication of the loading matrix (i.e., mixing matrix) with the new input data to be projected (i.e., deconvolved gene expression profiles of the corresponding cell type).

## Statistics and data visualization

All statistical and computational analyses were carried out in the R programming language. The $P$ values for testing the significance of the difference between two distributions were estimated using Wilcoxon's rank-sum test. Significance of overlap between different gene signature sets was computed using Fisher's exact test. In case of multiple comparisons, $P$ values were adjusted for FDR using Benjamini–Hochberg's method. We used a uniform FDR threshold <0.20 for reporting significant associations in this study. We used the ggplot2 library in R to create most of the graphics in this work. The network layouts of the cell state interactions which were created using the combination of **network** and **igraph** package in R. Heatmaps were created using **pheatmap** library in R. Gene ontology analysis of the positive and negative signature genes of the ICs was performed using **clusterProfiler** package in R and significantly enriched terms (FDR < 0.20) were plotted as heatmaps.

## Signature genes of ICs

In ICA, the original data **X** with n features (genes) and m samples is modeled as $\mathbf{X} = \mathbf{AS}$. In this setup for $k$ number of extracted components, **A** is the loading or mixing matrix consisting of $n$ genes and $k$ components. For each independent component, genes which were at least 2.5 standard deviations away from the mean column mean in **A** were taken as signature genes of the independent components. Since a priori we do not know which direction of the component is associated with a specific biological

process, we extracted positive signatures (i.e., 2.5 standard deviation above the mean) and negative signature (i.e., 2.5 standard deviation below the mean) for each component separately.

## Scoring single cells using signature genes of ICs and assessment of clustering in PCA space

We used the signature genes of distinct ICs derived from each cell type and scored each individual cell of the corresponding cell type in single-cell data using the *AddModuleSocre* function imported from the Seurat package in R (Stuart et al, 2019). Positively scoring cells indicates that a given signature gene set is enriched in those cells. To assess if the cells expressing the signature genes represent distinct cell states, we computed the Euclidean distances in PC space of scRNA-seq data among all pairs of positively scoring cells (i.e., within cluster) and compared it against the Euclidean distances between all pairs of enriched and non-enriched cells (i.e., between cluster). A significantly smaller distance (FDR < 0.20 and at least 10% shorter) among the positively scoring was taken to indicate clustering of IC signature genes in PC space.

## Marker genes of various cell states

Markers for recurrent cancer cell states were obtained from Berkley et al (Barkley et al, 2022). Markers for T cell states were obtained from Chu et al (Chu et al, 2023). Markers for Endothelial cell states were obtained from Zhang et al (Zhang et al, 2022a). Markers for the lymphocytic senescence were obtained from Martyshkina et al (Martyshkina et al, 2023). Markers for fibroblast senescence were obtained from Wechter et al (Wechter et al, 2023). Due to the lack of systematic data on T-cell senescence, we used fibroblast senescence model as a proxy for this analysis, reasoning that core senescence machinery involving p53-mediated DNA damage response, NF-κB–centered inflammatory programs, overlapping SASP cytokines, with broad chromatin remodeling and mitochondrial dysfunction via cGAS–STING pathway, should be conserved in all cell types. We used two different glioma-specific putative stemness signatures to assess the stemness features among the malignant cell ICs. PSS1 was defined based on the genes whose log-normalized gene expression (in TPM units) was significantly correlated (FDR < 0.2 and PCC > 0.5) with the log-normalized gene expression of the *IDH1* gene in the bulk transcriptomic data from TCGA. PSS1 consisted of 1140 genes in total. PSS2 was sourced from Venteicher et al (Venteicher et al, 2017).

## Enrichment of known cell state markers among the ICs

To assess the extent and significance of overlap between the signature genes of ICs and known transcriptional or cell state markers, we computed the odds ratio and $P$ value using Fisher's exact test. Whenever multiple comparisons were involved, we adjusted the resulting $P$ values with the Benjamini–Hochberg method.

## Calculation of the interaction penetrance and interaction load

For each interaction in CSIN, we assigned a binary status to each patient, indicating whether the interaction is active (assigned as 1)

 

or inactive (assigned as 0). For instance, an interaction in bin.1 would imply that, for patients where two ICs are simultaneously downregulated, the interaction is marked as active (assigned a value of 1); otherwise, it is assigned a 0. This process results in a binary matrix, where the rows represent interactions, and the columns represent patients.

The penetrance of each interaction was calculated as the sum of the values in each row of this binarized matrix divided by the total number of columns (i.e., patients), representing the fraction of patients in which the interaction is active (i.e., has a status of 1). On the other hand, the interaction load measures how many interactions are active in each sample and is defined as the proportion of interactions that are active in each patient sample.

## Somatic mutation data

For somatic mutation data for TCGA IDH-mutant glioma patients, we downloaded the .maf files derived from whole exome sequencing from GDC using the TCGAbiolinks library in R (Colaprico et al, 2016). The resulting .maf files were converted to a gene-by-sample matrix with each entry denoting the number of non-synonymous somatic mutations. The somatic mutation data for CGGA IDH-mutant glioma patients were downloaded as a gene-by-sample matrix format from the CGGA consortium website (http://www.cgga.org.cn/).

## Association of somatic mutations with ICs and CSIN

We first identified the genes that were recurrently mutated (at least 1 non-synonymous mutation in >1% of the samples) in TCGA IDH-mut glioma patients, resulting in a total of 139 genes. Next, we divided each IC derived from the 7 cell types into three binned activity states (coded as 0, 1, 2 from low to high) using quantile binning. To investigate the association between somatic mutations and ICs, we iterated through all possible gene-IC pairs (i.e., 139 genes × 70 ICs) and used a binomial regression to model the binned activity level of ICs based on the somatic mutation status of each gene. We extracted the resulting regression coefficients and $P$ values from the model, adjusted for multiple comparisons using Benjamini–Hochberg's method, and retained the significant associations with FDR < 0.20. To investigate the association between somatic mutations and cell state interactions, we screened through all possible gene-interaction pairs (139 genes * 160 interactions) and assessed the overlap between patients containing somatic mutations in each gene and patients where the interaction was active using Fisher's exact test. The resulting odds and $P$ values were extracted and adjusted for multiple comparisons using Benjamini–Hochberg's method. Significant associations with FDR < 0.20 were retained.

## Test for differential association between mutation and interactions relative to tumor grade

Here we assess whether there is a greater enrichment of anti-tumor interactions among the mutated samples in the early stages of tumor as compared to the later stages when the tumor has already progressed. To investigate this, for each somatic mutation having a significant association with one or more cell state interactions (Fig. 7B), we first partitioned the samples into a mutant and wild-type relative to the specific somatic mutation, resulting in 22 such partitions corresponding to the mutational status of 22 genes. Next, for each of these 22 genes, we collected its significantly associated interactions, and for each interaction, and counted the number of samples where interactions were active in early stages (i.e., WHO grade II) and later stages of tumor (i.e., WHO grade III/IV). For each mutated gene and each associated interaction, we counted the numbers of early stage mutant samples ($c1$), late stage mutant samples ($c2$), early stage wild-type samples ($c3$) and late stage wild-type samples ($c4$) where $c1$, $c2$, $c3$, $c4$ are the counts of samples where an interaction is active (based on the joint activity status of the two ICs).

For each mutant and wild-type row, we calculated the relative proportion of interactions in early vs. late stages (i.e., $r1 = c1/c2$ for mutant samples and $r2 = c3/c4$ for wild-type samples). The odds of differential enrichment in early stages relative to late stages were then obtained by normalizing $r1$ against $r2$ as $r1/r2$. Finally, we plotted the odds for pro- and anti-tumor interactions as boxplots.

As an alternative, we directly assessed the association between mutation interaction co-occurrence (MIC) and tumor grade using logistic regression. For each somatic mutation and interaction (Fig. 7B), MIC was defined as follows,

$$MIC(m, i) = \begin{cases} 1, & m = 1 \ and \ i = 1 \\ 0, & otherwise \end{cases}$$

where $m$ denotes the somatic mutation status of a given gene and $i$ denotes the activity state of a given interaction across samples. MIC score of 1 implies that a given sample is both mutated as well as has the active interaction. The strength of association between MIC and tumor grade was then estimated by regressing tumor grade against the MIC, while controlling for the individual effect of $m, i$ on tumor grade through a logistic regression model.

## Analysis of ligand–receptor interactions

Ligand–receptor pairs were sourced from the cellchatdb database (Jin et al, 2021) consisting of 1986 interactions between 547 ligands and 466 receptors. For each cell type pair, we counted the number of complementary ligand receptor pairs among the signature genes across all combinations of their ICs. We accounted for the difference in the size of signature gene sets by dividing the observed count with the expected number of ligand–receptor pairs. For two ICs having with $n$ and $m$ number of signature genes, and $c$ number of complementary ligand–receptor pairs, we normalized the count $c$ as to derive observed by expected (O/E) ratio as follows:

$$F(LR) = 1986/(547 * 466)$$

$$E = m * n * F$$

$$O/E = c/E$$

Where $F\ (LR)$ is the global density of ligand–receptor connections among a total of 547 ligands and 466 receptors in cellchatdb

database. *E* is the expected number of connections between two sets of *m* and *n* genes given a global density value of *F*. Finally, *O/E* denotes the observed count of *c* connections between *m* and *n* genes normalized by the expected value. For each cell type pair, we calculated the Spearman's correlation between O/E ratio against the number of interactions observed between each cell type pair in CSIN. We performed these steps using all combinations of positive and negative signature genes across all cell-type-specific ICs (i.e., positive genes from IC1 and negative genes from IC2 and so on).

### Detecting co-localized cell state interactions in spatial transcriptomics data

We obtained six IDH-mutant glioma Visium spatial read count datasets from GEO (accession GSE237183) and processed it using Seurat v5.2.1 (Hao et al, 2021). Read counts were normalized according to the procedure described in the source publication (Greenwald et al, 2024). We used Seurat's AddModuleScore function to score the activity of cell states in each spot for those states whose signatures contained at least ten genes. To determine if a cell state was active in a spot, we computed a threshold activity that was based on the 90th percentile of the activity scores of B cells and T cell state signatures. The choice of B and T cells was based on the observation that CODEX analyses of these samples showed negligible T and B cells (Greenwald et al, 2024). The thresholds were computed in a sample-specific fashion.

To determine whether two cell states X and Y were spatially co-localized in a slide, we computed a neighborhood overlap statistic that was defined as:

$$O(X, Y) = \frac{|\bigcup\{Nbds\ where\ X\ and\ Y\ are\ both\ active\}|}{|\{Nbds\ where\ X\ is\ active\}\bigcup\{Nbds\ where\ Y\ is\ active\}|}$$

where each spot's neighborhood (abbreviated as Nbds) consisted of its six nearest neighbors (except for spots on the tissue boundary) and the threshold to determine if X and Y were active was determined as above. We computed the significance of the observed co-localization by computing O(X, Y) after shuffling the assignments of cell state activity scores across each slide a hundred times and fitting a normal distribution to the obtained overlap scores. The one-sided *P* value was calculated as the probability that the observed O(X, Y) was greater than the mean background O(X, Y) estimated via shuffling. The co-localization of all possible cell state interactions was computed separately in each slide, and their *P* values were adjusted for multiple testing using the Benjamini–Hochberg method.

## Data availability

All datasets used in this study are previously published and freely available. The source code for CSI-TME is provided and freely available at GitHub - https://github.com/hannenhalli-lab/CSI-TME.

The source data of this paper are collected in the following database record: biostudies:S-SCDT-10_1038-S44320-026-00201-0.

## Peer review information

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

## Acknowledgements

This work is supported by the Intramural Research Program of the National Cancer Institute, Center for Cancer Research, NIH, and utilized the computational resources of the NIH HPC Biowulf cluster.

## Author contributions

**Arashdeep Singh**: Conceptualization; Resources; Data curation; Software; Formal analysis; Validation; Investigation; Visualization; Methodology; Writing—original draft; Writing—review and editing. **Bharati Mehani**: Data curation; Software; Formal analysis; Visualization; Writing—review and editing. **Vishaka Gopalan**: Data curation; Software; Formal analysis; Visualization; Methodology; Writing—review and editing. **Sushant Puri**: Resources; Data curation; Writing—review and editing. **Kenneth Aldape**: Resources; Data curation, Writing—review and editing. **Sridhar Hannenhalli**: Conceptualization; Resources; Data curation; Supervision; Funding acquisition; Validation; Investigation; Methodology; Writing—original draft; Project administration; Writing—review and editing.

Source data underlying figure panels in this paper may have individual authorship assigned. Where available, figure panel/source data authorship is listed in the following database record: biostudies:S-SCDT-10_1038-S44320-026-00201-0.

## Funding

## Disclosure and competing interests statement

The authors declare no competing interests.

# Expanded View Figures

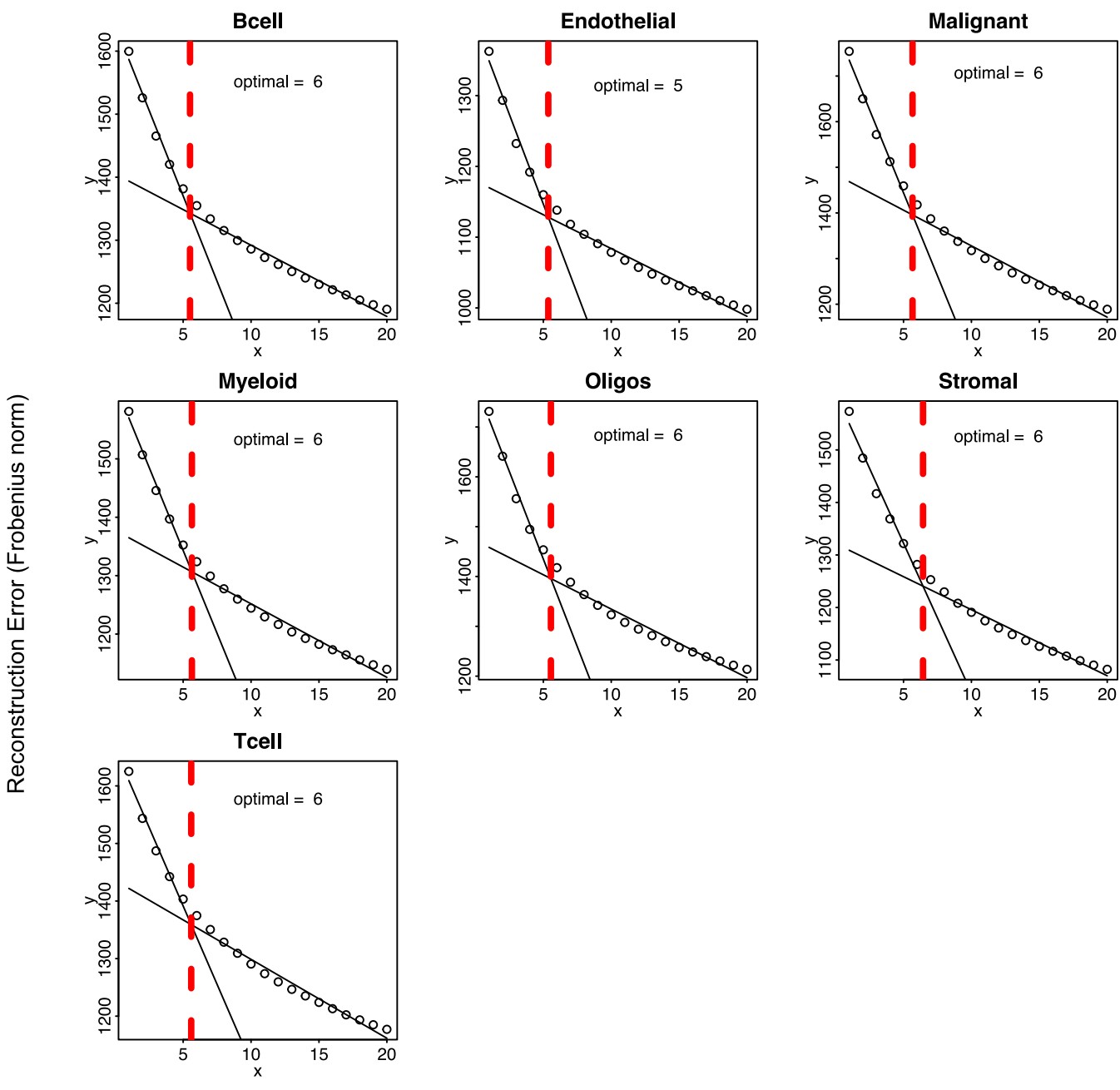

**Figure EV1.  Estimation of optimal rank for ICA factorization.**

Each scatter plot corresponds to a specific cell type shown at the top of the plot. *X*-axis spans the ICA factorizations starting from rank = 1 up to rank = 20, and *Y*-axis is the reconstruction error computed as Frobenius norm between original gene expression matrix and reconstructed gene expression matrix using ICA factorizations of ranks ranging from 1 to 20. Source data are available online for this figure.

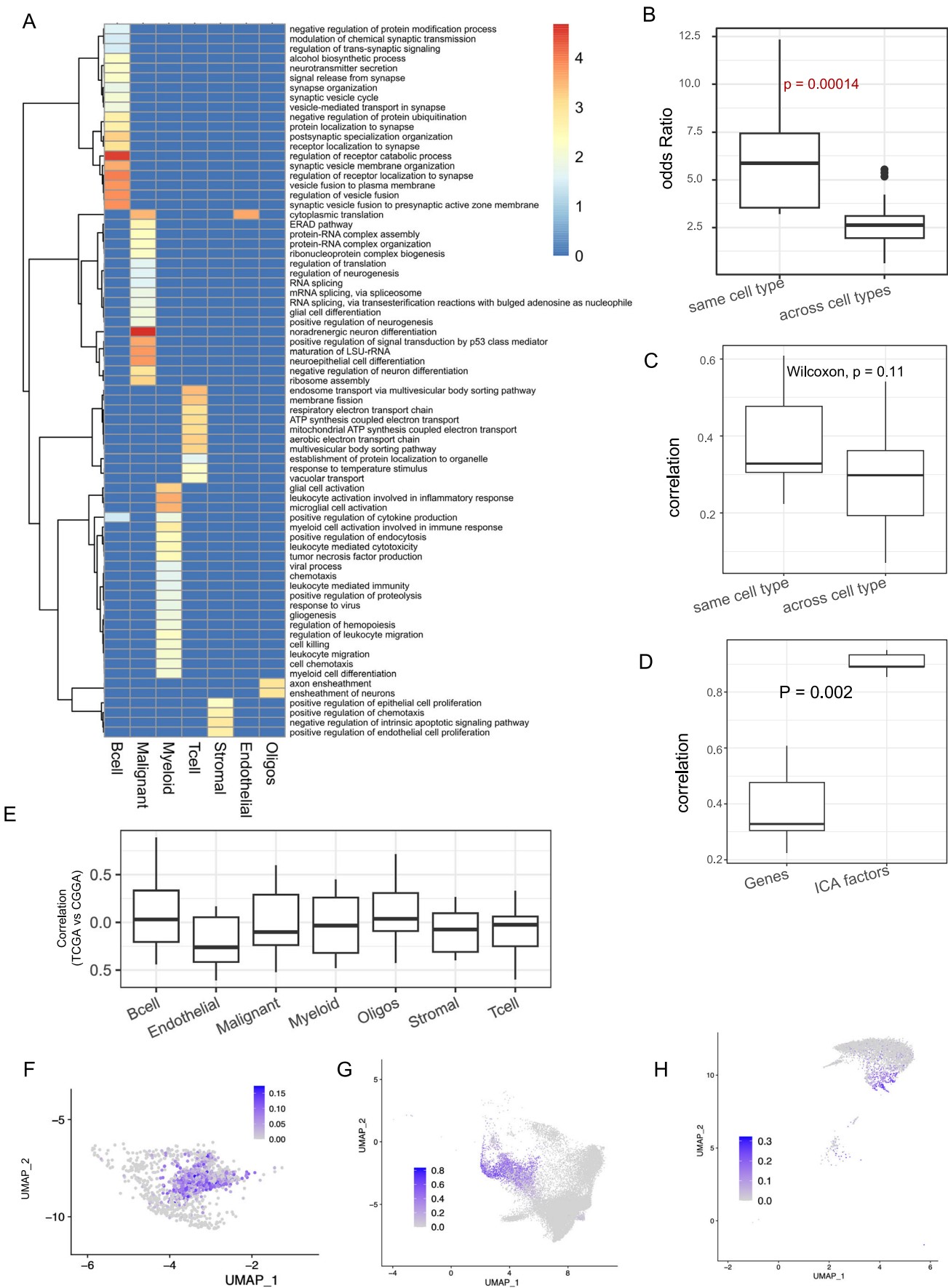

◀ **Figure EV2. Assessment of the quality of deconvolution and cell state inference.**

(A) Heatmap showing the enrichment of distinct functions among the top genes expressed by each cell type in the deconvolved data. The colors in the heatmap indicate the odds ratio of enrichment calculated as the fraction of genes belonging to the enriched functional category in the test set compared to the background set. The scale bar is provided on the right side. (B) Boxplot showing the odds ratio of enrichment among the top genes expressed between the same cell type and across different cell types in the deconvolved data in TCGA and CGGA (C). A box plot showing the cross-gene correlations (left column) of log-hazard ratio between TCGA and CGGA for the same cell type and across cell type comparisons. (D) Boxplot showing the distribution of correlation of hazard ratios for the same cell types between TCGA and CGGA, computed either using genes or using ICs. (B–D) *P* values from Wilcoxon's rank-sum test is shown. (E) Boxplot showing the distribution of correlation of hazard ratios for each cell type between TCGA and CGGA across 10 different randomizations of the input data. (B–E) The horizontal line in the middle is the median value, with lower and upper edges of the boxes corresponding to the 25th and 75th percentiles, and vertical lines corresponding to 1.5 times the interquartile range. (F–H) UMAP plots for T cells (F), malignant cells (G) and Oligodendrocytes (H) colored by the signature scores derived from the signature genes of IC1 (T cells), IC7 (Malignant cells), and IC7 (Oligodendrocytes), respectively. Source data are available online for this figure.

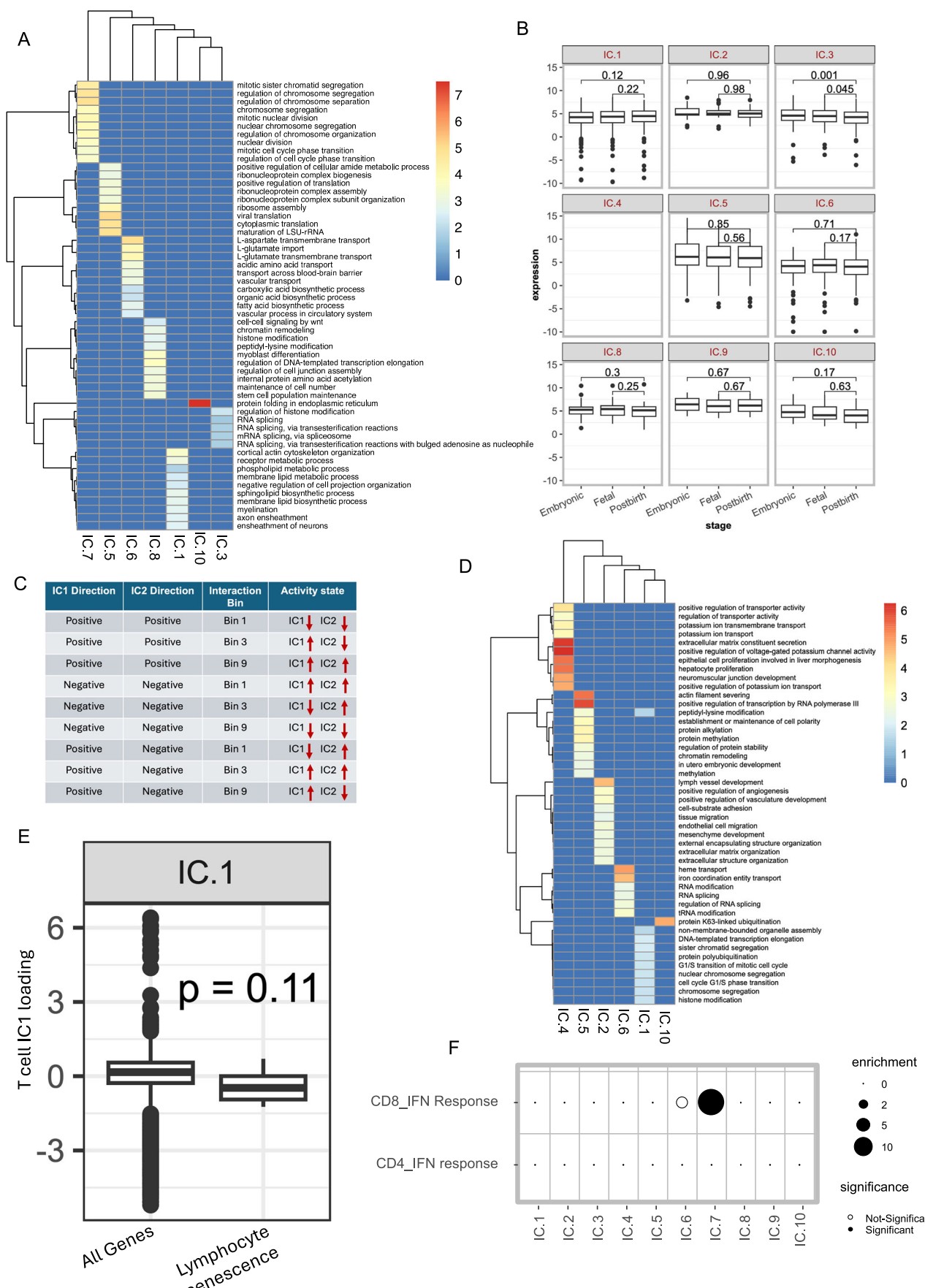

**Figure EV3. Identification of glioma stem cells and their interactions with other cell types in TME.**

(A) A heatmap showing the GO terms enriched across the negative signature genes of the Malignant cell ICs. The cell colors show the odds ratio of enrichment for each significant GO term in at least one of the comparisons. (B). A boxplot showing the expression (in log TPM units) of negative signature genes of various malignant cell ICs during the embryonic development of the human brain. The samples sizes (N) are as follows (IC1:233, IC2:24, IC3:277, IC4:0, IC5:106, IC6:164, IC8:58, IC:92, IC10:18) $P$ values from Wilcoxon's rank-sum test are shown. The pattern for IC.7 is shown in Fig. 3D and IC.4 is shown as blank, as there were no signature genes as per our criteria (C). A schematic showing the activity state of positive and negative signature genes of interacting ICs across different activity bins. Arrows indicate upregulation (upwards arrow) and downregulation (downwards arrow). For instance, the signature genes at the positive end of an interacting IC pairs, say IC1 and IC2 are simultaneously downregulated in Bin 1, simultaneously upregulated in Bin 9, and exhibit upregulation of IC1 and downregulation of IC2 in Bin 3. Analogously, the signature genes at the negative end of the interacting IC pairs, say IC1 and IC2, are simultaneously upregulated in Bin 1, simultaneously downregulated in Bin 9, and exhibit downregulation of IC1 and upregulation of IC2 in Bin 3. (D) A heatmap showing the GO terms enriched across the negative signature genes of the T cell ICs. The cell colors show the odds ratio of enrichment for each significant GO term in at least one of the comparisons. (E) A box plot showing the contribution of lymphocyte-specific senescence markers ($N = 6$) to IC1 of T cells along with the genome-wide ($N = 8747$) distribution. $P$ value from Wilcoxon's rank-sum test is shown. (B, E) The horizontal line in the middle is the median value with lower and upper edges of the boxes corresponding to the 25th and 75th percentiles, and vertical lines corresponding to 1.5 times the interquartile range. (F) A dot plot showing the odds ratio for the overlap between negative signature genes of T cells IC7 and interferon response signaling in CD8 and CD4 T cells. The size of dots is proportional to the odds ratio derived from Fisher's exact test, and $P$ values were adjusted for multiple comparisons across all the curated T cell state markers provided by Yanshuo et al. In both Fig. EV2A,D, colors in the heatmap indicate the odds ratio of enrichment calculated as the fraction of genes belonging to the enriched functional category in the test set compared to the background set. The scale bar is provided on the right side. The ICs are ordered as per the dendrogram on the top and only the ICs with at least one significantly enriched function are plotted. Source data are available online for this figure.

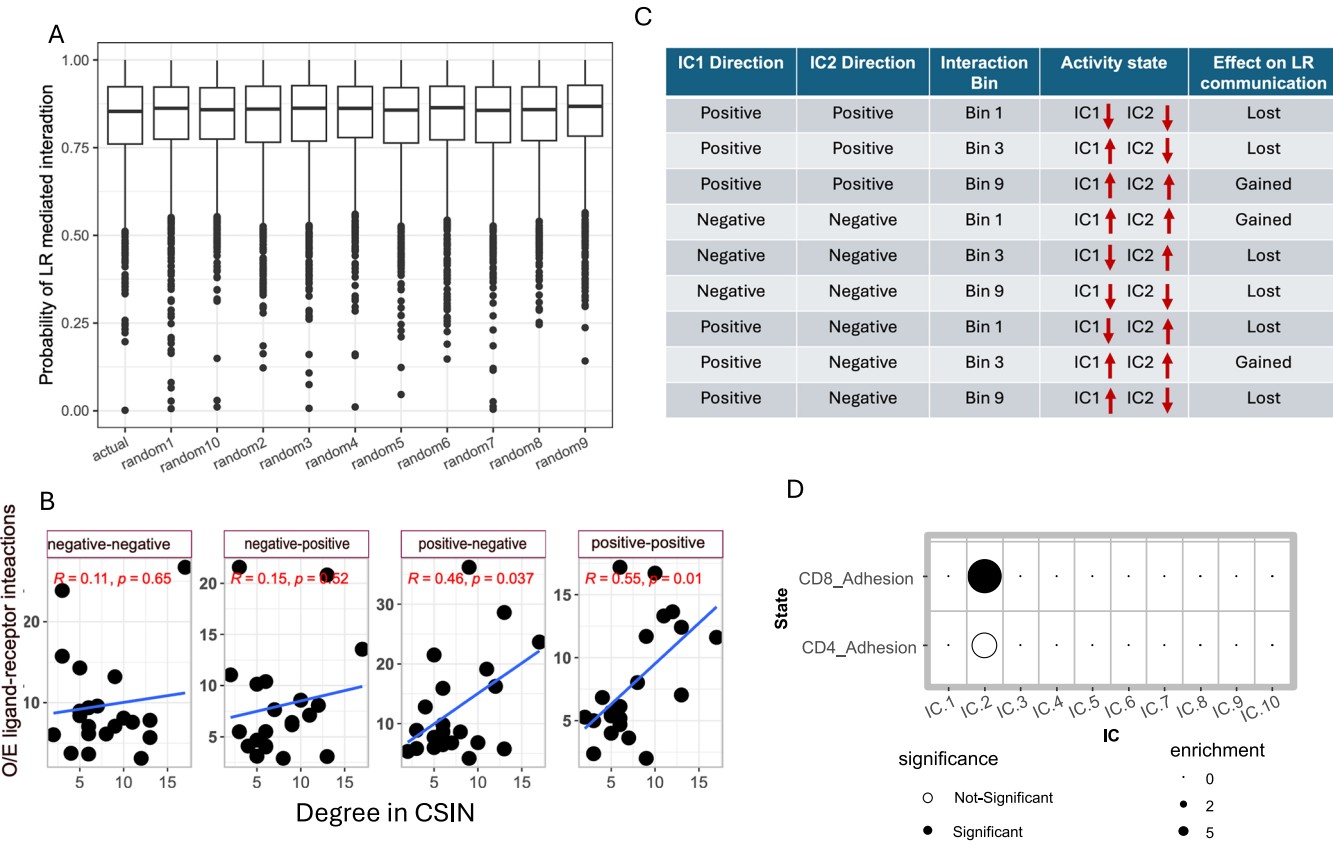

**Figure EV4. Analysis of ligand–receptor mediated cell state interactions.**

(A) Boxplot showing the probability for LR-mediated CSIs among the pairs of ICs in CSIN and 10 randomized networks. There is no statistically significant difference between the inferred CSIN and randomized networks. Samples sizes (N) from left to right are: 902, 1984, 1981, 1982, 1979, 1982, 1982, 1981, 1985, 1983, 1983. The horizontal line in the middle is the median value with lower and upper edges of the box corresponding to the 25th and 75th percentiles and vertical lines corresponding to 1.5 times the interquartile range. (B) Scatter plot between the degree of interactions between cell type pairs calculated from CSIN and observed by the expected (O/E) ratio of the complementary ligand receptor pairs present among the signature genes of all IC pairs between two cell types. We perform this calculation by considering all four possible combinations of the positive and negative signature genes of ICs as indicated at the top of each plot. Spearman's correlation coefficient and corresponding *P* value is shown in the plot. (C) A schematic table illustrating how the presence of complementary LR pairs among the negative or positive signature genes of ICs affects their activity across different interaction bins. The direction of arrows shows the status of positive and negative signature genes of their corresponding ICs across different interaction bins. A given LR interaction is deemed inactivated if one or both partners are downregulated. (D) A dot plot showing the odds ratio for the overlap between the negative signature genes of various T cell ICs and markers of T cell adhesion. Size of dots is proportional to the odds ratio derived from Fisher's exact test and solid points indicate the comparisons where the FDR-corrected *P* value < 0.20. Source data are available online for this figure.

A

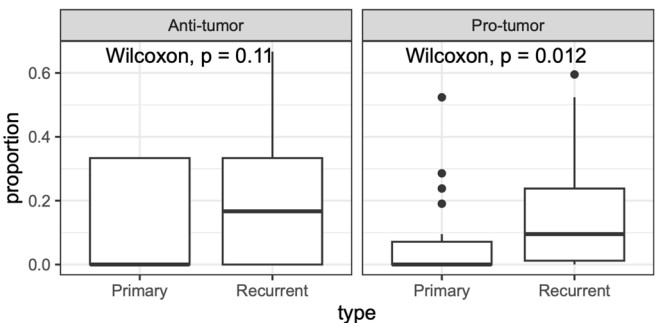

B

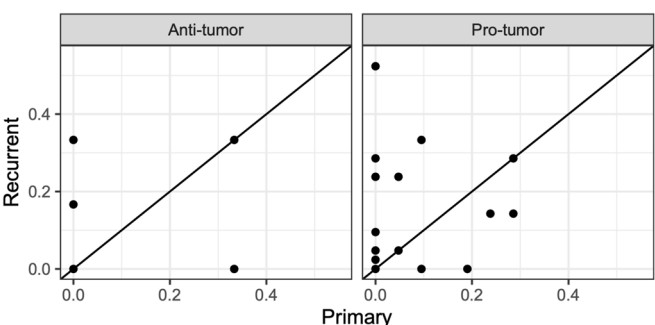

**Figure EV5.  Association of CSIN with therapy response.**

(A) Boxplots showing the distributions of CSI load for pro- and anti-tumor interactions among the primary ($N = 31$) and recurrent glioma ($N = 40$) patients. The horizontal line in the middle is the median value, with lower and upper edges of the box corresponding to the 25th and 75th percentiles, and vertical lines corresponding to 1.5 times the interquartile range. (B) A scatter plot with the same data as in (B). Each dot represents a paired value of interaction load derived from primary (x-axis) and recurrent (y-axis) tumors from the same patient. Source data are available online for this figure.

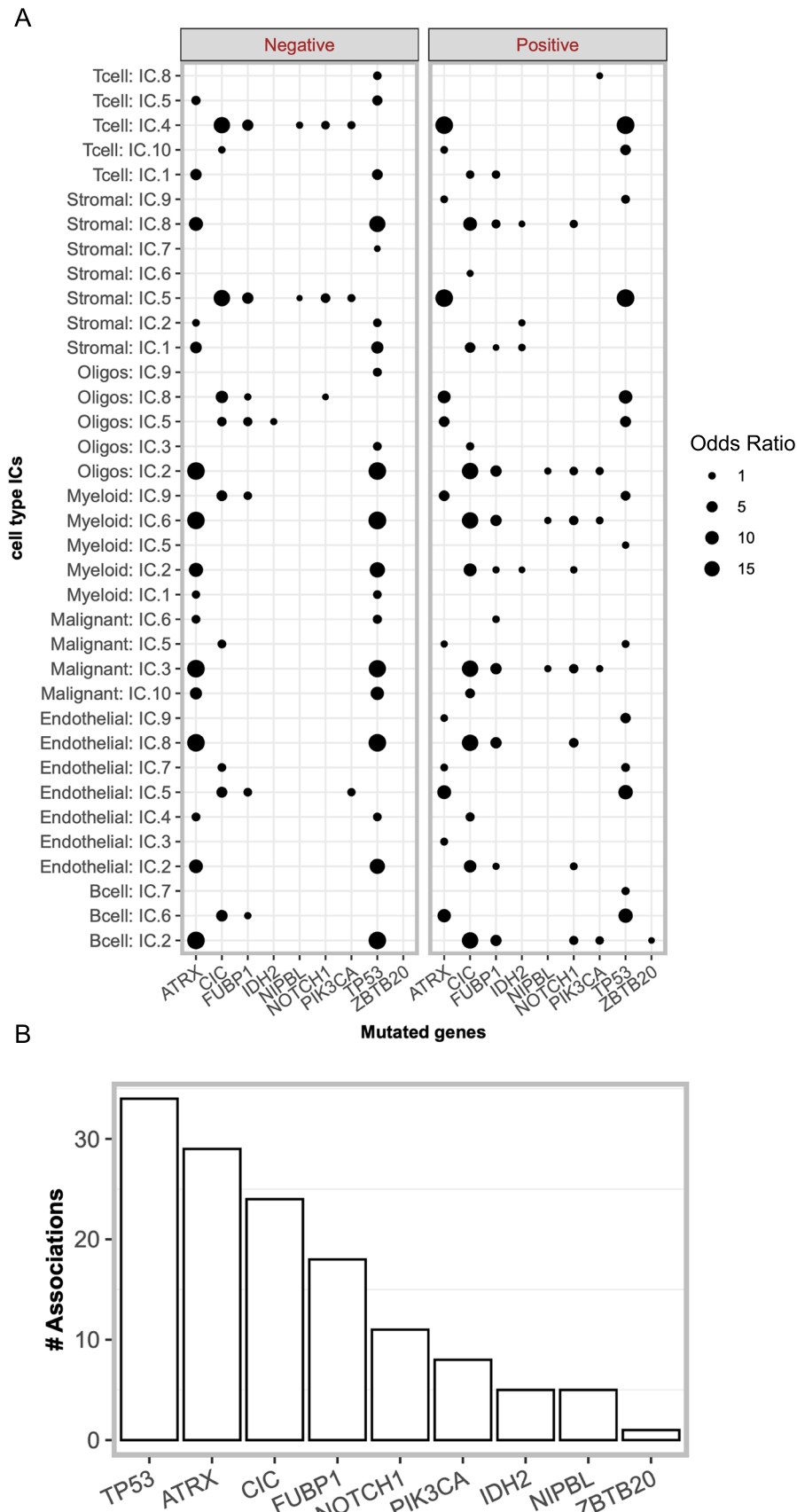

◀  **Figure EV6.  Association between somatic mutations and CSIN.**

(**A**) A dot plot showing the odds ratio for the overlap between IDH-mutant glioma patients having specific nonsynonymous somatic mutations (*x*-axis) and high activity score (~ top 33%) of ICs (*y*-axis). Size of dots is proportional to the odds ratio and solid points indicate Fisher's exact test FDR < 0.20. (**B**) A bar plot showing the number of significantly associated ICs for each significant gene in (**A**). Source data are available online for this figure.

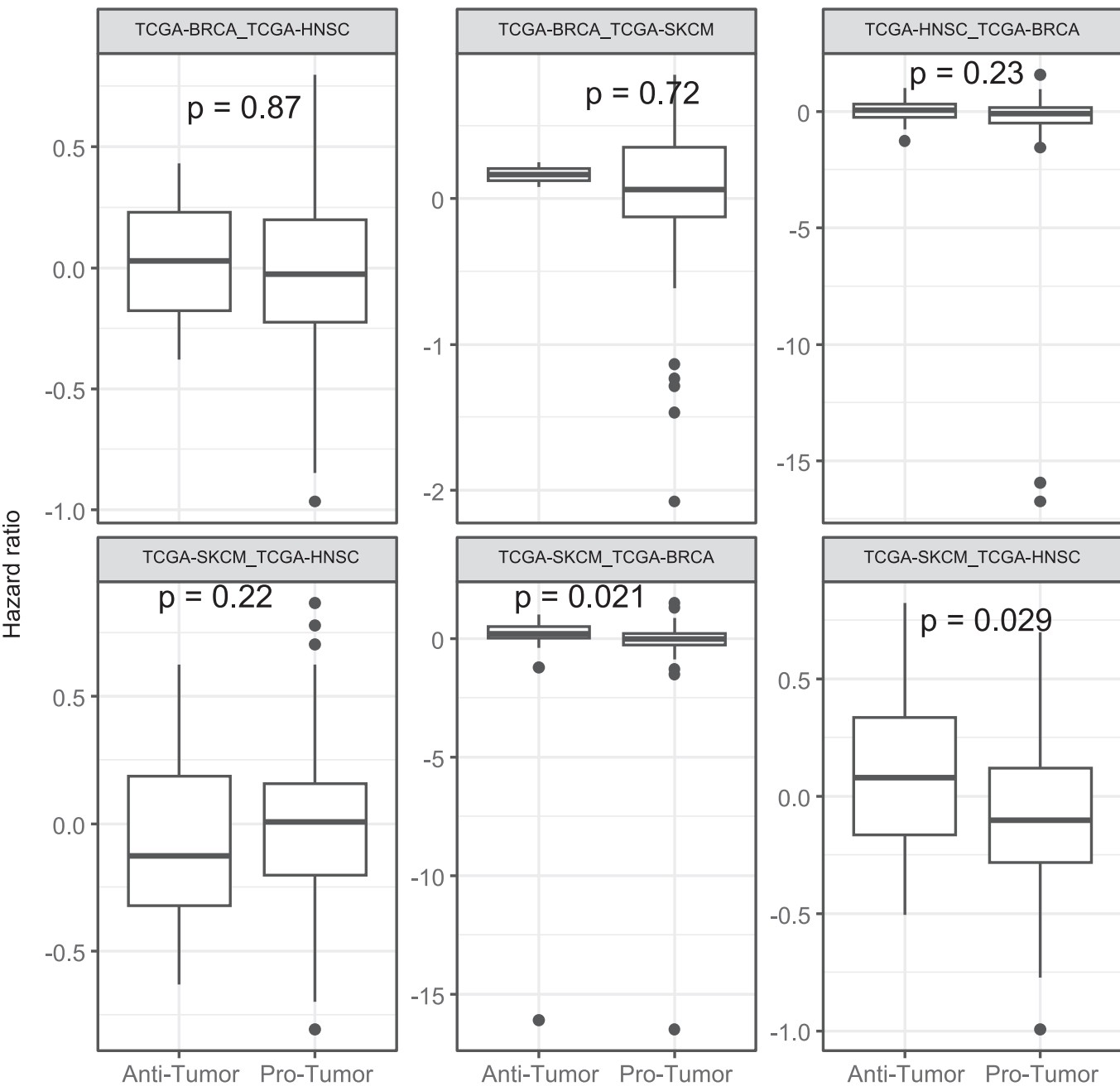

**Figure EV7. Cross cancer comparison of cell state interactions.**

Boxplots showing the distribution of hazard ratios for pro- and anti-tumor identified in one cancer type (source) and tested in another cancer type (target). The horizontal line in the middle is the median value with lower and upper edges of the box corresponding to the 25th and 75th percentiles and vertical lines corresponding to 1.5 times the interquartile range. Source and target cancer types included in each comparison are indicated at the top of each plot. *P* values are from Wilcoxon's rank-sum test. Source data are available online for this figure.

