## [Peer Review File · Molecular Systems Biology]

Identifying Prognostic Cell State Interactions in the Tumor Microenvironment of IDH-Mutant Gliomas

Arashdeep Singh, Bharati Mehani, Vishaka Gopalan, Sushant Puri, Kenneth Aldape, and Sridhar Hannenhalli

Corresponding author(s): Sridhar Hannenhalli (sridhar.hannenhalli@nih.gov)

Review Timeline:

Submission Date:	11th Jun 25
Editorial Decision:	16th Jul 25
Appeal Received:	18th Jul 25
Editorial Decision:	28th Jul 25
Revision Received:	17th Nov 25
Editorial Decision:	7th Jan 26
Revision Received:	30th Jan 26
Accepted:	11th Feb 26

Editor: Jingyi Hou

Transaction Report:

16th Jul 2025

RE: Manuscript MSB-2025-13172, Identifying Prognostic Cell State Interactions in the Tumor Microenvironment of IDH-Mutant Gliomas

Dear Dr. Hannenhalli,

Thank you again for submitting your work to Molecular Systems Biology. We have now received evaluations from all three reviewers. As you will see below, the reviewers have raised substantial concerns that unfortunately preclude publication of your manuscript in Molecular Systems Biology at this time.

While Reviewer #1 offered a more supportive assessment, both Reviewers #2 and #3 identified significant issues with the study. Of particular concern are the questions raised by Reviewer #2 regarding the broader applicability of the method to other cancer types, as well as Reviewer #3's concerns about the biological relevance of the proposed "cell-state interactions" and the signal-to-noise ratio associated with data transformation steps. These issues raise concerns about the underlying rationale and the overall robustness of the proposed approach. In particular, both Reviewers #2 and #3 rated the "Suitability for Publication" as "Low."

Under these circumstances and considering the overall low level of support provided by the reviewers, we see no other choice than to return the manuscript with the message that we cannot offer to publish it.

While we cannot pursue this manuscript further, we encourage you to transfer your study to our not-for-profit open-access sister journal, Life Science Alliance (LSA). We shared your manuscript and the accompanying reviews with LSA Executive Editor, Tim Fessenden, who is interested in these findings. He is pleased to offer consideration of this manuscript at LSA pending the following revisions:

- Comment on the biological significance of what ICs represent or measure, as requested by Reviewer 1 and as remarked on in point 3 by Reviewer 3.
- Address points 2 by Reviewer 2 and point 7 by Reviewer 3, on justifying the IC number used.
- Address point 3 by Reviewer 2, to make the online data repository more accessible.
- Temper claims to acknowledge that the deconvolution approach taken here is not validated, per Reviewer 3.

We understand that such a revision might need to be re-reviewed, in which case Dr. Fessenden will walk the Reviewers through our transfer process.

We encourage you to use the link below to transfer your manuscript to LSA. You do not need to revise the manuscript before transferring it to LSA. Once you transfer, Dr. Fessenden will email you an invitation to revise and resubmit, listing the same revision requests as mentioned above. Please feel free to reach out at t.fessenden@life-science-alliance.org if you have any questions about the LSA journal, the transfer process, or the revisions requested.

I am sorry that the review of your work did not result in a more favorable outcome on this occasion, but I hope that you will not be discouraged from sending your work to Molecular Systems Biology in the future. In any case, thank you for the opportunity to examine this work.

Kind regards,
Jingyi

Jingyi Hou, PhD
Senior Editor
Molecular Systems Biology

Reviewer #1:

In this paper, Singh et al. address the need to identify cell specific state interactions in the TME that are prognostic from bulk RNAseq. This need arises due to the lack of paired clinical data for single cell RNAseq and the lack of large scRNAseq datasets with appropriate read depth. The authors develop a pipeline which 1) uses single cell data to create cell type signatures to deconvolve bulk RNAseq into individual cell type expression, 2) applies independent component analysis (ICA) to generate ICs

which represent gene expression programs/transcriptional states from the deconvolved expression data 3) pairs every IC in such a way to represent the interactions between the joint activity between any two ICs from two different cell types, 4) runs a cox proportional hazard model to identify interactions that are significantly associated with survival.

Overall, the paper is very thorough and includes several validation sets sourced from the current literature to compare findings. Not only does the study utilize the Chinese Glioma Genome Atlas as the validation set, to investigate what these different ICs represent the authors utilize outside literature to compare their findings. The statistical techniques the authors used to prove robustness were also well done, including using cross-validation and bootstrapping.

In terms of weaknesses, it was a little unclear to me how the authors were comparing joint activity between the ICs when each IC is independent in each cell type. For example, depending on the cell type, this same set might be either up or down depending. Meanwhile, this paper is assessing the activity of what seems like independent gene modules that are jointly elevated, depleted, or going in opposite directions. I feel like the authors could do a better job explaining/ mentioning exactly what these ICs are. It reminds me a lot of WGCNA or a weighted gene co-expression network analysis.

Minor critiques:

-Why Use a dot plot to represent the Odds ratios of the overlaps (Figure 2B)? There is a package called GeneOverlap in R that uses a heatmap to show the Odds ratio between the two lists of genes along with the p-value in each of the heatmap squares to show the significance of the OR. I feel as though the box plot comparison in figure S1 is better than the dot plot as well.

Typos/Grammar:

-pg. 7: Remove comma between "myeloid, and leukocytes"

-pg 15: "we observed... the negative genes of three of the malignant cell ICs significantly..." (remove "the")

Reviewer #2:

In this manuscript, Singh et al. developed a computational pipeline named CSI-TME for inferring prognostic cell-state interactions (CSIs) using tumor associated bulk and single-cell RNA sequencing data. Conceptually, they transferred the gene interactions to transcriptional state interactions, rather than only focusing on the interactions between distinct cell types. CSI-TME identified a highly reproducible, therapy response and somatic mutation related cell-state interaction network (CSIN) in the TME of IDH mutant glioma, and the partial CSIs were mediated by ligand-receptor interactions. Most analytical results were derived from the TCGA cohort, with subsequent validation in the independent CGGA cohort. Overall, this manuscript is generally well-organized. My comments are as follows.

Major Comments:

1. This study currently focused on IDH-mutant gliomas, but the broader applicability of CSI-TME to other cancer types (or IDH-wt) remains an important open question. It is valuable to explore the roles of CSIs are cancer-type specific or generalizable. If supporting datasets are unavailable, the authors should at least provide their perspective with relevant literatures in the section of "Discussion".
2. The number of independent component (IC) is a key parameter for independent component analysis (ICA) and downstream analysis. The authors only stated that "it should be sufficient to capture the transcriptomic and functional heterogeneity in a cell type" in the section of "Methods". It is arbitrary. A comparative analysis demonstrating the rationale for selecting the number 10 (versus alternative values) should be provided.
3. While the GitHub repository (<https://github.com/hannenhalli-lab/CSI-TME>) provides several R scripts, a tutorial (either as a detailed README file or through external documentation) for introducing the usage of CSI-TME is required, which could facilitate broader adoption of the computational pipeline.
4. All figures in the manuscript require careful revision, including the size, font format, labels, etc. For example, Figure 1A contains much more information than Figure 1B, so the size of Figure 1A should be larger than Figure 1B for more clear visualization. For Figure 2F and 2G, implementing different colors to the bars or boxes facilitates distinguishing the anti-tumor and pro-tumor groups. Please ensure sufficient spacing between cell type labels and network nodes to enhance graphical clarity in Figure 2H and 2I.

Minor Comments:

1. Do CCGA and CGGA refer to two different cohorts? Or is "CCGA" just a spelling error?
2. As described in the section of "Methods", 10 independent components are extracted for each cell type. So the shape of patient IC matrix of different cell state should be same in Figure 1A. In addition, the symbol " \approx " seems to be inappropriate.
3. The last sentence in the Page 3 is confusing. Do three co-activity bins include bin 3, bin 7 and bin 9? But the bin 7 and 9 are with both high and low activity. And does S1 (S2) refer to IC1 (S2) in Figure 1B. If so, high activity of S1 and low activity of S2 corresponds to bin 7 rather than bin 3. The symbol "..." should be replaced by "Med" in Figure 1B.
4. In Figure 2A, does the pink point represent unknown cell type? If so, this cell type should be annotated due to its relative large cell number. Please provide the cell count for each identified cell type.
5. The meaning of color bar in the heatmap should be clarified in Figure S1A, S2A, S2D, S4C, S4E, 5C, and 7B.
6. In Figure 2C, the cross-IC correlations between TCGA and CGGA for each cell type are all closed to 0.8 and nearly same. Is it reasonable?
7. P value should be listed in Figure S1D.
8. Figure S1J (for positive signature genes) is the same as Figure S1K (for negative signature genes). For each cell type, all 10 ICs should be plotted rather than selecting a subset. Please check it carefully.
9. The authors mentioned that only the IC7 negative signature genes had higher expression during the embryonic phases of

brain development and significantly decreased in fetal brain and mature adults. But as shown in Figure S2B, the negative signature genes of IC3 also exhibited this trend, albeit less significant than IC7. Besides, IC4 rather than IC7 (has been shown in Figure 3D) should be plotted in Figure S2B.

10. Figure S2C requires clarification. Please expand the figure legend with detailed descriptions to enhance interpretability.

11. The rank of IC (or Dim) should be unified (Figure 2D, S1J, S1K, 3A, 3C, 3F, S2A, S2D, S2F, 5F, 5G, S3D, and S5A). The form of "IC.1 IC.2 IC.3 IC.4 IC.5 IC.6 IC.7 IC.8 IC.9 IC.10" is suggested.

12. In Figure 3E, only IC7 of malignant cells are mentioned in the manuscript to interact with immune ICs. How about IC5 and IC6?

13. It is interesting to observe that the negative signature genes of T cells IC1 significantly overlapped with the markers of senescent fibroblasts (Page 12). Is it caused by the substantial overlap between T cell and fibroblast senescence markers? If not, does it suggest that T cells can induce fibroblast senescence?

14. Fisher's exact test is used many times in the manuscript. Do both of "Fisher test" and "Fisher's test" refer to Fisher's exact test? Please use the consistent terminology throughout the manuscript.

15. It is counterintuitive to observe that CSIs dominant in IDH-A tumors are predominantly anti-tumor. The authors should provide further explanation about this point.

16. The authors claimed that CSIs can explain the survival of IDH-A and IDH-O patients which cannot be explained by co-deletion of 1p/10q chromosomal arms. Do these included patients all harbor this co-deletion? If not, stratifying the patients based on deletion status and conducting comparative survival analyses are required.

17. Figure 5A is also shown in the Figure S3B. To avoid redundancy, do not show the same figure repeatedly.

18. Using spatial transcriptomic data, the authors observed that negative signature genes of endothelial IC-3 and positive signature genes of the malignant IC-2 were significantly proximal in the Visium slides. Appropriate statistical testing should be performed to quantify the significance rather than relying solely on visual assessment.

19. Are "mutated genes" and "mutation burden" conceptually same in Figure 7A and the figure legend?

Reviewer #3:

Summary:

Singh et al. present CSI-TME, a computational pipeline that infers cell state interactions in IDH-mutant gliomas from bulk RNA sequencing data. To accomplish this, they make use of existing bulk RNA sequencing data from TCGA and CCGA, deconvolving it to produce gene expression matrices for each cell type. On these matrices, they perform individual component analysis (ICA), which are considered to represent latent transcriptional programs, or cell states. Some of the ICs are reported to correspond to known gene expression programs, such as T cell proliferation or the glioma stem cell state. They then identify cell states pairs which have some degree of joint activity, which are deemed to be cell state interactions (CSI). From these CSIs, a network is constructed, the cell state interaction network (CSIN). The authors then assess the extent to which different CSIs correspond to patient outcome, and each CSI is labelled pro-tumor or anti-tumor. Having annotated their CSIN, they show that patient outcome corresponds with the relative abundance of pro- or anti-tumor interactions. The authors refine their analysis by examining CSIs that have enriched ligand-receptor pairs, finding that interaction strength in the CSIN correlates with the abundance of ligand-receptor pairs. Finally, cell states are linked to somatic mutations, and Singh et al. suggest that somatic mutation may to some extent drive cell state formation.

General remarks:

This work builds on the already well-developed deconvolution methods available for bulk sequencing data by producing a pipeline to gain deeper insight into deconvolved gene expression data and cell state resolution. As described below, the authors do not validate their cell state resolution deconvolution before using these data to perform ICA. Since this is primarily a methods paper aimed at enabling deeper interpretation of widely available bulk data, the introduction of cell state-level deconvolution is a potentially valuable advancement. Yet without validation, readers have no way to assess its accuracy.

Moving to the second step, the CSI, I also have concerns about the signal/noise ratio when performing additional transformations on top of deconvolution, which is already a somewhat noisy process. Specifically, the discrepancy in the results between the two cohorts, TCGA and CCGA, is concerning. Therefore, biological findings from this approach need to be taken with a grain of salt due to the level of abstraction from the raw sequencing data and should be followed up with supportive evidence or validation.

Thus, while the ideas are novel, the methods are neither well-established nor thoroughly validated. If the methods or the biological insights from CSI-TME are robust and reproducible, which has not been confirmed (see below), this tool could be of use to any researcher who has abundant bulk sequencing data but lacks access to single cell data, hopefully in many cancer types. With that said, using single cell data to infer cell state interactions would likely be preferred in most cases.

Major points:

1. The introduction requires a thorough review and revision. The most concerning citation is found in the first paragraph of the introduction, the authors state that gliomas recur in just 50% of cases, citing 2019 paper from Miller et al (2019) (1). This is not correct, as even low-grade gliomas almost always recur (2,3). It appears that the cited paper looks at recurrence over about only about 6 years, giving a recurrence rate of 54% for that timespan. However, over longer timelines recurrence is much more likely. This needs to be corrected as it is quite misleading.

2. Any utility of the CSI-TME pipeline is highly dependent on the validity of the deconvolution used. However, the authors do not spend time validating the deconvolution itself. The ICA-derived cell states are annotated without a defined or standardized method, leaving the reader uncertain whether these states accurately reflect the known spectrum of cell states in IDH-mutant gliomas. This is particularly concerning given that the cellular composition of IDH-mutant gliomas has been extensively characterized, and expected profiles are well established. Downstream analyses are suspect without such validation.

3. Cell state interactions are inferred from co-occurrence co-absence, or mixed presence of IC enrichment between cell types (Fig. 1B, bins 1, 3, 7, 9). However, the authors do not support the inferred indirect CSIs, though this is the major innovation of the paper. It is surely possible that the presence or absence of cell states can co-occur without meaningful biological interaction, whether direct or indirect. How is it known that cell state co-occurrence (or other combinations) are not merely coincidental? Co-occurrence does not necessarily imply functional relation. Much of the paper from figure 2 onward is dependent on this assumption. The authors assess only the direct interactions by showing that cell state interactions do correlate with the abundance of ligand-receptor interactions (Fig. 5A). However, there are already tools to detect ligand-receptor interactions such as LIANA or CellphoneDB (4,5), and they rely on co-expression of the two populations, so this part is already well established and developed in the field. Thus, one of the novelties of this manuscript, the purported ability to detect indirect interactions, which cannot be done with ligand-receptor analysis, is not supported.

4. The authors show dozens of cell state interactions yet explore only a few of them (e.g. putative GSCs with B and T cells). Some selected CSIs do correlate with patient outcome (e.g. interferon response gene in T cells with PTN and FBXO5 in malignant cells), but overall, how many of these are biologically meaningful? I also noticed that in describing Figure 4, the authors mention that only 103/160 (~ 62%) interactions have the same direction of hazard in TCGA and CCGA. This does not inspire confidence in the overall robustness of CSI-TME.

5. Some interactions conflict with biological expectations. In principle, challenging conventional wisdom is a benefit. However, in the absence of further validation and characterization, this starts to look like noise. For example, is it believable that a top interaction is between B cells and malignant cells? Fig. 3G is the interaction plot for cell states in T cells and malignant cells. It looks like there is only one anti-tumor interaction. It is surely expected that many T cell-malignant cell interactions would be detrimental, but that all but one interaction is pro-tumor seems dubious given the critical role of T cells in anti-tumor immunity. In Fig. 6B, it looks like there are more pro- and anti-tumor interactions in recurrent tumors. This is at odds with expectation. Overall, figures 6B and C are probably not interpretable with only 3 CSIs. These findings could be believable with elaboration and/or validation, but otherwise this looks like noise or general unreliability of the pipeline.

6. In the introduction, it is asserted that cohorts of single cell RNA sequencing data for IDH-mutant gliomas are "lacking". While it is true that bulk datasets are more abundant, single cell datasets do exist for gliomas (6-8). The authors even use some of these datasets to annotate the ICA-derived cell states (though, as noted in Comment #2, this is done loosely and without a clearly defined methodology). However, they do not use these datasets to validate any of their results-despite the fact that interactions inferred from bulk data should also be detectable in single-cell data. While the available single-cell datasets may be too small for full exploration, they are sufficiently large for validation purposes.

7. In the methods section, the authors state that the decision to use 10 individual components was arbitrary. However, it is not obvious that all 10 ICs are biologically meaningful for all cell types. Nor is it obvious that all important ICs have been investigated. The authors should show that the ICs they used capture meaningful variation and that additional ICs do not capture important additional information.

Minor points

1. In the introduction, in the third paragraph, the Singh et al. say that tumor and microenvironmental cells can exist in multiple transcriptional cell states. This is true, but two of the three citations reference studies in IDH-WT glioblastoma (Nefel et al. 2019 and Batchu et al. 2023), which is not the disease they are reporting on. They should instead cite papers such as Tirosh et al. (2019) or Ochocka et al. (2021) which show this in IDH-mutant glioma (6,7).

2. Figure 3G is difficult to read because the legends use the same colors but are not labelled as to which represents nodes and which represents edges.

3. Fig 4J is difficult to interpret with so few patients in the interaction group.

4. There are numerous typos which should be addressed.

References

1. Miller JJ, Loebel F, Juratli TA, et al. Accelerated progression of IDH mutant glioma after first recurrence. *Neuro-Oncol.* 2019;21(5):669-677. doi:10.1093/neuonc/noz016

2. Daniels TB, Brown PD, Felten SJ, et al. Validation of EORTC Prognostic Factors for Adults With Low-Grade Glioma: A Report

Using Intergroup 86-72-51. *Int J Radiat Oncol.* 2011;81(1):218-224. doi:10.1016/j.ijrobp.2010.05.003

3. Stewart J, Sahgal A, Chan AKM, et al. Pattern of Recurrence of Glioblastoma Versus Grade 4 IDH-Mutant Astrocytoma Following Chemoradiation: A Retrospective Matched-Cohort Analysis. *Technol Cancer Res Treat.* 2022;21:15330338221109650. doi:10.1177/15330338221109650

4. Dimitrov D, Türei D, Garrido-Rodriguez M, et al. Comparison of methods and resources for cell-cell communication inference from single-cell RNA-Seq data. *Nat Commun.* 2022;13(1):3224. doi:10.1038/s41467-022-30755-0

5. Troulé K, Petryszak R, Cakir B, et al. CellPhoneDB v5: inferring cell-cell communication from single-cell multiomics data. *Nat Protoc.* Published online March 25, 2025:1-29. doi:10.1038/s41596-024-01137-1

6. Ochocka N, Segit P, Walentynowicz KA, et al. Single-cell RNA sequencing reveals functional heterogeneity of glioma-associated brain macrophages. *Nat Commun.* 2021;12(1):1151. doi:10.1038/s41467-021-21407-w

7. Tirosh I, Venteicher AS, Hebert C, et al. Single-cell RNA-seq supports a developmental hierarchy in human oligodendroglioma. *Nature.* 2016;539(7628):309-313. doi:10.1038/nature20123

8. Venteicher AS, Tirosh I, Hebert C, et al. Decoupling genetics, lineages, and microenvironment in IDH-mutant gliomas by single-cell RNA-seq. *Science.* 2017;355(6332):eaai8478. doi:10.1126/science.aai8478

** As a service to authors, EMBO Press offers the possibility to directly transfer declined manuscripts to another EMBO Press title or to the open access journal *Life Science Alliance* launched in partnership between EMBO Press, Rockefeller University Press and Cold Spring Harbor Laboratory Press. The full manuscript and if applicable, reviewers' reports, are automatically sent to the receiving journal to allow for fast handling and a prompt decision on your manuscript. For more details of this service, and to transfer your manuscript please click on Link Not Available. **

The authors appealed the decision.

28th Jul 2025

RE: Manuscript MSB-2025-13172R-Q, "Identifying Prognostic Cell State Interactions in the Tumor Microenvironment of IDH-Mutant Gliomas"

Dear Dr. Hannenhalli,

Thank you for your message related to our decision on your manuscript MSB-2025-13172. I apologize for the delay in getting back to you, which was due to the high number of submissions that we have been receiving. I have now had the chance to consider the manuscript and the points raised in your appeal letter, and have also discussed them with the editorial team.

We appreciate that you are willing to revise the manuscript and that in your preliminary response you clarify some points related to the report of Reviewers #2 and #3. Nevertheless, we think that beyond providing further clarifications, significant additional analyses would be required in order to convincingly address the issues related to the robustness and the broader applicability of the proposed method. As such, at this stage it seems unclear to us whether the revisions would make the study suitable for publication in *Molecular Systems Biology*.

However, considering that the reviewers did have positive words for the topic and novelty of the study, we would not be opposed to considering a revised and extended study as a new submission, provided that all the raised concerns have been satisfactorily addressed.

A resubmitted work would have a new number and receipt date. It will be editorially evaluated afresh and its novelty will be re-assessed at the time of submission. As you probably understand, given that addressing the issues raised involves substantial additional analyses with unclear outcome, we can give no guarantee about the eventual acceptability of the study. If you do decide to resubmit the extended/revised manuscript, we would ask you to enclose with your resubmission a point-by-point response to the points raised in the present review.

I apologise once again for the slow response and I hope that the comments above will be helpful in deciding how to further proceed with this work.

Kind regards,
Jingyi

Jingyi Hou, PhD
Senior Editor
Molecular Systems Biology

** As a service to authors, EMBO Press offers the possibility to directly transfer declined manuscripts to another EMBO Press title or to the open access journal *Life Science Alliance* launched in partnership between EMBO Press, Rockefeller University Press and Cold Spring Harbor Laboratory Press. The full manuscript and if applicable, reviewers' reports, are automatically sent to the receiving journal to allow for fast handling and a prompt decision on your manuscript. For more details of this service, and to transfer your manuscript please click on Link Not Available. **

Reviewer #1:

In this paper, Singh et al. address the need to identify cell specific state interactions in the TME that are prognostic from bulk RNAseq. This need arises due to the lack of paired clinical data for single cell RNAseq and the lack of large scRNAseq datasets with appropriate read depth. The authors develop a pipeline which 1) uses single cell data to create cell type signatures to deconvolve bulk RNAseq into individual cell type expression, 2) applies independent component analysis (ICA) to generate ICs which represent gene expression programs/transcriptional states from the deconvolved expression data 3) pairs every IC in such a way to represent the interactions between the joint activity between any two ICs from two different cell types, 4) runs a cox proportional hazard model to identify interactions that are significantly associated with survival.

Overall, the paper is very thorough and includes several validation sets sourced from the current literature to compare findings. Not only does the study utilize the Chinese Glioma Genome Atlas as the validation set, to investigate what these different ICs represent the authors utilize outside literature to compare their findings. The statistical techniques the authors used to prove robustness were also well done, including using cross-validation and bootstrapping.

In terms of weaknesses, it was a little unclear to me how the authors were comparing joint activity between the ICs when each IC is independent in each cell type. For example, depending on the cell type, this same set might be either up or down depending. Meanwhile, this paper is assessing the activity of what seems like independent gene modules that are jointly elevated, depleted, or going in opposite directions. I feel like the authors could do a better job explaining/ mentioning exactly what these ICs are. It reminds me a lot of WGCNA or a weighted gene co-expression network analysis.

We are thankful to the reviewer for their thoughtful review of our work on CSI-TME. We are highly encouraged by the positive comments and below, we address the main concerns raised by the reviewer.

The first concern is the lack of clarity on how joint activity of the ICs was computed. Before defining the joint activities, we first defined the activity of individual ICs in a specific sample. For this, we used the IC score of the mixing matrix derived from ICA, with upregulation corresponding to the percentile rank of > 67 (top third) of the IC scores, and downregulation with percentile rank < 33 (bottom third). Next, for two candidate IC pairs (say ic1 and ic2) in a given bin (say bin 9), we stratified the patients into two categories using an **AND** condition such that patients where both ic1 and ic2 were upregulated (since we are considering bin 9) were assigned a value of 1 and remaining patients were assigned a value of 0. We then performed cox-regression based on the binarized joint activity vector of an IC pair while controlling for the activity of individual ICs. We repeated the same procedure for all 3 activity bins (1, 3, 9) considered in this study. We have now added an additional text at page 5 in the legend of Figure 1 to clarify this.

Coming to the second set of questions around WGCNA, some clarification is in order. Indeed, as the reviewer says, ICs within a cell type are statistically independent. However, we only assess 'interactions' between ICs of two distinct cell types, and never for the same cell type. Next,

regarding WGCNA, while some of its objectives are indeed shared with CSI-TME, these are essentially two different techniques with two different goals. WGCNA is a clustering tool and identifies modules of co-expressed genes based on observed gene-gene correlation patterns. CSI-TME identified network of cell states (ICs) across cell types, similar to WGCNA, but in contrast to WGCNA, an edge between two ICs (not genes) does not represent correlation, but instead indicates that the co-occurrence of the two states is associated with a clinical outcome; note that the two such ICs are not necessarily correlated, as in WGCNA-like approach. We have tried to make this contrast more explicit at line # 132

Minor critiques:

-Why Use a dot plot to represent the Odds ratios of the overlaps (Figure 2B)? There is a package called GeneOverlap in R that uses a heatmap to show the Odds ratio between the two lists of genes along with the p-value in each of the heatmap squares to show the significance of the OR. I feel as though the box plot comparison in figure S1 is better than the dot plot as well.

We thank the reviewer for this comment. As per the suggestion, we explored the GeneOverlap Package in R. However, there is one technical issue that prevented us from being able to use this package. When comparing the overlap of top expressed genes between different cell types, the size of universe, i.e., the genes that were confidently deconvolved in both candidate cell types, changes for every pair of cross-cell type comparisons. However, the **newGOM** function from GeneOverlap package that is used to compute overlap statistics for all pairwise comparisons across 7 cell types accepts only a single universe for all the comparisons. Thus, using same universe for all cell type pairs, for instance, between Malignant cells in TCGA and CGGA, and between T cells would be wrong, as the deconvolved genes are substantially different in each cell type across two cohorts. Given this limitation of GeneOverlap, we address the reviewer's main concern about lack of p-values in the plot, by stating the p-value for each overlap in the figure 2B legend.

Typos/Grammar:

-pg. 7: Remove comma between "myeloid, and leukocytes"

We thank the reviewer for pointing this mistake and have made a correction.

-pg 15: "we observed... the negative genes of three of the malignant cell ICs significantly..." (remove "the")

We thank the reviewer for pointing this mistake and have made this correction

Reviewer #2:

In this manuscript, Singh et al. developed a computational pipeline named CSI-TME for inferring prognostic cell-state interactions (CSIs) using tumor associated bulk and single-cell RNA sequencing data. Conceptually, they transferred the gene interactions to transcriptional state interactions, rather than only focusing on the interactions between distinct cell types. CSI-TME identified a highly reproducible, therapy response and somatic mutation related cell-state interaction network (CSIN) in the TME of IDH mutant glioma, and the partial CSIs were mediated by ligand-receptor interactions. Most analytical results were derived from the TCGA

cohort, with subsequent validation in the independent CGGA cohort. Overall, this manuscript is generally well-organized. My comments are as follows.

We are thankful to the reviewer for their encouraging and valuable feedback. In response, we provide detailed clarifications, incorporate additional analysis wherever deemed necessary, and make the suggested changes as described below.

Major Comments:

1. This study currently focused on IDH-mutant gliomas, but the broader applicability of CSI-TME to other cancer types (or IDH-wt) remains an important open question. It is valuable to explore the roles of CSIs are cancer-type specific or generalizable. If supporting datasets are unavailable, the authors should at least provide their perspective with relevant literatures in the section of "Discussion".

Response – We thank the reviewers for raising this point and fully agree with the need for additional analysis. To demonstrate broader applicability, we have now applied CSI-TME to 3 additional cancer types including TCGA-BRCA, TCGA-SKCM, and TCGA-HNSC. In all three cases, we observed that CSI-TME was able to identify pro- and anti-tumor CSIs that were cross validated using 70% accuracy in 10 bootstraps. Consistent with our findings in IDH-mut, in each case, there were far greater pro-tumor interactions in the TME, and importantly, the CSIs were associated with resistance to various targeted and immunotherapies in an expected manner. Most importantly, we observed that CSIs detected in TCGA-BRCA were associated with the progression of breast pre-cancerous lesions. These results, supporting general applicability of CSI-TME. We now also provide a streamlined pipeline that can be used to train CSI-TME on any new cohort of interest.

Additionally, as recommended by the reviewer, we have performed cross-cancer comparison of the CSIs by utilizing the ICA models inferred in one cancer type to identify cell state interactions in other cancer types. This analysis suggests that CSIs are highly cancer type specific in terms of their clinical impact. This is broadly consistent with the inter-tumor heterogeneity as well as the fact that prognostic genes are highly cancer type specific (<https://pmc.ncbi.nlm.nih.gov/articles/PMC9042322/>).

We have now added a new section at line 786 describing these results

2. The number of independent component (IC) is a key parameter for independent component analysis (ICA) and downstream analysis. The authors only stated that "it should be sufficient to capture the transcriptomic and functional heterogeneity in a cell type" in the section of "Methods". It is arbitrary. A comparative analysis demonstrating the rationale for selecting the number 10 (versus alternative values) should be provided.

Response – We appreciate this point. Unlike PCA, where each successive component explains a decreasing amount of variance of the data, ICA currently lacks an agreed-upon standard approach for estimating the optimal number of ICs. There are two prominent approaches published for IC, – MSTD and optICA, that we had tried. However, results from two approaches did not converge. For example, in the case of stromal cells, as shown below, MSTD criteria estimated > 40 optimal components while optICA estimated just 8.

For the purposes of CSI-TME, we chose 10 ICs based on the analysis of reconstruction error through various iterations of ICs with rank 1 up to 20. We used segmental curve fitting to approximate the IC rank beyond which there was no significant gain in the model accuracy (reconstruction error). The accuracy was ascertained by calculating the Frobenius norm between original and reconstructed gene expression matrix. As shown below, we observed that optimal IC rank ranged from 5-6 in all cases.

Figure 1 – Reconstruction Error analysis to determine optimal components

Each scatterplot corresponds to a specific cell type shown at the top of the plot. X-axis spans the ICA factorizations starting from rank = 1 up to rank = 20, and Y-axis is the reconstruction error computed as Frobenius norm between original gene expression matrix and reconstructed gene expression matrix using ICA factorizations of ranks ranging from 1 to 20. The solid black lines are the lines of best fit obtained by splitting the data into two parts in such a way that sum of residuals is minimized.

Human protein atlas, where 81 cell types were clustered into 557 clusters

(<https://www.proteinatlas.org/humanproteome/single+cell/single+cell+type>), i.e., ~7 cell states per cell type. Therefore, we decided to extract 10 components for each cell type, as a reasonable upper bound to capture the underlying cell state compositions while

accommodating for a potentially higher number of cell states. We have now included this analysis in Supplementary Figure S1 and mentioned in methods section at line 1013.

3. While the GitHub repository (<https://github.com/hannenhalli-lab/CSI-TME>) provides several R scripts, a tutorial (either as a detailed README file or through external documentation) for introducing the usage of CSI-TME is required, which could facilitate broader adoption of the computational pipeline.

Response – We agree that regrettably the previously published version of the GitHub repository was not very user friendly. We have now streamlined the workflow to run CSI-TME and provided clear instructions to identify statistically significant and cross-validated lists of CSIs. There are two main modules: (1) Train CSI-TME on a new cohort (2) Find active interaction in a given sample based on pre-existing models. Each module can be run by typing 3-4 shell commands. Detailed tutorial is now provided at <https://github.com/hannenhalli-lab/CSI-TME>.

4. All figures in the manuscript require careful revision, including the size, font format, labels, etc. For example, Figure 1A contains much more information than Figure 1B, so the size of Figure 1A should be larger than Figure 1B for more clear visualization. For Figure 2F and 2G, implementing different colors to the bars or boxes facilitates distinguishing the anti-tumor and pro-tumor groups. Please ensure sufficient spacing between cell type labels and network nodes to enhance graphical clarity in Figure 2H and 2I.

Response – We appreciate these comments and have made the suggested changes in the updated figures

Minor Comments:

1. Do CCGA and CGGA refer to two different cohorts? Or is "CCGA" just a spelling error?

We thank the reviewer for pointing this out and regret the oversight which is now rectified.

2. As described in the section of "Methods", 10 independent components are extracted for each cell type. So the shape of patient \subseteq IC matrix of different cell state should be same in Figure 1A. In addition, the symbol " \approx " seems to be inappropriate.

We agree. We have now updated the figure with same dimension for cell state matrices as well as replaced the symbol \approx with an arrow

3. The last sentence in the Page 3 is confusing. Do three co-activity bins include bin 3, bin 7 and bin 9? But the bin 7 and 9 are with both high and low activity. And does S1 (S2) refer to IC1 (S2) in Figure 1B. If so, high activity of S1 and low activity of S2 corresponds to bin 7 rather than bin 3. The symbol "... " should be replaced by "Med" in Figure 1B.

We appreciate these points from the reviewer. To answer the reviewer's question on activity bins, we considered bin 1, 3, 7, and 9 separately. Bin 1 corresponds to a simultaneously low, and bin 9 corresponds to a simultaneously high activity of the two IC pairs. Bin 3 and 7 corresponds to the high activity of one of the IC and low for the other IC. In this scenario, If we keep IC1 along x-axis, and IC2 along y-axis, then low IC1 and high IC2 will correspond to Bin 7. On the other hand, high IC1 and low IC2 will correspond to bin 3. However, since we treat both orderings of each IC pair, considering only Bin 3 does cover both scenarios.

Additionally, we have replaced the sign “...” with “med” in Figure 1B.

4. In Figure 2A, does the pink point represent unknown cell type? If so, this cell type should be annotated due to its relative large cell number. Please provide the cell count for each identified cell type.

We thank the reviewer for raising this point. Indeed, as per our analysis, we could not identify any reliable markers to annotate that cell type despite the large number of cells. These seven cell types are consistent with a previous study that investigated the publicly available part of the single cell data used in our study(<https://www.nature.com/articles/s41467-022-28372-y>). There were unannotated cell types even in that original published study. This could be due to several reasons, including the lack of comprehensive reference atlases, incomplete knowledge of marker genes, and the presence of transitional cell states that are difficult to classify or due to technical issues (PMC6582955, 30617341). To ensure robustness, we decided to use only the cell types that could be annotated unambiguously. We have now indicated this at line 983. We have now added the cell counts per cell type in Supplementary table S1.

5. The meaning of color bar in the heatmap should be clarified in Figure S1A, S2A, S2D, S4C, S4E, 5C, and 7B.

We thank the reviewer for raising this point. We have now clarified the meaning of color bars in figure legends.

6. In Figure 2C, the cross-IC correlations between TCGA and CGGA for each cell type are all closed to 0.8 and nearly same. Is it reasonable?

We thank the reviewer for this comment. The correlations for 7 cell types in this study shown in figure 2C range from 0.85 to 0.95, highlighting that prognostic effect of ICs discovered in TCGA is highly reproducible. For clarity, below we have included the scatter plots for cross-IC correlation between TCGA and CGGA for each cell type, ruling out any statistical anomaly.

7. P value should be listed in Figure S1D.

We thank the reviewer for pointing this out. We have now added a p-value to that figure, which is now S2D in the current draft

8. Figure S1J (for positive signature genes) is the same as Figure S1K (for negative signature genes). For each cell type, all 10 ICs should be plotted rather than selecting a subset. Please check it carefully.

We apologize and thank the reviewer for pointing out this mistake. We have now corrected this. In figure S1J/K (now S2J/K), we intend to plot all those ICs which have at least 3 marker

genes distributed on the same chromosome. Some ICs did not meet this criterion and hence were excluded. But we have now expanded the figure to include all the ICs with some plots having missing entries. This is now indicated in the figure legends wherever relevant and highlighted in red color.

9. The authors mentioned that only the IC7 negative signature genes had higher expression during the embryonic phases of brain development and significantly decreased in fetal brain and mature adults. But as shown in Figure S2B, the negative signature genes of IC3 also exhibited this trend, albeit less significant than IC7. Besides, IC4 rather than IC7 (has been shown in Figure 3D) should be plotted in Figure S2B.

Response – We appreciate this point. However, it is worth noting that significance levels of IC.7 are far greater than IC.3. But we do not contend that IC.3 could also be relevant to some aspects of brain development. Notably, negative signature genes of IC3 also included multiple stemness related genes including NKX3-1, Esrrb/Nr5a2, and Tfcpl1 (doi:10.1038/s41556-018-0136-x, doi:10.1242/dev.199604) as well as factors critical for cancer stemness including IL-8 and miR-221 (doi:10.18632/oncotarget.5979, doi:10.2147/OTT.S161760. We have pointed this at line 386.

Secondly, the reason IC.4 is not plotted is because it did not have at least 3 signature genes. But, in response to the reviewer's comment, we have made this change with a blank plot for IC.4 and have indicated the same in the figure legend. This change is present in figure S3B (previously S2B) and we have indicated the same in the legend

10. Figure S2C requires clarification. Please expand the figure legend with detailed descriptions to enhance interpretability.

We thank and agree with the reviewers. We have now expanded the figure legend, which is now S3C, at line 471 to clarify this.

11. The rank of IC (or Dim) should be unified (Figure 2D, S1J, S1K, 3A, 3C, 3F, S2A, S2D, S2F, 5F, 5G, S3D, and S5A). The form of "IC.1 IC.2 IC.3 IC.4 IC.5 IC.6 IC.7 IC.8 IC.9 IC.10" is suggested.

We agree with the reviewer. We have now ordered of ICs as suggested. Some key points -

- We rectified the IC ordering in 2D, S1J (now S2J), S1K (now S2K), 3A, 3C
- Ordering was already consistent in 3F, S2F (now S3F), 5F, 5G, S3D (S4D)
- We could not modify the ordering in S2A, S2D (Now S3A, D) because the enriched GO terms were clustered to visualize the similarity across ICs. We have now added the dendrogram at the top and explained it in the figure legend
- in S5A (now S6A), we replaced "Dim" with "IC" to indicate independent components, for each cell type, those ICs are aligned in an increasing order from bottom to top. But we could not add missing ICs as only the ICs with at least one association are shown, as indicated in the legend.

12. In Figure 3E, only IC7 of malignant cells are mentioned in the manuscript to interact with immune ICs. How about IC5 and IC6?

We would first like to clarify that in figure 3E, we had shown the interactions of IC5 and IC6 as well. Specifically, we observed that IC5, which represents the OPC-like lineage interacted with IC4 and IC8 of B cells. Likewise, IC6, which represents Astrocyte-like lineage, also interact with immune cells, including IC5 and IC3 of T cells and IC8 of B cells. We have now added additional text at line 434 highlighting these cases.

13. It is interesting to observe that the negative signature genes of T cells IC1 significantly overlapped with the markers of senescent fibroblasts (Page 12). Is it caused by the substantial overlap between T cell and fibroblast senescence markers? If not, does it suggest that T cells can induce fibroblast senescence?

This is an interesting point, prompting us to look deeper. First, currently the most widely used cell type to study senescence is fibroblast. Furthermore, the core of oncogenic hallmarks such as replication, proliferation, etc., are believed to be conserved across cell types, and indeed the signatures for these hallmarks, as provided in databases such as cancerSea, are cell type-agnostic. More specifically for senescence, indeed there is strong evidence that fibroblasts and T cells share core senescence hallmarks—irreversible arrest via DDR→p53–p21 and p16INK4A–Rb, NF-κB–centered inflammatory programs with overlapping SASP cytokines, and broad chromatin remodeling—while mitochondrial dysfunction/ROS and the mitochondria→CCF→cGAS–STING axis are mechanistically established in fibroblasts and plausibly conserved in T cells but not yet directly validated in primary human T cells. For our purposes we therefore used fibroblasts senescence signatures as a proxy for the T cell senescence. Also, currently there is no strong evidence to suggest that T cells induce fibroblast senescence. We have now added this additional discussion at line 1113.

14. Fisher's exact test is used many times in the manuscript. Do both of "Fisher test" and "Fisher's test" refer to Fisher's exact test? Please use the consistent terminology throughout the manuscript.

We are sorry for the confusion and appreciate this feedback. We now consistently use the phrase 'Fisher's exact'.

15. It is counterintuitive to observe that CSIs dominant in IDH-A tumors are predominantly anti-tumor. The authors should provide further explanation about this point.

We thank the reviewer for highlighting this issue that we find intriguing as well. Upon further research we found that the average age of onset for IDH-A is much lower than that for IDH-O tumors (mentioned in <https://doi.org/10.1038/s41598-023-32153-y>, <https://pubmed.ncbi.nlm.nih.gov/35083156/>). We confirmed this in the TCGA cohort, as shown below.

We further tested if the enrichment of anti-tumor interactions among the IDH-A tumors is a result of their younger age by testing the association between anti-tumor interaction load and IDH subtype (i.e. IDH-A vs IDH-O), while directly controlling for age in a logistic regression model:

Antitumor load ~ IDH_Class + age

Despite controlling for age, enrichment of anti-tumor interactions among IDH-A tumor remained. This observation is worthy of future investigation. We have now reported this additional observation and acknowledgement of this observation at line 505.

16. The authors claimed that CSIs can explain the survival of IDH-A and IDH-O patients which cannot be explained by co-deletion of 1p/10q chromosomal arms. Do these included patients all harbor this co-deletion? If not, stratifying the patients based on deletion status and conducting comparative survival analyses are required.

We appreciate this comment. Upon further inspection, we found that all the IDH-O tumors were co-deleted for chromosomal arm 1p/10q and none of the IDH-A were. Therefore, the further split of IDH-A and IDH-O based on CSIs is not confounded by the differential co-deletion status of the subgroups.

17. Figure 5A is also shown in the Figure S3B. To avoid redundancy, do not show the same figure repeatedly.

Thank you for this suggestion. We have now made this change.

18. Using spatial transcriptomic data, the authors observed that negative signature genes of endothelial IC-3 and positive signature genes of the malignant IC-2 were significantly proximal in the Visium slides. Appropriate statistical testing should be performed to quantify the significance rather than relying solely on visual assessment.

Response - We appreciate this comment. We would like to highlight that we used an extensive statistical framework to detect spatial co-localization of ICs. This is described in the methods section with a heading "Detecting co-localized cell state interactions in spatial transcriptomics data". Briefly, to determine whether two cell states X and Y were spatially co-localized in a slide, we computed a neighborhood overlap statistic that was defined as:

$$O(X, Y) = \frac{|U\{Nbds \text{ where } X \text{ and } Y \text{ are both active}\}|}{|\{Nbds \text{ where } X \text{ is active}\} \cup \{Nbds \text{ where } Y \text{ is active}\}|}$$

where each spot's neighborhood (abbreviated as Nbds) consisted of its six nearest neighbors (except for spots on the tissue boundary) and the threshold to determine if X and Y were active was determined as above. We computed the significance of the observed co-localization by computing $O(X, Y)$ after shuffling the assignments of cell state activity scores across each slide a hundred times and fitting a normal distribution to the obtained overlap scores. The one-sided p-value was calculated as the probability that the observed $O(X, Y)$ was greater than the mean background $O(X, Y)$ estimated via shuffling. The co-localization of all possible cell state interactions was computed separately in each slide and their p-values were adjusted for multiple testing using the Benjamini-Hochberg method.

19. Are "mutated genes" and "mutation burden" conceptually same in Figure 7A and the figure

legend?

We thank the reviewer for this comment and apologize for the confusion. Yes indeed, those are the same things. We have now defined the mutation burden as the number of mutated genes in the figure legend of 7A.

Reviewer #3:

Summary:

Singh et al. present CSI-TME, a computational pipeline that infers cell state interactions in IDH-mutant gliomas from bulk RNA sequencing data. To accomplish this, they make use of existing bulk RNA sequencing data from TCGA and CCGA, deconvolving it to produce gene expression matrices for each cell type. On these matrices, they perform individual component analysis (ICA), which are considered to represent latent transcriptional programs, or cell states. Some of the ICs are reported to correspond to known gene expression programs, such as T cell proliferation or the glioma stem cell state. They then identify cell states pairs which have some degree of joint activity, which are deemed to be cell state interactions (CSI). From these CSIs, a network is constructed, the cell state interaction network (CSIN). The authors then assess the extent to which different CSIs correspond to patient outcome, and each CSI is labelled pro-tumor or anti-tumor. Having annotated their CSIN, they show that patient outcome corresponds with the relative abundance of pro- or anti-tumor interactions. The authors refine their analysis by examining CSIs that have enriched ligand-receptor pairs, finding that interaction strength in the CSIN correlates with the abundance of ligand-receptor pairs. Finally, cell states are linked to somatic mutations, and Singh et al. suggest that somatic mutation may to some extent drive cell state formation.

We appreciate and thank the reviewer for providing feedback on our work. We would first like to clarify a potential misunderstanding - the CSIN is identified by directly modeling the patient survival based on the joint activity of the cell state pairs, while controlling for the individual cell states. In particular, and this may be the point of misunderstanding, we did not assess if the cell state pairs have significant joint activity, but rather whether the joint activity is associated with prognosis. So, for instance, two states may not have significant co-occurrence but when they do co-occur those patients have a distinct survival profile – such state pairs will be detected by our approach. This is also reflected in the reviewers' major comment #3 where we have again clarified this issue. We regret the misunderstanding and have now provided additional clarification defining cell state interactions at line 132.

Below we provide a detailed response to the each of reviewers' comments

General remarks:

This work builds on the already well-developed deconvolution methods available for bulk sequencing data by producing a pipeline to gain deeper insight into deconvolved gene expression data and cell state resolution. As described below, the authors do not validate their

cell state resolution deconvolution before using these data to perform ICA. Since this is primarily a methods paper aimed at enabling deeper interpretation of widely available bulk data, the introduction of cell state-level deconvolution is a potentially valuable advancement. Yet without validation, readers have no way to assess its accuracy.

Response – This is indeed an important concern. However, we would like to note that we did not perform ‘cell-state level’ deconvolution *per se*. Instead, as the reviewer correctly mentioned in their **Summary** paragraph, we used CODEFACS to get ‘cell type’ specific gene expression profiles across samples, and only in a subsequent step, we performed Independent Component Analysis to get the latent factor representation of the cell type specific gene expression matrices, for each cell type separately.

Now coming to the main concern raised by the reviewer here – in the previous version of manuscript, we had performed three different validations – (1) Clustering of individual cell states (independent components) in single cell PCA space (2) Consistency between the prognostic effect of cell states in TCGA and CGGA. (3) Mapping the cell states to previously known cell states in GBM.

However, we appreciate the need for additional robustness analysis of the inferred ICs. Toward this, we have now assessed whether the ICs can be robustly recovered in the bootstrapped data. If ICs were merely statistical noise, we should not detect those in the repeated bootstraps. For each cell type, we performed the IC detection in the 10 independent sets of 70% bootstrapped data and calculated the correlation between each IC in the full data and each IC in the bootstrapped data. Very encouragingly, we observed that each IC from full data was highly and uniquely correlated with a unique IC in the bootstrapped data (PCC > 0.9 in all cases). This implies that across 10 different bootstraps, each IC was robustly captured even if we used 70% of the full data, thus ruling out the statistical noise.

Additionally, we performed a more stringent test as follows. Instead of projecting the deconvolved cell types from CGGA onto the latent space of TCGA, we independently inferred ICs in CGGA and performed the same analysis as above, i.e., we assessed the extent to which a TCGA IC is robustly and uniquely recapitulated independently in CGGA. Despite all the technical and genetic differences between Chinese and American population, we again observed that TCGA ICs were uniquely and robustly captured in the CGGA data.

Lastly, we note that ICs derived from TCGA have significantly and strongly consistent effect on patient survival in TCGA as well as in CGGA as shown in figure 2C, highlighting the robustness of signals across cohorts. To further ascertain that ICs are not mere statistical noise, we performed an additional analysis. Under the assertion that ICs are mere statistical noise, the same level of concordance should also be observed if we fitted the IC models in randomized gene expression data. Therefore, we randomly permuted the deconvolved TCGA gene expression data and repeated the entire procedure. As in figure S2E, we computed the hazard ratios of the randomized ICs in TCGA and compared it against the hazard ratios of the projected ICs in the CGGA data. Interestingly, we observed that across 10 iterations, for each cell type, the median correlation in the randomized data was ~0 in most cases. These results indicate that the observed ICs are not mere statistical noise and described at line 210

So, taken together

- To deconvolve cell type-specific gene expression in each sample, we used CODEFACS, which has been extensively validated in the original publication. We then used the deconvolved cell type-specific gene expression to infer ICs in each cell type.
- CODEFACS reliably infers cell type specific expression of several thousand genes on average across cell types, which are sufficient to infer cell states, as demonstrated in scRNA-seq datasets.
- 70% of the ICs are clustered in PCA space suggesting that they represent transcriptional states
- we could map many of the ICs to previously annotated cell states of gene expression programs, which is expected to be incomplete
- ICs have consistent effect on survival in TCGA and CGGA (not observed for ICs derived from permuted gene expression data)
- ICs can be robustly detected in bootstrapped TCGA data and independently in CGGA data

These observations provide a strong support for the biological meaningfulness of the ICs. We have now indicated this additional analysis at lines 210, 240, and 251.

Moving to the second step, the CSI, I also have concerns about the signal/noise ratio when performing additional transformations on top of deconvolution, which is already a somewhat noisy process. Specifically, the discrepancy in the results between the two cohorts, TCGA and CCGA, is concerning. Therefore, biological findings from this approach need to be taken with a grain of salt due to the level of abstraction from the raw sequencing data and should be followed up with supportive evidence or validation.

Thus, while the ideas are novel, the methods are neither well-established nor thoroughly validated. If the methods or the biological insights from CSI-TME are robust and reproducible, which has not been confirmed (see below), this tool could be of use to any researcher who has

abundant bulk sequencing data but lacks access to single cell data, hopefully in many cancer types. With that said, using single cell data to infer cell state interactions would likely be preferred in most cases.

Response – We appreciate this comment. The reviewer refers to two key points – (i) the discrepancy between TCGA and CGGA and (ii) and why not use single-cell data. We provide our response to these comments below

- (i) Discrepancy – It must be noted that TCGA and CGGA are glioma cohorts from two different populations, which not only differ in terms of technical protocols but also in their genetics. To put this into perspective, we computed gene-gene cross-sample correlations independently in TCGA and CGGA and assessed the correlation of these gene-pair correlations between TCGA and CGGA. As expected, the correlation, though statistically significant, is far from perfect ($R = 0.62$).

This highlights that perfect concordance cannot be expected across such technically and genetically distinct cohorts. Despite these differences, we observed a significant concordance in cell state interactions (Figure 2G), i.e., the TCGA anti-tumor interactions are predominately anti-tumor in CGGA (and likewise for pro-tumor interactions). We believe that contrary to the claim of discrepancy, figure 2G points to a significant and substantial concordance.

- (ii) Secondly, CSIs reported by CSI-TME, by design, have prognostic value. As we have highlighted by the italicized text in the introduction at line 87, the current single cell cohorts are (a) small, typically few tens of patients, and (b) lack clinical data. CSI-TME directly models the clinical data, which is not possible with single cell cohorts, thus providing a unique advantage.

Importantly, to further boost our confidence in CSI-TME, we have now applied it to three more TCGA cancer types including BRCA, SKCM, and HNSC. As before, we performed the CODEFACS deconvolution, followed by ICA, followed by modeling of prognostic cell state

combinations using Cox regression. In all three cases, we observed that CSI-TME was able to identify pro- and anti-tumor CSIs that were cross validated using 70% accuracy in 10 bootstraps. Consistent with our findings in IDH-mut, in each case, there were far greater pro-tumor interactions in the TME, and importantly, the CSIs were associated with resistance to various targeted and immunotherapies in a consistent manner. Most importantly, we observed that CSIs detected in TCGA-BRCA were associated with the progression of breast pre-cancerous lesions. These results, support general applicability of CSI-TME beyond IDH mutant glioma in identifying clinically relevant pro- and anti-tumor cell state interactions. These results are now shown by adding a new section at line 786.

Major points:

1. The introduction requires a thorough review and revision. The most concerning citation is found in the first paragraph of the introduction, the authors state that gliomas recur in just 50% of cases, citing 2019 paper from Miller et al (2019) (1). This is not correct, as even low-grade gliomas almost always recur (2,3). It appears that the cited paper looks at recurrence over about only about 6 years, giving a recurrence rate of 54% for that timespan. However, over longer timelines recurrence is much more likely. This needs to be corrected as it is quite misleading.

Response – We thank the reviewer for their feedback and apologize for the oversight. We now revised the introduction after consulting with an additional expert in IDH-mut gliomas. In particular, we have corrected the 6-year recurrence rate to 54% as well as updated the text to reflect the life-time recurrence rate following initial diagnosis and treatment. These changes are now incorporated and highlighted in the introduction. We have rewritten the first paragraph of introduction to address these issues.

2. Any utility of the CSI-TME pipeline is highly dependent on the validity of the deconvolution used. However, the authors do not spend time validating the deconvolution itself. The ICA-derived cell states are annotated without a defined or standardized method, leaving the reader uncertain whether these states accurately reflect the known spectrum of cell states in IDH-mutant gliomas. This is particularly concerning given that the cellular composition of IDH-mutant gliomas has been extensively characterized, and expected profiles are well established. Downstream analyses are suspect without such validation.

Response – We appreciate this comment. This comment is essentially the same as first general remark by the reviewer, i.e., lack of validation for cell states. As we have shown in response to the first general remark

- To deconvolve cell type-specific gene expression in each sample, we used CODEFACS, which has been extensively validated in the original publication. We then used the deconvolved cell type-specific gene expression to infer ICs in each cell type.
- CODEFACS reliably infers cell type specific expression of several thousand genes on average across cell types, which are sufficient to infer cell states, as demonstrated in scRNA-seq datasets.

- 70% of the ICs are clustered in PCA space suggesting that they represent transcriptional states
- we could map many of the ICs to previously annotated cell states of gene expression programs, which is expected to be incomplete
- ICs have consistent effect on survival in TCGA and CGGA (not observed for ICs derived from permuted gene expression data)
- ICs can be robustly detected in bootstrapped TCGA data and independently in CGGA data

These observations provide a strong support for the biological meaningfulness of the ICs. We have now indicated this additional analysis at lines 210, 240, and 251.

With regards to annotation of ICA-derived cell states, previous research

(<https://www.nature.com/articles/nature20123>,

<https://www.science.org/doi/full/10.1126/science.aai8478>), identified three primary cell states in IDH-mut gliomas including astrocytic-like, OPC-like, and stem-like lineages. As we show in Fig 3A, three of our inferred ICs unambiguously map to these three cell states. Additionally, some ICs also mapped to recurrent cell states (<https://www.nature.com/articles/s41588-022-01141-9>) that resembles hypoxic or EMT-like cell states (Figure 5F). With the availability of more resolved cell state catalogs, it may be possible to map the rest of the inferred ICs to specific state.

3. Cell state interactions are inferred from co-occurrence co-absence, or mixed presence of IC enrichment between cell types (Fig. 1B, bins 1, 3, 7, 9). However, the authors do not support the inferred indirect CSIs, though this is the major innovation of the paper. It is surely possible that the presence or absence of cell states can co-occur without meaningful biological interaction, whether direct or indirect. How is it known that cell state co-occurrence (or other combinations) are not merely coincidental? Co-occurrence does not necessarily imply functional relation. Much of the paper from figure 2 onward is dependent on this assumption. The authors assess only the direct interactions by showing that cell state interactions do correlate with the abundance of ligand-receptor interactions (Fig. 5A). However, there are already tools to detect ligand-receptor interactions such as LIANA or CellphoneDB (4,5), and they rely on co-expression of the two populations, so this part is already well established and developed in the field. Thus, one of the novelties of this manuscript, the purported ability to detect indirect interactions, which cannot be done with ligand-receptor analysis, is not supported.

Response – We appreciate this concern. This concern is also raised in the overall summary provided by the reviewer. As we mention above as a clarification, CSI-TME objective is not to detect co-occurrence of cell state pairs. Instead, we detect the cell state pairs that, when they co-occur, have significant effect on the patients' survival, over and beyond what could be explained by the activity of individual cell states alone. For this, we specifically model the overall survival of cancer patients to learn the interaction terms between candidate cell state pairs. The final CSIN is constructed by using an FDR < 0.20 and ensuring at least 70% cross

validation accuracy in bootstraps. Therefore, the concern of statistical noise and lack of functional impact are explicitly addressed in CSI-TME.

Moreover, as we have shown and now extended, the interactions detected by CSI-TME are significantly associated with response to therapies in multiple cancer types. Importantly, for breast cancer, we observed that detected CSIs were significantly more active in the pre-malignant lesions that progressed to full-fledged tumors as compared to the ones that did not progress, thus highlighting their general clinical relevance, as opposed to mere coincidence. Lastly the comment about LIANA and CELLPHONEDB, regrettably stems from ambiguous interpretation of the term 'interactions', as mentioned above in summary response. As the reviewer mentions, indeed while both these methods detect highly co-occurring (or co-expressed) L,R pairs as a means to identify interaction. In contrast, CSI-TME does not identify highly co-occurring cell state pairs, but rather, it identifies cell state pairs, which when co-occur, associate with prognosis. The reliance on clinical outcome to identify 'interactions' sets apart CSI-TME from these other methods, which do not rely on clinical outcome and focus solely on statistical co-occurrence of ligand and the receptor, based on prior knowledge of ligand-receptor pairs. Another point of contrast is that these other tools work on scRNA-seq data; while cell state interactions can be detected based on scRNA-seq cohorts, large enough cohorts with matched clinical data are currently not available and therefore CSI-TME relies on deconvolution to exploit the vast cohorts of clinical bulk RNA-seq datasets (also explained below in response to comment #6).

4. The authors show dozens of cell state interactions yet explore only a few of them (e.g. putative GSCs with B and T cells). Some selected CSIs do correlate with patient outcome (e.g. interferon response gene in T cells with PTN and FBXO5 in malignant cells), but overall, how many of these are biologically meaningful? I also noticed that in describing Figure 4, the authors mention that only 103/160 (~ 62%) interactions have the same direction of hazard in TCGA and CGGA. This does not inspire confidence in the overall robustness of CSI-TME.

Response - We thank the reviewer for these comments. We reiterate that all the CSIs that we identify in this study, by design, correlated with patient outcome. The issue of reproducibility is brought up again, and we would like to reiterate that TCGA and CGGA are glioma cohorts from two different population, which not only differ in terms of technical protocols but also in their genetics. To put this into perspective, we computed gene-gene cross-sample correlations independently in TCGA and CGGA, and assessed the correlation of these gene-pair correlations between TCGA and CGGA as we explained above. As expected, the correlation, though statistically significant, is far from perfect ($R = 0.62$). This highlights that perfect concordance cannot be expected across such technically and genetically distinct cohorts. Despite these differences at the gene correlation level, we observed a significant concordance in cell state interactions (Figure 2G), i.e., the TCGA anti-tumor interactions are predominately anti-tumor CGGA (and likewise for pro-tumor interactions). We believe that contrary to the claim of discrepancy, figure 2G points to a significant and substantial concordance. Further, when we computed the cell state interactions using the permuted gene expression data, we did not observe any consistency between TCGA and CGGA as we had shown in figure S3H. Together, these results suggest that despite the technical issues, we can successfully mine prognostic cell state combinations that are reproducible to a significant extent.

5. Some interactions conflict with biological expectations. In principle, challenging conventional wisdom is a benefit. However, in the absence of further validation and characterization, this starts to look like noise. For example, is it believable that a top interaction is between B cells and malignant cells? Fig. 3G is the interaction plot for cell states in T cells and malignant cells. It looks like there is only one anti-tumor interaction. It is surely expected that many T cell-malignant cell interactions would be detrimental, but that all but one interaction is pro-tumor seems dubious given the critical role of T cells in anti-tumor immunity. In Fig. 6B, it looks like there are more pro- and anti-tumor interactions in recurrent tumors. This is at odds with expectation. Overall, figures 6B and C are probably not interpretable with only 3 CSIs. These findings could be believable with elaboration and/or validation, but otherwise this looks like noise or general unreliability of the pipeline.

We appreciate this feedback from the reviewer. This comment has multiple parts and we quote and address those below

- “is it believable that a top interaction is between B cells and malignant cells”

We would like to again reiterate that in CSI-TME, the term interaction implies a significant prognostic effect based on joint activity that cannot be explained by the activity of individual cell type. While we agree that the current research in the TME of IDH-mut glioma has not focused on B cells, we *did not* claim that any of these interactions constitute a top interaction *per se*. The main reason for their neglected role could be their low abundance in the TME of IDH-mut glioma which makes it difficult to study experimentally. Nevertheless, low abundance does not rule out their functional significance (<https://pmc.ncbi.nlm.nih.gov/articles/PMC8545794/>, <https://www.nature.com/articles/s41577-021-00652-6>). In our work, we detected a total of 17 interactions among B cell and malignant cell states. 7 of these are anti-tumor and 10 are pro-tumor. Past research indeed supports the pro- and anti-tumor roles of B cells in multiple cancers including, including a different but related disease IDH-wt glioma . For example, B cell signatures were associated with worsened patients’ outcomes (<https://doi.org/10.1016/j.ccell.2023.02.017>). In contrast, a mouse study showed that CD40-activated 4-1BB+ B cells were associated with enhanced antigen presentation and improved patient survival (<https://pubmed.ncbi.nlm.nih.gov/32991668/>). Also, B-cell based therapies are newly emerging treatment modalities in glioma including IDH-mutant cases (<https://www.jci.org/articles/view/177384>). Thus, although typically neglected due to their low abundance or lack of representation in single cell datasets, B cells can have prognostic and clinically relevant role in the TME. We have now added additional discussion addressing this at line 911 .

- “Fig. 3G is the interaction plot for cell states in T cells and malignant cells. It looks like there is only one anti-tumor interaction. It is surely expected that many T cell-malignant cell interactions would be detrimental, but that all but one interaction is pro-tumor ”
We would like to note that figure 3G is *not for all* interactions between T cell and malignant cell ICs. Instead, figure 3G represent the interactions between all the

signature genes of a specific T cell state and malignant cell state IC pair, i.e. T cell IC.7 and Malignant cell IC.7, which was identified as a pro-tumor CSI. The purpose of this plot is to identify and investigate the specific interesting gene pairs that might underlie cell state interactions. Overall, we see that as expected, most gene pairs indeed exhibit a pro-tumor interaction, consistent with the pro-tumor interactions of the cell states. For example, in this case, we observed that bona fide interferon response genes IFIT1 and IFIT3 in T cells exhibited a significant pro-tumor interaction with PTN and FBXO5 genes in malignant cells, respectively. In particular, interaction between IFIT1 gene in T cells and PTN gene in malignant cells, previously shown to have potential immune-regulatory function (Sorrelle, Dominguez, and Brekken 2017), implies that downregulation of PTN in malignant cells may suppress interferon signaling in T cells and the associated cytotoxic response. We are sorry for this confusion and have added additional text explaining this at line 456.

- “In Fig. 6B, it looks like there are more pro- and anti-tumor interactions in recurrent tumors. This is at odds with expectation. Overall, figures 6B and C are probably not interpretable with only 3 CSIs. These findings could be believable with elaboration and/or validation, but otherwise this looks like noise or general unreliability of the pipeline”

Higher penetrance of pro-tumor interactions in recurrent tumors is expected. However, a higher penetrance of anti-tumor interactions in recurrent tumors is indeed puzzling and difficult to interpret. We note that the trend for anti-tumor CSIs in Fig 6B is not significant ($p = 0.4$), due to paucity of such CSIs. Furthermore, in addition to interaction penetrance, we also analyzed the overall load of pro- and anti-tumor interactions (i.e. number of interactions active in a given sample), where we essentially have the same number of datapoints (i.e. number of samples) for pro- and anti-tumor interactions (Figure S4B). In this alternative analysis, we again observed that recurrent tumors had clearly higher load of pro-tumor interactions as compared to primary cases, consistent with their higher aggressiveness. On the other hand, due to their paucity, the trends for anti-tumor CSIs remains inconclusive. We speculate that this lack of trend for anti-tumor CSIs could be partly due to the fact that anti-tumor interactions are likely to be a homeostatic response (as we argue in Figure 7) and may be active at similar levels in both primary and secondary tumors

6. In the introduction, it is asserted that cohorts of single cell RNA sequencing data for IDH-mutant gliomas are "lacking". While it is true that bulk datasets are more abundant, single cell datasets do exist for gliomas (6-8). The authors even use some of these datasets to annotate the ICA-derived cell states (though, as noted in Comment #2, this is done loosely and without a clearly defined methodology). However, they do not use these datasets to validate any of their results-despite the fact that interactions inferred from bulk data should also be detectable in single-cell data. While the available single-cell datasets may be too small for full exploration, they are sufficiently large for validation purposes.

Response – We appreciate this comment, however, there is still a dearth of large sufficiently powered single cell cohorts with matching clinical data. The reviewer cited three references; however, we note that citation Ochocka et al. contains the data from mouse models (not humans), Tirosh et al. contains data from only six samples and Venteicher et al. has data from only 16 samples. This hinders the modeling of clinical outcomes based on the activity of cell state pairs based on a couple of dozens of data points at the most.

Also, we have used the findings and data from Tirosh et al. and Venteicher et al. to validate and interpret our findings from CSI-TME. This is mentioned at line 1123 in methods section and line 368

Furthermore, as we show now, CSI-TME is applicable not just to IDH-mut glioma, but to several other cancer types (line 786).

7. In the methods section, the authors state that the decision to use 10 individual components was arbitrary. However, it is not obvious that all 10 ICs are biologically meaningful for all cell types. Nor is it obvious that all important ICs have been investigated. The authors should show that the ICs they used capture meaningful variation and that additional ICs do not capture important additional information.

Response – We appreciate this point. Unlike PCA, where each successive component explains a decreasing amount of variance of the data, ICA currently lacks an agreed-upon standard approach for estimating the optimal number of ICs. There are two prominent approaches published for IC ,– MSTD and optICA, that we had tried. However, results from two approaches did not converge. For example, in the case of stromal cells, as shown below, MSTD criteria estimated > 40 optimal components while optICA estimated just 8.

For the purposes of CSI-TME, we chose 10 ICs based on the analysis of reconstruction error through various iterations of ICs with rank 1 up to 20. We used segmental curve fitting to approximate the IC rank beyond which there was no significant gain in the model accuracy (reconstruction error). The accuracy was ascertained by calculating the Frobenius norm between original and reconstructed gene expression matrix. As shown below, we observed that optimal IC rank ranged from 5-6 in all cases.

Figure 2 – Reconstruction Error analysis to determine optimal components

Each scatterplot corresponds to a specific cell type shown at the top of the plot. X-axis spans the ICA factorizations starting from rank = 1 up to rank = 20, and Y-axis is the reconstruction error computed as Frobenius norm between original gene expression matrix and reconstructed gene expression matrix using ICA factorizations of ranks ranging from 1 to 20. The solid black lines are the lines of best fit obtained by splitting the data into two parts in such a way that sum of residuals is minimized.

Human protein atlas, where 81 cell types were clustered into 557 clusters

(<https://www.proteinatlas.org/humanproteome/single+cell/single+cell+type>), i.e., ~7 cell states per cell type. Therefore, we decided to extract 10 components for each cell type, as a reasonable upper bound to capture the underlying cell state compositions while accommodating for a potentially higher number of cell states. We have now included this analysis in Supplementary Figure S1 and mentioned in methods section at line 1013.

Minor points

1. In the introduction, in the third paragraph, the Singh et al. say that tumor and microenvironmental cells can exist in multiple transcriptional cell states. This is true, but two of the three citations reference studies in IDH-WT glioblastoma (Nefitel et al. 2019 and Batchu et al. 2023), which is not the disease they are reporting on. They should instead cite papers such as Tirosh et al. (2019) or Ochocka et al. (2021) which show this in IDH-mutant glioma (6,7).

Response – We thank the reviewers for pointing this out and agree that Nefitel et al. and Batchu et al. are focused on IDH-wt glioma, however, there is an additional citation, Abdelfattah et al., that is focused on IDH-mut glioma. We have already cited Tirosh et al. in our manuscript when discussing our findings on glioma stem cells. However, we have now moved the Tirosh et al. to the introduction and have replaced the references to Nefitel et al. and Batchu et al. with

Ochocka et al.

2. Figure 3G is difficult to read because the legends use the same colors but are not labelled as to which represents nodes and which represents edges.

We thank the reviewer for pointing this out. We have now used a different color scheme clearly indicate the nodes and edges in figure 3G

3. Fig 4J is difficult to interpret with so few patients in the interaction group.

We agree that there are just 5 patients in IDH-O Antitumor category in figure 4J. However, it is notable that despite the small number of patients in both TCGA (Figure 4G) and CGGA (Figure 4J), the trend stands out very clear and consistent in both cohorts. Therefore, although active in small number of patients, IDH-O patients dominated by anti-tumor interactions survive better than IDH-O patients dominated by pro-tumor interactions in the TME. We have now explicitly stated this in the manuscript at line 527.

4. There are numerous typos which should be addressed.

We thank the reviewer and have made sure that manuscript is free from typos and errors.

References

1. Miller JJ, Loebel F, Juratli TA, et al. Accelerated progression of IDH mutant glioma after first recurrence. *Neuro-Oncol.* 2019;21(5):669-677. doi:10.1093/neuonc/noz016
2. Daniels TB, Brown PD, Felten SJ, et al. Validation of EORTC Prognostic Factors for Adults With Low-Grade Glioma: A Report Using Intergroup 86-72-51. *Int J Radiat Oncol.* 2011;81(1):218-224. doi:10.1016/j.ijrobp.2010.05.003
3. Stewart J, Sahgal A, Chan AKM, et al. Pattern of Recurrence of Glioblastoma Versus Grade 4 IDH-Mutant Astrocytoma Following Chemoradiation: A Retrospective Matched-Cohort Analysis. *Technol Cancer Res Treat.* 2022;21:15330338221109650. doi:10.1177/15330338221109650
4. Dimitrov D, Türei D, Garrido-Rodriguez M, et al. Comparison of methods and resources for cell-cell communication inference from single-cell RNA-Seq data. *Nat Commun.* 2022;13(1):3224. doi:10.1038/s41467-022-30755-0
5. Troulé K, Petryszak R, Cakir B, et al. CellPhoneDB v5: inferring cell-cell communication from single-cell multiomics data. *Nat Protoc.* Published online March 25, 2025:1-29. doi:10.1038/s41596-024-01137-1
6. Ochocka N, Segit P, Walentynowicz KA, et al. Single-cell RNA sequencing reveals functional heterogeneity of glioma-associated brain macrophages. *Nat Commun.* 2021;12(1):1151. doi:10.1038/s41467-021-21407-w
7. Tirosh I, Venteicher AS, Hebert C, et al. Single-cell RNA-seq supports a developmental hierarchy in human oligodendroglioma. *Nature.* 2016;539(7628):309-313.

doi:10.1038/nature20123

8. Venteicher AS, Tirosh I, Hebert C, et al. Decoupling genetics, lineages, and microenvironment in IDH-mutant gliomas by single-cell RNA-seq. *Science*. 2017;355(6332):eaai8478.

doi:10.1126/science.aai8478

7th Jan 2026

Manuscript Number: MSB-2025-13172RR-Q

Title: Identifying Prognostic Cell State Interactions in the Tumor Microenvironment of IDH-Mutant Gliomas

Author: Arashdeep Singh

Bharati Mehani

Vishaka Gopalan

Sushant Puri

Kenneth Aldape

Sridhar Hannenhalli

Dear Dr. Hannenhalli,

Thank you for submitting the revised version of your manuscript. We have now received feedback from the two reviewers. As outlined in their comments below, both reviewers agree that the manuscript has been substantially improved. However, they have also raised several relatively minor issues that need to be addressed before we can accept the manuscript for publication.

On a more editorial level:

1. Please remove all main and supplementary figures from the manuscript file and upload them as separate, production-quality figure files. Figure legends should be provided at the end of the manuscript. Supplementary figures should be renamed Expanded View (EV) Figures, which will be displayed as collapsible/expandable items online. EV Figures should be cited in the text as Figure EV1, Figure EV2, etc., and their corresponding legends should be included in the main manuscript immediately following the legends for the main figures.
2. Please include up to five keywords in the manuscript file.
3. Remove "Authors' contribution" section from the manuscript file.
4. The references need to be formatted according to the Molecular Systems Biology reference style. Please list up to 10 co-authors of a paper before adding et al. in the reference list. Citations should be listed in alphabetical order. DOI numbers should be removed for published articles.
5. The supplementary tables are complex and should be updated to Dataset EV1-Dataset EV3 throughout (including source file names, titles in the system, and callouts in the manuscript). Legends must be provided on a separate sheet/tab within each dataset file.
6. We updated our journal's competing interests policy in January 2022 and request authors to consider both actual and perceived competing interests. Please review the policy (<https://link.springer.com/partners/embo-press/editorial-policies#Competing%20interest%20disclosures>) and update your competing interests if necessary.

Please use the heading "Disclosure statement and competing interests".

7. Before submitting your revision, primary datasets and computer code produced in this study need to be deposited in an appropriate public database.

The accession numbers and database should be listed in a formal "Data Availability" section (placed after Materials & Method) that follows the model below (see also <https://link.springer.com/partners/embo-press/editorial-policies#Data%20availability%20statement>). Please note that the Data Availability Section is restricted to new primary data that are part of this study.

Data availability

Additional information on source data and instruction on how to label the files are available <

<https://link.springer.com/journal/44320/submission-guidelines#cms-Source-data> >

9. Please provide a "standfirst text" summarizing the study in one or two sentences (approximately 250 characters, including space), three to four "bullet points" highlighting the main findings and a " visual abstract" (550px width and 400-600 px height, PNG format) to highlight the paper on our homepage.

Please refer to published papers for examples.

10. Please download and fill our Reagents and Tools Table template (.docx), which you can find in our author guidelines: <https://link.springer.com/journal/44320/submission-guidelines#structuredmethods>.

11. Please address the following issues in figure legends:

- Please note that the figure 4I, J is mislabeled as figure 4F, G in the manuscript. This needs to be rectified.
- Please note that the exact p values are not provided in the legends of figures 3B, D; 4G, I; 5J, 6C, 7C; S2 B, C, D, E, I, J, L; S2D, K, L
- Please indicate the statistical test used for data analysis in the legends of figures 2G, 3B, D; 4G, H, I, J; 5H-J; 6C; S2 D
- Please note that the box plots need to be defined in terms of minima, maxima, centre, bounds of box and whiskers, and percentile in the legends of figures 2G, 3B, D; 4A, H; 6A, B; 7D, E; 8B, C, D, E, F; S2 B, C, D, E, I, J, K, L; S3B, E; S4A, S5A
- Please note that information related to n is missing in the legends of figures 2G, 3B, D; 4A, H; 6A, B; 7D, E; 8B, C, D, E, F; S3B, E; S4A, S5A

12. The co-author's email address bounced: Bharati Mehani (bharati.mehani@nih.gov

). Please either remove this author from the author list in the system and re-add her using a valid email address, or provide the updated email address so we can revise the author's account accordingly.

13. When you resubmit your manuscript, please download our CHECKLIST (<https://media.springernature.com/original/springer-cms/rest/v1/content/27825796/data/v1>) and include the completed form in your submission.

Please note that the Author Checklist will be published alongside the paper as part of the transparent process .

Kind regards,

Jingyi

Jingyi Hou, PhD
Senior Editor
Molecular Systems Biology

If you do choose to resubmit, please click on the link below to submit the revision online before 6th Feb 2026.

EMBO Press transparent editorial process initiative (see our Editorial at <https://dx.doi.org/10.1038/msb.2010.72> , Molecular Systems Biology will publish online a Review Process File to accompany accepted manuscripts. When preparing your letter of response, please be aware that in the event of acceptance, your cover letter/point-by-point document will be included as part of this File, which will be available to the scientific community. More information about this initiative is available in our Instructions to Authors. If you have any questions about this initiative, please contact the editorial office (msb@embo.org).

Reviewer #2:

The authors have addressed the majority of my previous concerns regarding the manuscript. However, several issues require

further clarification.

1. For "Major Comment 1", the authors claimed that they have performed cross-cancer comparison of the CSIs. However, I cannot locate the related results in the section titled "CSI-TME is broadly applicable to multiple cancer types in identifying clinically relevant cell state interactions". Additionally, what does "a median time of 7.3 years 11/18/2025 6:38:00 PM" (line 802-803) mean?
2. For "Minor Comment 2", has Figure 1A been updated? The figure in the revised version appears identical to the original one.
3. For "Minor Comment 3", I am still unclear about the definitions of Bin 3 and Bin 7. In Figure 1B, the x-axis represents IC2, and the y-axis stands for IC1. So why does low IC1 and high IC2 correspond to Bin 7 rather than Bin 3?

Reviewer #3:

The revision is thoughtful and comprehensive, and most of my concerns have been satisfactorily addressed. The added analyses and clarifications substantially strengthen the manuscript and improve its overall clarity. I believe my remaining concerns are addressable, and would provide the reader with a full and clear picture.

1. TCGA-CGGA comparison and validation claims

In their response letter, the authors state that discrepancies between TCGA and CGGA are expected because the two cohorts differ not only technically but also genetically. However, the manuscript itself does not clearly describe these inter-cohort differences. Instead, TCGA and CGGA are presented as "discovery" and "validation" datasets, respectively, which suggests to the reader that results derived in TCGA should be recapitulated in CGGA.

If these cohorts are indeed intended to function as discovery and validation datasets, it remains unclear to what extent the results actually overlap. While the manuscript demonstrates directional concordance of CSI hazard-ratio effects across cohorts (e.g., pro-tumor CSIs in TCGA tending to remain pro-tumor in CGGA), I was not able to find a quantification of the actual CSI overlap—in other words, how many TCGA CSIs are independently detected or statistically significant in CGGA at a defined threshold.

If I understood correctly, the reported "directionality" is assessed by taking TCGA-derived CSIs and checking whether they have the same direction in CGGA, rather than testing whether they would be recovered independently in CGGA. I therefore ask the authors to provide an explicit replication or overlap metric (e.g., the fraction of CSIs significant in both cohorts, or an overlap statistic such as a Jaccard index), which would allow readers to more directly assess the strength of cross-cohort validation. If the CSIs are not recovered independently, the authors should explicitly state this in the manuscript and explain why. Is this due to the number of samples in CGGA? If so, is there a minimum cohort size required to obtain reliable CSI estimates? Alternatively, are the TCGA and CGGA cohorts too different? If TCGA and CGGA differ genetically or clinically in ways that influence deconvolution performance, IC stability, or CSI reproducibility—as stated in the response letter—this should be clearly articulated in the main text. In that case, referring to TCGA and CGGA as discovery and validation cohorts is potentially misleading, if genetic differences are expected to yield different CSIs. If this is indeed the case, I recommend that the authors (i) specify what genetic or clinical distinctions they are referring to, and (ii) discuss how these differences affect the interpretation and limitations of cross-cohort reproducibility.

2. Minor comment: terminology regarding "cellular interactions"

In the manuscript, the authors state:

"Overall, implementing a novel computational strategy, CSI-TME provides new insights into the cellular interactions in the IDH-mut glioma TME with a potential application in patient stratification for therapeutic interventions."

The revised Introduction does a much better job clarifying that CSIs are not direct cell-cell interactions, but rather joint transcriptional state activities with prognostic relevance. However, this sentence reintroduces ambiguity by referring again to "cellular interactions." I suggest rephrasing this sentence to avoid confusion and to remain consistent with the conceptual framing established earlier in the manuscript.

Reviewer #2

The authors have addressed the majority of my previous concerns regarding the manuscript. However, several issues require further clarification.

We are thankful to the reviewer for their critical feedback on our work and valuable comments that helped to consolidate the findings and strengthen our claims. In this round of revision, we have addressed and clarified the any remaining concerns Please check out point-by-point response below

1. For "Major Comment 1", the authors claimed that they have performed cross-cancer comparison of the CSIs. However, I cannot locate the related results in the section titled "CSI-TME is broadly applicable to multiple cancer types in identifying clinically relevant cell state interactions". Additionally, what does "a median time of 7.3 years 11/18/2025 6:38:00 PM" (line 802-803) mean?

We regret this critical oversight and have now included an additional supplementary figure EV7 showing these results.

The error "a median time of 7.3 years 11/18/2025 6:38:00 PM" at line 802 arose during the pdf conversion. We have taken care of this error in the latest version.

2. For "Minor Comment 2", has Figure 1A been updated? The figure in the revised version appears identical to the original one.

We apologize for the oversight and now have updated the figure 1A.

3. For "Minor Comment 3", I am still unclear about the definitions of Bin 3 and Bin 7. In Figure 1B, the x-axis represents IC2, and the y-axis stands for IC1. So why does low IC1 and high IC2 correspond to Bin 7 rather than Bin 3?

We appreciate this comment and try to clarify it again here. We realize that this confusion arises because of the axis labelling, and indeed as the reviewer pointed out, ICs are mislabeled in figure 1B. We have now shown the IC1 along the X-axis and IC2 along the Y-axis.

Reviewer #3

The revision is thoughtful and comprehensive, and most of my concerns have been satisfactorily addressed. The added analyses and clarifications substantially strengthen

the manuscript and improve its overall clarity. I believe my remaining concerns are addressable and would provide the reader with a full and clear picture.

We are thankful to the reviewer for their critical feedback on our work and valuable comments that helped to consolidate the findings and strengthen our claims. In this round of revision, we have addressed and clarified the any remaining concerns Please check out point-by-point response below

1. TCGA-CGGA comparison and validation claims

In their response letter, the authors state that discrepancies between TCGA and CGGA are expected because the two cohorts differ not only technically but also genetically. However, the manuscript itself does not clearly describe these inter-cohort differences. Instead, TCGA and CGGA are presented as "discovery" and "validation" datasets, respectively, which suggests to the reader that results derived in TCGA should be recapitulated in CGGA.

If these cohorts are indeed intended to function as discovery and validation datasets, it remains unclear to what extent the results actually overlap. While the manuscript demonstrates directional concordance of CSI hazard-ratio effects across cohorts (e.g., pro-tumor CSIs in TCGA tending to remain pro-tumor in CGGA), I was not able to find a quantification of the actual CSI overlap-in other words, how many TCGA CSIs are independently detected or statistically significant in CGGA at a defined threshold. If I understood correctly, the reported "directionality" is assessed by taking TCGA-derived CSIs and checking whether they have the same direction in CGGA, rather than testing whether they would be recovered independently in CGGA. I therefore ask the authors to provide an explicit replication or overlap metric (e.g., the fraction of CSIs significant in both cohorts, or an overlap statistic such as a Jaccard index), which would allow readers to more directly assess the strength of cross-cohort validation.

If the CSIs are not recovered independently, the authors should explicitly state this in the manuscript and explain why. Is this due to the number of samples in CGGA? If so, is there a minimum cohort size required to obtain reliable CSI estimates?

Alternatively, are the TCGA and CGGA cohorts too different? If TCGA and CGGA differ genetically or clinically in ways that influence deconvolution performance, IC stability, or CSI reproducibility-as stated in the response letter-this should be clearly articulated in the main text. In that case, referring to TCGA and CGGA as discovery and validation cohorts is potentially misleading, if genetic differences are expected to yield different CSIs. If this is indeed the case, I recommend that the authors (i) specify what genetic or clinical distinctions they are referring to, and (ii) discuss how these differences affect the interpretation and limitations of cross-cohort reproducibility.

We thank the reviewer for their thoughtful comments. To address the concern on independent detection of CSIs in validation cohort, we have now additionally done what the reviewer suggested, i.e. - we fitted the ICA models directly and independently in the CGGA cohort (without projecting the ICA from TCAG to CGGA), identified the CSIs, and then compared how many were shared with the TCGA cohort. Encouragingly, we found that 82 / 160 TCGA interactions (51%) were detected independently in the CGGA cohort with a p-value threshold of 0.05.

We have now added these results to the appendix file and referred to it in the main manuscript at line 249.

2. Minor comment: terminology regarding "cellular interactions"

In the manuscript, the authors state:

"Overall, implementing a novel computational strategy, CSI-TME provides new insights into the cellular interactions in the IDH-mut glioma TME with a potential application in patient stratification for therapeutic interventions."

The revised Introduction does a much better job clarifying that CSIs are not direct cell-cell interactions, but rather joint transcriptional state activities with prognostic relevance. However, this sentence reintroduces ambiguity by referring again to "cellular interactions." I suggest rephrasing this sentence to avoid confusion and to remain consistent with the conceptual framing established earlier in the manuscript.

We agree with the reviewer and regret the confusion. We have now changed the word cellular interactions to cell state interactions.

Editorial Comments –

On a more editorial level:

1. Please remove all main and supplementary figures from the manuscript file and upload them as separate, production-quality figure files. Figure legends should be provided at the end of the manuscript. Supplementary figures should be renamed Expanded View (EV) Figures, which will be displayed as collapsible/expandable items online. EV Figures should be cited in the text as Figure EV1, Figure EV2, etc., and their corresponding legends should be included in the main manuscript immediately following the legends for the main figures.

We have made these changes.

2. Please include up to five keywords in the manuscript file.

We have included the following key words

IDH-mut glioma, Tumor microenvironment, cell state interactions, independent component analysis, homeostasis

3. Remove "Authors' contribution" section from the manuscript file.

We have made this change

4. The references need to be formatted according to the Molecular Systems Biology reference style. Please list up to 10 co-authors of a paper before adding et al. in the reference list. Citations should be listed in alphabetical order. DOI numbers should be removed for published articles.

We have made this change.

5. The supplementary tables are complex and should be updated to Dataset EV1-Dataset EV3 throughout (including source file names, titles in the system, and callouts in the manuscript). Legends must be provided on a separate sheet/tab within each dataset file.

We have made this change.

6. We updated our journal's competing interests policy in January 2022 and request authors to consider both actual and perceived competing interests. Please review the policy (<https://link.springer.com/partners/embo-press/editorial-policies#Competing%20interest%20disclosures>) and update your competing interests if necessary.

Please use the heading "Disclosure statement and competing interests".

We have added a disclosure statements declaring no competing interests

7. Before submitting your revision, primary datasets and computer code produced in this study need to be deposited in an appropriate public database.

Please remember to provide a reviewer password if the datasets are not yet public. The accession numbers and database should be listed in a formal "Data Availability" section (placed after Materials & Method) that follows the model below (see also <https://link.springer.com/partners/embo-press/editorial-policies#Data%20availability%20statement>). Please note that the Data Availability Section is restricted to new primary data that are part of this study.

Data availability

- RNA-Seq data: Gene Expression Omnibus GSE46843

(<https://www.ncbi.nlm.nih.gov/geo/query/acc.cgi?acc=GSE46843>)

- [data type]: [name of the resource] [accession number/identifier/doi] ([URL or identifiers.org/DATABASE:ACCESSION])

We have provided the code for CSI-TME on GitHub. There were no primary datasets produced in this study.

Additional information on source data and instruction on how to label the files are available < <https://link.springer.com/journal/44320/submission-guidelines#cms-Source-data> >

All the source data are uploaded.

9. Please provide a "standfirst text" summarizing the study in one or two sentences (approximately 250 characters, including space), three to four "bullet points" highlighting the main findings and a " visual abstract" (550px width and 400-600 px height, PNG format) to highlight the paper on our homepage.

Please refer to published papers for examples.

We provide the following standfirst text.

This study introduces **CSI-TME**, a computational framework that leverages large bulk transcriptomic cancer cohorts with matching clinical data to identify **clinically relevant cell state interactions in the tumor microenvironment**.

Additionally, we have now provided an abstract graphics describing the study

10. Please download and fill our Reagents and Tools Table template (.docx), which you can find in our author guidelines: <https://link.springer.com/journal/44320/submission-guidelines#structuredmethods>.

We now provide the requested Table

OK

11. Please address the following issues in figure legends:

- Please note that the figure 4I, J is mislabeled as figure 4F, G in the manuscript. This needs to be rectified.

We have corrected the figure legends.

- Please note that the exact p values are not provided in the legends of figures 3B, D; 4G, I; 5J, 6C, 7C; S2 B, C, D, E, I, J, L; S2D, K, L

We have now provided the exact p-values for figure 3B,3D, 4G, 4I, 7C and S2D. For the remaining figures, there is following explanation -

– In figure 6C- the labelling $p < 0.05$ indicates the significant threshold that we used to call an interaction as enriched in one set of recurrent tumors. We have now changed the labelling to avoid the confusion.

– Exact p-value is already reported in figure S2B and C.

– In figure S2E,J, we did not perform any hypothesis testing but simply reported the distribution correlation between hazard ratios between discovery and validation cohort for randomized input data (S2E), and the number of interactions across different cell types (S2D).

- Please indicate the statistical test used for data analysis in the legends of figures 2G, 3B, D; 4G, H, I, J; 5H-J; 6C; S2 D

We have made these changes in the figure legends.

- Please note that the box plots need to be defined in terms of minima, maxima, centre, bounds of box and whiskers, and percentile in the legends of figures 2G, 3B, D; 4A, H; 6A, B; 7D, E; 8B, C, D, E, F; S2 B, C, D, E, I, J, K, L; S3B, E; S4A, S5A

We have made these change in the figure legends.

- Please note that information related to n is missing in the legends of figures 2G, 3B, D; 4A, H; 6A, B; 7D, E; 8B, C, D, E, F; S3B, E; S4A, S5A

TBD we have added the sample sizes to these figures

12. The co-author's email address bounced: Bharati Mehani (bharati.mehani@nih.gov). Please either remove this author from the author list in the system and re-add her using a valid email address, or provide the updated email address so we can revise the author's account accordingly.

We have updated the email address for this author.

13. When you resubmit your manuscript, please download our CHECKLIST (<https://media.springernature.com/original/springer-cms/rest/v1/content/27825796/data/v1>) and include the completed form in your submission.

Please note that the Author Checklist will be published alongside the paper as part of the transparent process .

We have completed the author checklist.

11th Feb 2026

Manuscript number: MSB-2025-13172RRR

Title: Identifying Prognostic Cell State Interactions in the Tumor Microenvironment of IDH-Mutant Gliomas

Dear Dr. Hannenhalli,

Thank you again for sending us your revised manuscript. We are now satisfied with the modifications made and I am pleased to inform you that your paper has been accepted for publication.

You may qualify for financial assistance for your publication charges - either via a Springer Nature fully open access agreement or an EMBO initiative. Check your eligibility: <https://link.springer.com/journal/44320/how-to-publish-with-us>

Sincerely,
Jingyi

Jingyi Hou, PhD
Senior Editor
Molecular Systems Biology

>>> Please note that it is Molecular Systems Biology policy for the transcript of the editorial process (containing referee reports and your response letter) to be published as an online supplement to each paper. If you do NOT want this, you will need to inform the Editorial Office via email immediately. More information is available here: <https://link.springer.com/partners/embo-press/editorial-policies#Peer%20review>